# FINE-GRAINED TOKEN ALLOCATION VIA OPERATION PRUNING FOR EFFICIENT MLLMS

## ABSTRACT

Token reduction accelerates Multimodal Large Language Models (MLLMs) by reducing excessive tokens, but overlooks structural redundancy differences, where critical and redundant modules process identical token loads. For fine-grained computation control, we define an "operation" as the computation for a module to process a group of tokens and introduce the operation pruning framework to enable modules to selectively process tokens. Built on this framework, we propose **D**epth-wise **O**peration **P**runing (**DOP**), a data-driven method that searches for strategies to prune redundant operations and save computational budget for critical modules to process more tokens than uniform allocation by minimizing divergence from the original model's output probability distribution on a small validation set while satisfying computational constraints. For efficient optimization, DOP applies depth-wise pruning to reduce policy space and uses an additive approximation to minimize required validation runs. Depth-wise pruning partitions operations by module type and token group, and prunes operations in deeper layers before those in shallower layers within each module-group pair. The additive approximation obtains individual divergences by independently varying each policy parameter, and then sums them to approximate the joint divergence of simultaneously changing all policy parameters, reducing required validation runs from exponential to linear with respect to the number of policy parameters. Comprehensive evaluations show that DOP establishes new state-of-the-art performance across 6 MLLMs and 13 benchmarks against 12 baselines. On LLaVA-Next-7B, DOP achieves 86% TFLOPS reduction and 83% latency reduction on real GPU with only 1% performance loss. Our extensive ablation studies further demonstrate DOP's data and time efficiency as well as strong generalization capabilities.

## 1 INTRODUCTION

Multimodal Large Language Models (MLLMs) (Liu et al., 2024b;a; Li et al., 2023b; Team et al., 2023; Bai et al., 2023b; Chen et al., 2023b) face significant computational overhead primarily in their language model decoders (Achiam et al., 2023; Touvron et al., 2023; Peng et al., 2023; Abdin et al., 2024), which dramatically outweigh the lightweight visual encoders (*e.g.*, in LLaVA-1.5, 7B/13B decoder parameters versus 0.3B encoder parameters), due to excessive tokens increasing the computation in Multi-Head Attention (MHA) and Multi-Layer Perceptron (MLP) modules. As the majority of tokens are from visual input, reducing visual tokens is a common practice to accelerate MLLMs by reducing MHA/MLP computations, *e.g.*, token pruning methods (Li et al., 2024; Yao et al., 2024; Chen et al., 2024; Zhang et al., 2024c), built-in token reduction like resizing input images to smaller resolutions (Bai et al., 2025; Shi et al., 2025b). However, they often make all modules in the decoder process the same visual tokens, overlooking structural redundancy differences where certain modules contribute more critically to outputs than others.

To achieve better performance-efficiency tradeoffs by allocating more tokens to critical modules and less for redundant ones, we introduce the operation pruning framework, where an "operation" is the computation for a module to process a group of tokens, and pruning means having that module selectively not process this group of tokens. Built on this framework, we propose **D**epth-wise **O**peration **P**runing (**DOP**), a data-driven method that prunes redundant operations with minimal impact on outputs and saves computational budget for critical modules to process more tokens than uniform allocation, as shown in Figure 1.

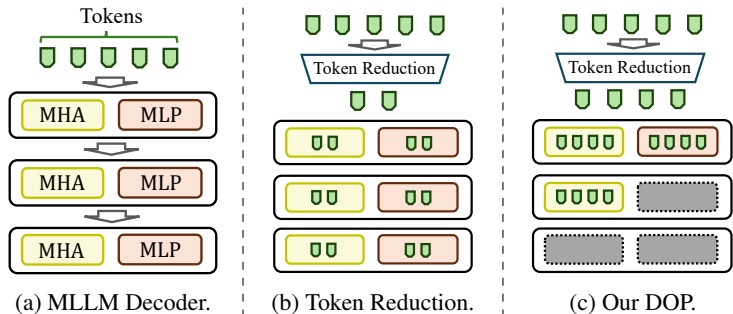

(a) MLLM Decoder.  (b) Token Reduction.  (c) Our DOP.

Figure 1: Comparison of our DOP and token reduction: (a) Computational overhead in MLLM decoders primarily stems from MHA and MLP modules processing excessive tokens. (b) Token reduction (Chen et al., 2024; Zhang et al., 2024a; 2025) has all modules process the same token load regardless of structural redundancy differences. (c) Our DOP prunes redundant operations and saves computational budget for critical modules to process more tokens than uniform allocation.

DOP optimizes pruning policies by minimizing divergence from the original model's output probability distribution on a small validation set while satisfying computational constraints. For efficient optimization, DOP employs depth-wise pruning constraints and additive approximation. Depth-wise pruning reduces the policy space complexity from exponential to quadratic with respect to layer count by enforcing that operations for tokens in the same group within each module type (MHA or MLP) are pruned in depth order—deeper operations before shallower operations. Thus, a DOP policy for a single group of tokens consists of the token count within the group and the processing depth for each module type, where tokens are entirely excluded from each module type's computation beyond that type's specified depth. The optimization cost is further reduced by applying an additive approximation that replaces joint divergence with the sum of individual divergences from changing each policy parameter independently, reducing the required validation runs from exponential to linear with respect to the number of policy parameters.

Extensive experiments validate DOP as an effective, efficient, and versatile MLLM acceleration method through comprehensive evaluations across 6 different MLLMs, 13 benchmarks, and comparisons against 12 baselines. When only pruning visual operations, as all baselines only prune visual tokens, DOP consistently outperforms state-of-the-art baselines by preserving up to 7% more performance. Notably, on LLaVA-NeXT-7B, DOP reduces TFLOPs by 77% with no performance loss on average, and reduces TFLOPs by 86% with just 1% loss as well as 83% prefilling CUDA latency reduction. DOP is highly flexible and consistently improves performance with three different token reduction methods. Moreover, according to our ablation, DOP demonstrates both data efficiency, requiring as few as 25 validation samples and two minutes to optimize a good policy, and strong generalization capabilities, performing well when transferred to different tasks and models.

Our main contributions are summarized as follows:

1. We fundamentally extend MLLM acceleration beyond token reduction to fine-grained operation pruning, thereby enabling precise and fine-grained computation reduction via module-level token allocation based on both token redundancy and structural redundancy.

2. We propose Depth-wise Operation Pruning (DOP), an effective method for optimizing operation pruning policies, with depth-wise pruning constraints and additive approximation to make optimization extremely time- and data-efficient.

3. Our comprehensive evaluations and ablation studies show that DOP achieves state-of-the-art performance-efficiency tradeoffs with strong cross-task and cross-model generalization.

## 2 RELATED WORK

**Token Reduction** reduces MLLM (Liu et al., 2024b;a; Li et al., 2023b; Dai et al., 2023; Team et al., 2023; Bai et al., 2023a;b; Chen et al., 2023b; Liu et al., 2023a; Abdin et al., 2024; Lin et al., 2023b; Zhu et al., 2025; Bai et al., 2025) computation by reducing visual tokens to minimize the computation for MHA and MLP modules to process these tokens, with most being token pruning methods that directly discard redundant tokens (Chen et al., 2024; Zhang et al., 2024c; Lin et al.,

2024; Ye et al., 2024; Zhang et al., 2024b; Yang et al., 2024; Xing et al., 2024; Alvar et al., 2025; Wen et al., 2025; Zhang et al., 2025; Zeng et al., 2025), while others like token merging combine redundant tokens into critical tokens (Shang et al., 2024; Yang et al., 2024; Zhang et al., 2024c). Prior works mainly explore "what" tokens to prune by finding redundancy indicators. Early methods used LLM decoder attention (Chen et al., 2024; Zhang et al., 2024c; Ye et al., 2024; Xing et al., 2024), but later works noted position bias and switched to [CLS] attention (Zhang et al., 2024a; Yang et al., 2024; Shang et al., 2024) and feature diversity (Wen et al., 2025; Alvar et al., 2025). Recent CDPruner (Zhang et al., 2025) achieves state-of-the-art with conditional diversity. Additionally, many MLLMs have built-in token reduction like visual input resizing (Bai et al., 2025; Zhu et al., 2025; Shi et al., 2025a).

However, these methods rarely consider structural redundancy. Only a few progressive pruning works (Ye et al., 2024; Xing et al., 2024; Zhang et al., 2024c) explore this by allocating more tokens to shallower layers and fewer to deeper layers, performing well under loose budgets but poorly under tight budgets due to coarse-grained layer-level redundancy that overlooks module-level differences. For instance, a deeper MHA may be more critical than a shallower MLP. Additionally, progressive pruning has usage limitations like FlashAttention incompatibility or requiring redundancy ranking, limiting compatibility with diversity-based or resizing methods.

**KV caching** (Kwon et al., 2023) accelerates MLLM inference by reusing computed key-value pairs of processed tokens, dividing the inference process into a prefilling stage, where all tokens are fully processed to generate the first token, and subsequent decoding stages, where KV cache exempts certain DOP components (*e.g.*, MHA and MLP modules) from full-token processing. Our DOP primarily optimizes the prefilling stage, which aligns with prior token pruning approaches (Shang et al., 2024; Chen et al., 2024; Zhang et al., 2024c; Lin et al., 2024; Ye et al., 2024; Zhang et al., 2024b; Yang et al., 2024; Xing et al., 2024). Zhong et al. (2024) also confirms prefilling is compute-bound, while decoding is memory-bandwidth-bound, validating our focus on prefilling efficiency.

We discuss additional related work in Appendix B, including MLLM architectures and the distinction between DOP and Neural Architecture Search.

## 3 DEPTH-WISE OPERATION PRUNING

We aim to eliminate redundant MLLM computations while minimizing the performance impact. We target the parameter-dense decoder ($20\times$ larger than the visual encoder) during the compute-bound prefilling stage (Zhong et al., 2024), following Shang et al. (2024); Chen et al. (2024); Yang et al. (2024), by decomposing computation into operations and pruning the most redundant ones.

To formalize this approach, we first provide a general formulation without introducing specific settings. Let $\mathbf{o}$ denote an operation and $\mathcal{O} = \{\mathbf{o}_1, \mathbf{o}_2, \dots\}$ denote the set of all operations, where each operation is the computation for an MHA/MLP module to process a group of tokens. Then, operation pruning is represented as $\mathcal{O} \setminus \mathcal{P}$, where any subset $\mathcal{P} \subseteq \mathcal{O}$ can be a pruning policy. We aim to find the optimal policy $\mathcal{P}^*$ that maximizes performance while meeting computation constraints. Given $\tau$ as the maximum computation allowed, the pruning policy optimization is:

$$\mathcal{P}^* = \underset{\mathcal{P} \subseteq \mathcal{O}}{\arg\max} \quad \text{Performance}(\mathcal{O} \setminus \mathcal{P}) \quad \text{s.t.} \quad \text{Computation}(\mathcal{O} \setminus \mathcal{P}) \leq \tau \tag{1}$$

Section 3.1 below will concretize the operation pruning framework with analysis, while Section 3.2 introduces our optimization details for the pruning policy.

### 3.1 OPERATION PRUNING FRAMEWORK

As shown in Figure 2a, we decompose MLLM decoder computations along three dimensions into atomic operations: token groups, layer positions, and modules. Each operation is defined as:

$$\mathbf{o} = (g, l, m), \quad \mathbf{o} \in \mathcal{O} = \mathcal{G} \times \mathcal{L} \times \mathcal{M}. \tag{2}$$

Each operation $(g, l, m)$ represents the computation for the $m$ module in layer $l$ to process the tokens in group $g$. $\mathcal{O}$ is the set of all operations, $\mathcal{G} = \{\mathbf{g}_s, \mathbf{g}_v, \mathbf{g}_t, \mathbf{g}_o\}$ is the set of token groups based on their sources: system, visual, text, and output, respectively, $\mathcal{L} = \{1, \dots, L\}$ represents the set of layer indices, and $\mathcal{M} = \{MHA, MLP\}$ is the set of module types. These are detailed below.

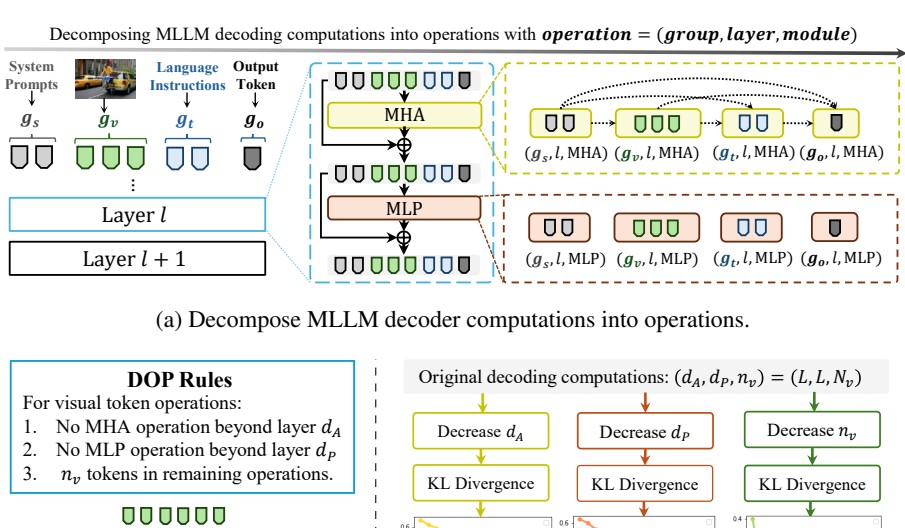

(a) Decompose MLLM decoder computations into operations.

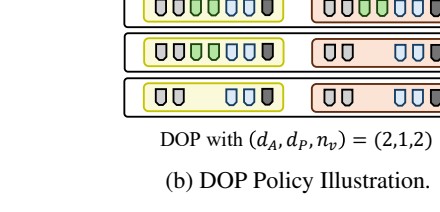

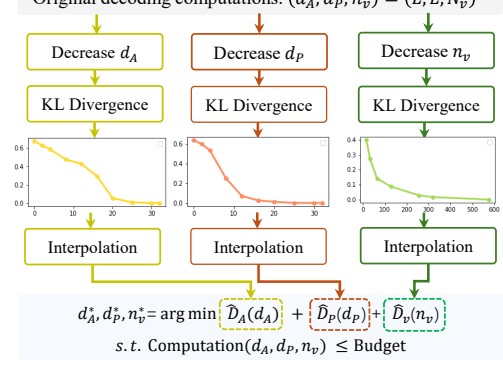

(b) DOP Policy Illustration.

(c) DOP Policy Optimization.

Figure 2: **Overview of DOP.** (a) DOP decomposes MLLM decoder computations into operations defined by $(group, layer, module)$ as atomic computation carriers. Tokens are categorized into four groups based on input sources, and we prune operations for the visual token group in MHA and MLP modules. (b) DOP employs depth-wise pruning: visual tokens bypass MHA operations beyond layer $d_A$ and MLP operations beyond layer $d_P$, with remaining operations processing $n_v$ selected visual tokens. (c) DOP reduces optimization cost through additive approximation by replacing joint divergence with the sum of individual divergences measured by independently reducing $d_A$, $d_P$, $n_v$ and computing respective KL divergences from original model's output probability distribution. Evaluation points are sparsely sampled, with other points obtained through interpolation.

**Token Groups.** DOP groups tokens to avoid excessive operations from token-by-token division, reducing optimization cost. We categorize tokens into four groups based on input sources: $\mathbf{g}_s$ from system prompts, $\mathbf{g}_v$ from visual inputs, consisting of $n_v$ critical visual tokens selected through existing visual token reduction methods (Zhang et al., 2024a; 2025), $\mathbf{g}_t$ from language instructions, and output tokens $\mathbf{g}_o$. While our framework supports pruning operations for all four groups, we primarily focus on pruning operations for visual tokens $\mathbf{g}_v$ due to their prevalence and for fair comparison with baselines only pruning visual tokens (Chen et al., 2024; Zhang et al., 2024a; 2025).

**Visual Token Reduction.** To get the visual token group $\mathbf{g}_v$, we employ existing visual token reduction methods to reduce the original $N_v$ visual tokens to $n_v$ critical tokens. DOP directly integrates with different visual token reduction methods. In this work, we mainly adopt two token pruning methods: VisPruner (Zhang et al., 2024a) based on [CLS] attention and similarity, and CDPruner (Zhang et al., 2025) based on conditional diversity, along with visual input resizing for Qwen2.5-VL (Bai et al., 2025) and InternVL3 (Zhu et al., 2025).

**Layer Positions and Module Types.** We use layer position $l$ and module type $m$ to identify specific MHA/MLP operations. The layer position $l \in \{1, \ldots, L\}$ where $L$ is the number of decoder layers. Within each layer, we focus on two modules consuming most computation within a layer: *MHA* for mutual token feature updating, and *MLP* for independent token-wise feature transformation.

**Policy Space Size.** While our fine-grained operation pruning paradigm enables more precise computation control, it exponentially expands the policy space. For a 32-layer decoder with $|\mathcal{M}| = 2$ module types and $N_v = 576$ visual tokens, there are $|\mathcal{O}_v| = 32 \times 2 = 64$ visual operations. With binary decisions for each operation and 577 possible token counts (0 to 576), the total pruning con-

figurations become: $2^{64} \times 577 \approx 1.0 \times 10^{22}$. Such an enormous policy space necessitates efficient optimization to avoid prohibitive computational costs.

### 3.2 Pruning Policy Optimization

In this section, we introduce DOP's policy optimization for pruning visual operations based on depth-wise pruning and additive approximation, as shown in Figures 2b and 2c. Note that we focus on visual token operations since they are most numerous and constitute the majority of computational overhead, while also enabling fair comparison with baselines that only prune visual tokens.

#### 3.2.1 Depth-wise Operation Pruning

Deeper layers are typically more redundant in LLMs (Gromov et al.). While deeper operations are not necessarily more redundant than shallower ones across different module types or token groups, we observe that for the same group-module type $(g, m)$, a deeper operation $(g, l_2, m)$ remains generally more redundant than a shallower operation $(g, l_1, m)$ with $l_2 > l_1$.

**Depth-wise Pruning Constraint**. For each module type $m \in \{MHA, MLP\}$ and token group $g$, operations are pruned consecutively from the deepest layer. Operation $(g, l, m)$ being pruned implies that operations $(g, l+1, m), \ldots, (g, L, m)$ are also pruned.

**Depth-wise Pruning Policy**. When applying depth-wise pruning only for visual token group $\mathbf{g}_v$ operations, a pruning policy can be parameterized by 3 parameters:

$$(d_A, d_P, n_v), \text{ where } d_A, d_P \in \{0, \ldots, L\}, n_v \in \{0, \ldots, N_v\}, \tag{3}$$

representing MHA depth, MLP depth, and the number of visual tokens, respectively. Here, $L$ is the total number of layers and $N_v$ is the original visual token count. As shown in Figure 2b, all MHA operations $\{(\mathbf{g}_v, l, MHA)\}_{l=d_A+1}^{L}$ beyond layer $d_A$ and all MLP operations $\{(\mathbf{g}_v, l, MLP)\}_{l=d_P+1}^{L}$ beyond layer $d_P$ are pruned:

$$\mathcal{P}(d_A, d_P) = \{(\mathbf{g}_v, l, MHA)\}_{l=d_A+1}^{L} \cup \{(\mathbf{g}_v, l, MLP)\}_{l=d_P+1}^{L}. \tag{4}$$

The remaining visual operations $\{(\mathbf{g}_v, l, MHA)\}_{l=1}^{d_A} \cup \{(\mathbf{g}_v, l, MLP)\}_{l=1}^{d_P}$ process only $n_v$ visual tokens in $\mathbf{g}_v$ that are retained after token reduction.

In effect, this policy creates a "visual processing cutoff" where visual tokens are first reduced to $n_v$ before decoder, then processed through shallow MHA and MLP modules up to depths $d_A$ and $d_P$ respectively, while deeper MHA and MLP modules operate without any visual information.

**Policy Space Size.** For a 32-layer decoder with $|\mathcal{M}| = 2$ and $N_v = 576$, depth-wise pruning reduces possible pruning policies from $1.0 \times 10^{22}$ to $(|\mathcal{L}| + 1)^{|\mathcal{M}|} \times (N_v + 1) = 33^2 \times 577 = 628,353$. However, this remains huge, motivating further optimization complexity reduction.

#### 3.2.2 Policy Optimization with Additive Approximation

The optimization of the three DOP policy parameters constitutes a combinatorial optimization problem where a naive approach is to exhaustively evaluate all possible policies, formulated as follows:

$$(d_A^*, d_P^*, n_v^*) = \arg \min_{\substack{d_A, d_P \in \{0, 1, \ldots, L\} \\ n_v \in \{0, 1, \ldots, N_v\}}} \mathcal{D}(d_A, d_P, n_v) \quad \text{s.t.} \quad \text{TFLOPs}(d_A, d_P, n_v) \leq \tau, \tag{5}$$

where $L$ is the total number of decoder layers and $N_v$ is the original visual token count. We minimize $\mathcal{D}(d_A, d_P, n_v)$, the divergence from the original model's output distribution caused by pruning policy $(d_A, d_P, n_v)$, as greater deviation typically indicates worse performance, and divergence can be computed without labels, reducing data leakage concerns. Theoretical FLOPs (TFLOPs) is used to measure computational cost after pruning with budget $\tau$. We detail the computation of divergence and TFLOPs in Appendices C.2 and C.3 respectively.

To solve Equation (5), we must evaluate $\mathcal{D}(d_A, d_P, n_v)$ for all possible policies with $(L+1)^2 \times (N_v + 1) + 1$ model runs over the validation set, while subsequent TFLOPs computation, exhaustive search through recorded divergences, and comparisons can be completed with negligible cost. Therefore, we reduce optimization cost by minimizing the required validation runs.

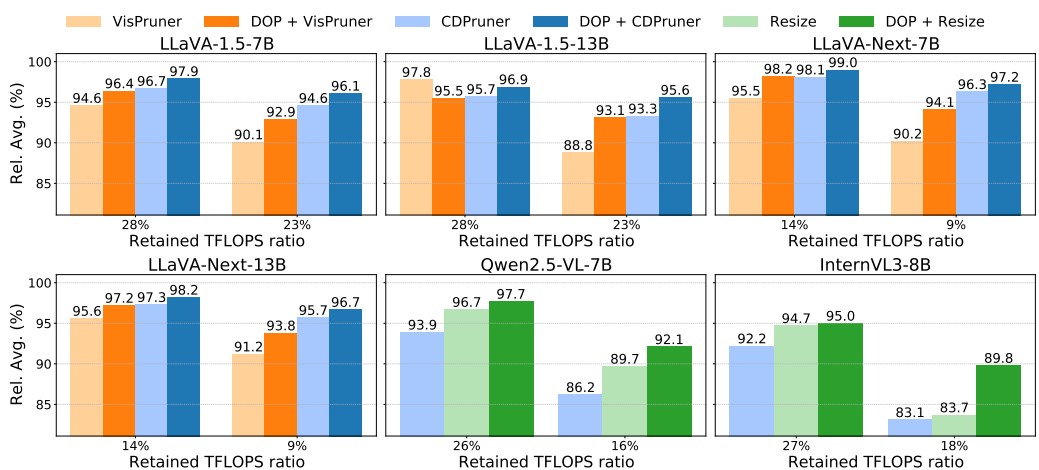

Figure 3: Overview of DOP performance across LLaVA-1.5-7B&13B (Liu et al., 2024b), LLaVA-Next-7B&13B (Liu et al., 2024a), Qwen2.5-VL-7B (Bai et al., 2025) and InternVL3-8B (Zhu et al., 2025). Rel. Avg. is the mean performance ratio relative to the original model, averaged across benchmarks used for each model. DOP is flexible with various token reduction methods and consistently boosts the performance of corresponding baselines.

**Additive Approximation.** We apply an additive approximation by replacing the joint divergence $\mathcal{D}(d_A, d_P, n_v)$ in Equation (5) with the sum of individual divergences:

$$\mathcal{D}_{\text{sum}}(d_A, d_P, n_v) = \mathcal{D}_A(d_A) + \mathcal{D}_P(d_P) + \mathcal{D}_v(n_v) = \mathcal{D}(d_A, L, N_v) + \mathcal{D}(L, d_P, N_v) + \mathcal{D}(L, L, n_v).$$

We only need $L+1$, $L+1$, and $N_v+1$ model runs for $\mathcal{D}_A(d_A)$, $\mathcal{D}_P(d_P)$, and $\mathcal{D}_v(n_v)$ respectively, resulting in $2L + N_v + 4$ validation runs including one for the original model.

Since solving Equation (5) does not require exact $\mathcal{D}(d_A, d_P, n_v)$ values but only relative policy rankings based on it, the key to yielding good policies when using $\mathcal{D}_{\text{sum}}$ to replace $\mathcal{D}$ is ensuring that $\mathcal{D}_{\text{sum}}$ results in largely consistent policy rankings with $\mathcal{D}$ such that the following largely holds:

$$\mathcal{D}_{\text{sum}}(d_A, d_P, n_v) > \mathcal{D}_{\text{sum}}(d'_A, d'_P, n'_v) \Rightarrow \mathcal{D}(d_A, d_P, n_v) > \mathcal{D}(d'_A, d'_P, n'_v), \qquad (6)$$

which is validated by sampling policies, computing both $\mathcal{D}$ and $\mathcal{D}_{\text{sum}}$, and calculating Spearman's rank correlation between their rankings in Appendix E.1. Results show strong positive correlations across models ($\rho \geq 0.799$, $p < 10^{-48}$), supporting our additive approximation.

**Final Policy Optimization.** Besides additive approximation, we employ sparse sampling for divergence evaluation with interpolation (Appendix C.1) and add visual token count constraints to avoid overly conservative operation pruning (Appendix C.4), yielding the final optimization:

$$(d_A^*, d_P^*, n_v^*) = \arg \min_{(d_A, d_P, n_v)} \hat{\mathcal{D}}_A(d_A) + \hat{\mathcal{D}}_P(d_P) + \hat{\mathcal{D}}_v(n_v) \qquad (7)$$

$$\text{s.t.} \quad \text{TFLOPs}(d_A, d_P, n_v) \leq \tau$$

$$n_v \geq n_{\min}$$

where $\hat{\mathcal{D}}_A$, $\hat{\mathcal{D}}_P$, $\hat{\mathcal{D}}_v$ are interpolated individual divergences for MHA depth, MLP depth, and visual token count respectively and $n_{\min}$ is the minimum visual token count.

With $I$ sampling points per parameter, we need only $3 \cdot I + 1$ validation runs as a one-time cost. Once completed, optimizing policies for different constraints only requires solving Equation (7) by enumerating policies, computing TFLOPs, and comparing the sum of recorded individual divergences for policies that satisfy the constraints with negligible cost compared to validation runs.

## 4 EXPERIMENTS

In this section, we evaluate DOP on 6 MLLM models and 13 multimodal benchmarks against 12 baselines, report real GPU acceleration and optimization costs, demonstrate flexibility with three token reduction methods, and conduct extensive ablations on cross-task and cross-architecture generalization as well as optimization data-efficiency.

| Method | VQA$^{v2}$ | GQA | VizWiz | SQA$^I$ | Seed$^I$ | VQA$^{Text}$ | POPE | MME | MMB | MMB$^{CN}$ | Rel. Avg. |
|---|---|---|---|---|---|---|---|---|---|---|---|
| *Upper Bound: 100% TFLOPs (= 576 Visual Tokens Every Layer)* | | | | | | | | | | | |
| LLaVA-1.5-7B | 78.5 | 61.9 | 50.1 | 69.5 | 66.2 | 58.2 | 85.9 | 1506.5 | 64.7 | 58.1 | 100% |
| *Retain 28% TFLOPs (= 64 Visual Tokens Every Layer)* | | | | | | | | | | | |
| FastV (Chen et al., 2024) | 55.9 | 46.0 | 49.1 | **70.1** | 51.9 | 51.6 | 35.5 | 973.5 | 50.1 | 42.1 | 76.7% |
| FitPrune (Ye et al., 2024) | 63.7 | 52.3 | 51.1 | 68.0 | 55.8 | 51.2 | 73.6 | 1287.5 | 58.5 | 49.7 | 88.5% |
| DivPrune (Alvar et al., 2025) | 74.1 | 57.5 | **53.6** | 68.0 | 61.7 | 54.5 | 85.5 | 1334.7 | 60.1 | 52.3 | 95.0% |
| VisPruner (Zhang et al., 2024a) | 72.7 | 55.4 | 53.3 | 69.1 | 58.5 | 55.8 | 80.4 | 1369.9 | 61.3 | 55.1 | 94.6% |
| **DOP$_V$ (Ours)** | 74.6 | 57.1 | **53.6** | 69.2 | 60.3 | **56.5** | 83.1 | 1394.6 | 61.6 | **56.5** | 96.4% |
| CDPruner (Zhang et al., 2025) | 75.4 | 58.6 | 53.4 | 68.1 | 62.5 | 55.3 | **87.5** | 1415.1 | 61.1 | 53.2 | 96.7% |
| **DOP$_{CD}$ (Ours)** | **76.2** | **59.6** | 53.4 | 68.3 | **63.6** | 55.8 | **87.6** | **1436.3** | **62.4** | 55.1 | **97.9%** |
| *Retain 23% TFLOPs (= 32 Visual Tokens Every Layer)* | | | | | | | | | | | |
| DivPrune (Alvar et al., 2025) | 71.2 | 54.9 | 53.3 | 68.6 | 58.7 | 52.9 | 81.5 | 1284.9 | 57.6 | 49.1 | 91.8% |
| VisPruner (Zhang et al., 2024a) | 67.7 | 52.2 | 53.0 | **69.2** | 54.3 | 53.9 | 72.7 | 1271.0 | 58.4 | 52.7 | 90.1% |
| **DOP$_V$ (Ours)** | 71.0 | 54.8 | **53.9** | 69.1 | 56.7 | **54.5** | 79.6 | 1306.5 | 59.4 | **53.7** | 92.9% |
| CDPruner (Zhang et al., 2025) | 73.6 | 57.0 | 53.1 | **69.5** | 60.9 | 53.2 | **87.9** | 1373.0 | 59.6 | 49.6 | 94.6% |
| **DOP$_{CD}$ (Ours)** | **74.7** | **58.1** | 53.6 | 69.3 | **62.2** | 54.2 | **87.9** | 1397.5 | 60.1 | 52.2 | **96.1%** |

Table 1: **Performance comparison on LLaVA-1.5-7B (Liu et al., 2024b).** Rel. Avg. is the mean performance ratio relative to the original model averaged across all benchmarks. **Bold** indicates the best at each TFLOPs ratio. DOP$_V$ uses VisPruner to reduce visual tokens, while DOP$_{CD}$ uses CD-Pruner. Our comparison matches actual TFLOPs and notes the per-layer token counts that achieve these TFLOPs. See Tables 8 and 9 for full comparison across LLaVA-1.5-7B&13B.

## 4.1 EXPERIMENTAL SETUP

**MLLM Models.** We apply DOP to 6 MLLM models, including LLaVA-1.5 (7B & 13B) (Liu et al., 2024b) and LLaVA-Next (7B & 13B) (Liu et al., 2024a), Qwen2.5-VL (7B) (Bai et al., 2025) and InternVL3-8B (Zhu et al., 2025). Model details are in Appendix D.1.

**Evaluation Benchmarks.** We select 13 benchmarks for evaluation, including 9 general VQA tasks: VQA$^{V2}$ (Goyal et al., 2017), VizWiz (Gurari et al., 2018), ScienceQA (SQA$^I$) (Saikh et al., 2022), GQA (Hudson & Manning, 2019), SeedBench (Seed$^I$) (Li et al., 2023a), MME (Fu et al., 2023), POPE (Li et al., 2023c), MMBench (MMB), MMBench-CN (MMB$^{CN}$) (Liu et al., 2023b); and 4 text-oriented VQA tasks: TextVQA(VQA$^{text}$) (Singh et al., 2019) AI2D (Kembhavi et al., 2016) ChartQA (Masry et al., 2022), and OCRBench (Liu et al., 2024c). Details are in Appendix D.2.

**Baselines.** We compare DOP with 12 baselines, including FastV (Chen et al., 2024), FitPrune (Ye et al., 2024), PyramidDrop (PDrop) (Xing et al., 2024), SparseVLM (Zhang et al., 2024c), PruMerge (Shang et al., 2024), VisionZip (Yang et al., 2024), TRIM (Song et al., 2025), DART (Wen et al., 2025), DivPrune (Alvar et al., 2025), VisPruner (Zhang et al., 2024a) and CDPruner (Zhang et al., 2025). PDrop, FitPrune, and SparseVLM are progressive pruning methods, with FitPrune using the same validation set for pruning policy optimization. Details are in Appendix D.3.

**DOP Configurations.** DOP only prunes visual token operations, sampling $K = 6$ points for visual token count, MHA depth, and MLP depth respectively to evaluate output divergence on 100 randomly selected TextVQA training samples for each model. Each model applies the same optimized policy across all benchmarks at each TFLOPs ratio. For LLaVA, we evaluate DOP$_V$ and DOP$_{CD}$ with VisPruner and CDPruner for visual token reduction, while for Qwen2.5-VL and InternVL3, we evaluate DOP$_R$ with input image resizing. Additional configuration details are in Appendix D.4.

## 4.2 RESULTS AND COMPARISONS

**Overall Performance.** Figure 3 shows DOP's performance across all selected models. DOP consistently improves different token reduction methods and outperforms state-of-the-art baseline CD-Pruner by up to 7%. The advantages of DOP become more pronounced at lower TFLOPs ratios, demonstrating that tighter computational budgets require more precise and fine-grained control over computation, validating DOP's improvement direction. All policies are inTables 15 and 16

**LLaVA-series Models.** Tables 1 and 2 present selected evaluation results on LLaVA-1.5-7B (Liu et al., 2024b) and LLaVA-Next-7B (Liu et al., 2024a), with full comparison across LLaVA-1.5-7B&13B and LLaVA-Next-7B&13B in Tables 8 to 11. We evaluate DOP$_V$ and DOP$_{CD}$ with Vis-

| Method | VQA$^{v2}$ | GQA | VizWiz | SQA$^I$ | Seed$^I$ | VQA$^{Text}$ | POPE | MME | MMB | MMB$^{CN}$ | Rel. Avg. |
|---|---|---|---|---|---|---|---|---|---|---|---|
| *Upper Bound: 100% TFLOPs (= 2880 Visual Tokens Every Layer)* | | | | | | | | | | | |
| LLaVA-NeXT-7B | 81.3 | 62.5 | 55.2 | 67.5 | 70.2 | 60.3 | 86.8 | 1511.8 | 65.8 | 57.3 | 100.0% |
| *Retain 14% TFLOPs (= 320 Visual Tokens Every Layer)* | | | | | | | | | | | |
| FastV Chen et al. (2024) | 61.5 | 49.8 | 51.3 | 66.6 | 58.6 | 52.2 | 49.5 | 1099.0 | 53.4 | 42.5 | 80.2% |
| DivPrune Alvar et al. (2025) | 77.2 | 61.1 | 55.6 | 67.7 | 67.1 | 56.2 | 84.7 | 1423.3 | 63.9 | 55.7 | 96.9% |
| VisPruner | 75.7 | 58.4 | **57.0** | 69.5 | 62.8 | 57.6 | 80.4 | 1370.1 | 63.6 | 55.8 | 95.5% |
| **DOP$_V$ (Ours)** | 77.6 | 60.4 | 56.4 | **69.7** | 65.2 | **59.2** | 84.6 | 1455.7 | 65.1 | **57.5** | 98.2% |
| CDPruner Zhang et al. (2025) | 78.4 | 61.6 | 55.8 | 67.8 | 67.3 | 57.4 | 87.2 | 1453.0 | 65.5 | 55.7 | 98.1% |
| **DOP$_{CD}$ (Ours)** | **79.1** | **61.7** | 55.4 | 68.0 | **68.4** | 57.9 | **87.4** | **1472.9** | **66.8** | 56.9 | **99.0%** |
| *Retain 9% TFLOPs (= 160 Visual Tokens Every Layer)* | | | | | | | | | | | |
| DivPrune Alvar et al. (2025) | 75.0 | 59.3 | 56.1 | 67.1 | 64.9 | 54.1 | 80.0 | 1356.6 | 62.9 | 53.7 | 94.2% |
| VisPruner | 70.6 | 54.7 | **57.1** | 68.6 | 58.1 | 56.0 | 72.7 | 1226.0 | 60.5 | 51.4 | 90.2% |
| **DOP$_V$ (Ours)** | 74.1 | 57.6 | 56.5 | **68.7** | 61.2 | **57.1** | 79.4 | 1355.6 | 62.1 | **55.1** | 94.1% |
| CDPruner Zhang et al. (2025) | 76.7 | 60.8 | 55.2 | 67.5 | 65.9 | 55.4 | 86.8 | 1425.3 | 64.2 | 53.8 | 96.3% |
| **DOP$_{CD}$ (Ours)** | **77.5** | **61.2** | 55.1 | 67.5 | **66.7** | 56.7 | **87.5** | **1446.4** | **64.8** | 55.1 | **97.2%** |

Table 2: **Performance comparison on LLaVA-NeXT-7B (Liu et al., 2024a).** Rel. Avg. is the mean performance ratio relative to the original model averaged across all benchmarks. **Bold** indicates the best at each TFLOPs ratio. DOP$_V$ uses VisPruner to reduce visual tokens, while DOP$_{CD}$ uses CDPruner. Our comparison matches TFLOPs and notes the per-layer token counts needed to achieve these budgets. Full comparison across LLaVA-Next-7B&13B are in Tables 10 and 11.

| Model | TFLOPs | Method | Prefilling (ms) | | | Sample (ms) | | | Total Time (min) | | | Cost |
|---|---|---|---|---|---|---|---|---|---|---|---|---|
| | | | EA | SA | FA | EA | SA | FA | EA | SA | FA | (min) |
| LLaVA-1.5-7B | 100% | - | 73 | 59 | 60 | 133 | 118 | 122 | 27 | 25 | 26 | - |
| | 28% | CDPruner | 28 | 26 | 26 | 93 | 86 | 85 | 13 | 12 | 12 | - |
| | | DOP$_{CD}$ | 28 | 26 | 26 | 93 | 86 | 87 | 13 | 12 | 13 | 5 |
| LLaVA-Next-7B | 100% | - | 494 | 253 | 254 | 593 | 346 | 350 | 64 | 43 | 44 | - |
| | 14% | CDPruner | 52 | 45 | 45 | 141 | 127 | 127 | 17 | 16 | 16 | - |
| | | DOP$_{CD}$ | 53 | 44 | 44 | 143 | 128 | 127 | 17 | 16 | 16 | 12 |

Table 3: **Efficiency and Cost on NVIDIA A100-80G GPU.** We report the average prefilling and sample latencies as well as total response time over 5000 TextVQA samples for LLaVA-1.5-7B and LLaVA-Next-7B using three attention implementations: Eager (EA), SDPA (SA), and FlashAttention2 (Dao et al., 2022) (FA). We also report the one-time cost for evaluating sampled individual divergences on 100 TextVQA samples with eager attention, where other costs for optimizing DOP are negligible. We only report DOP$_{CD}$ as DOP$_V$ has identical efficiency and cost.

Pruner and CDPruner for visual token reduction respectively. DOP consistently outperforms all baselines, demonstrating superior performance-efficiency tradeoffs and flexibility across visual token reduction methods. TextVQA-optimized policies generalize well across tasks. Notably, while progressive pruning methods (FitPrune, PDrop, SparseVLM) performance rapidly degrades at lower TFLOPs ratios, DOP's improvements become more pronounced, validating the benefits of refining token allocation granularity from layer-level to module-level.

**Qwen2.5-VL and InternVL3.** Tables 12 and 13 evaluate DOP$_R$ with input image resizing on Qwen2.5-VL-7B (Bai et al., 2025) and InternVL3-8B (Zhu et al., 2025). We include text-oriented VQA benchmarks (TextVQA, ChartQA, AI2D, OCRBench) that are more sensitive to visual information. DOP consistently improves performance across all settings, with notable improvements up to 24% on text-oriented benchmarks, demonstrating that DOP preserves more critical visual details than uniform allocation. Additionally, while CDPruner and DivPrune struggle on these non-LLaVA models against basic FastV and resizing baselines—contrasting with their advantages on LLaVA—this raises concerns about overfitting to model-specific patterns. In contrast, DOP maintains consistent gains across architectures, indicating that fine-grained token allocation effectively improves performance-efficiency tradeoffs across different architectures.

**Efficiency and Cost on Real GPU.** Table 3 reports efficiency gains and optimization costs on a single NVIDIA A100-80G GPU for LLaVA-1.5-7B and LLaVA-Next-7B, with the costs for other models shown in Table 21. DOP's TFLOPs reduction translates well to real device efficiency, with particularly notable improvements in prefilling stage latency, which is our target. We tested three commonly used attention implementations and found DOP directly compatible with all. Compared

| Validation | Evaluation Set | | | | | | | | | | Rel. Avg. |
|---|---|---|---|---|---|---|---|---|---|---|---|
| | VQA$^{v2}$ | GQA | VizWiz | SQA$^I$ | Seed$^I$ | VQA$^{Text}$ | POPE | MME | MMB | MMB$^{CN}$ | |
| VQA$^{Text}$ | **76.2** | **59.6** | **53.4** | **68.3** | 63.6 | **55.8** | **87.6** | 1436.3 | 62.4 | 55.1 | **97.9%** |
| Seed$^I$ | 60.5 | 50.3 | 45.9 | 67.6 | **64.2** | 50.5 | 85.0 | **1442.4** | **62.7** | **55.3** | 91.8% |
| VQA$^{v2}$ | 76.0 | 59.5 | 53.3 | **68.3** | 63.5 | **55.8** | **87.6** | 1408.6 | 61.7 | 54.1 | 97.3% |

Table 4: **Cross-task Generalization Ablation** with optimizing DOP$_{CD}$ for LLaVA-1.5-7B at 28% TFLOPs with 100 samples from TextVQA, SeedBench, and VQAv2. TextVQA and VQAv2 policies generalize well across tasks, while SeedBench policies show poor generalization.

| Model | LLaVA-1.5 | LLaVA-Next | Qwen2.5-VL | InternVL3 |
|---|---|---|---|---|
| Size | 13B | 7B / 13B | 7B | 8B |
| TFLOPs | 28% | 14% / 14% | 26% | 27% |
| Direct | 96.9% | 99.0% / 98.2% | 97.7% | 95.0% |
| Transfer | 96.7% | 98.8% / 98.2% | 97.4% | 94.3% |

Table 5: **Cross-Model Generalization Ablation.** Direct denotes DOP optimized directly on the target model, while Transfer denotes DOP with divergences transferred from DOP on LLaVA-1.5-7B (detailed in Appendix C.5). We use DOP$_{CD}$ for LLaVA and DOP$_R$ for Qwen2.5-VL and InternVL3. Results are the relative average performance (**Rel. Avg.**) compared to each original model.

to CDPruner under the same computational budget, DOP achieves comparable acceleration. Our optimization is also highly efficient with costs under 18 minutes for all models (Table 14), further reducible to 2 minutes using 25 samples (Appendix F.1), confirming deployment feasibility.

# 5 ABLATION STUDY

In this section, we conduct ablation studies on cross-task and cross-model generalization capabilities. Additional ablations are in Appendix F.

**Cross-task Generalization.** Table 4 evaluates DOP's cross-task generalization by optimizing policies on VQAv2 and SeedBench with 100 samples each, and shows a clear ranking: TextVQA policy $\geq$ VQAv2 policy $\gg$ SeedBench policy. According to Appendix F.2, this stems from different redundancy distributions across task types. SeedBench exhibits significantly greater deep-layer redundancy due to its multiple-choice format requiring only shallow visual processing, leading to aggressive deep pruning that fails to generalize to tasks requiring deep operations. Conversely, policies from tasks requiring deep operations generalize well to tasks like SeedBench, validating our TextVQA optimization choice.

**Cross-model Generalization.** Table 5 shows the results of generalizing DOP$_{CD}$ on LLaVA-1.5-7B to other models by transferring the individual divergences according to Appendix C.5. Results show that transferred DOP achieves comparable performance to direct optimization on target models, with differences within 0.7%. This indicates similar redundancy distributions across MLLMs, enabling strong cross-model generalization and reducing deployment costs by optimizing once and transferring to all models. Additionally, transferring DOP$_{CD}$ divergences to DOP$_R$ for Qwen2.5-VL and InternVL3 demonstrates effective generalization across different token reduction methods.

# 6 CONCLUSION

In this work, we investigated computational redundancy in MLLMs and addressed the limitation of prior token reduction that overlooks structural redundancy by proposing DOP, an operation pruning method that enables adjusting different modules' token loads based on their redundancy levels. DOP ensures efficient optimization through depth-wise pruning constraints and additive approximation techniques. Our comprehensive evaluation demonstrates that DOP preserves up to 7% more performance than state-of-the-art token reduction methods across 6 MLLM models and 13 tasks, with TFLOPs reductions translating effectively to acceleration on real GPUs. Ablation studies further reveal DOP's optimization efficiency and strong generalization capabilities, confirming its practical value for deploying MLLMs in resource-constrained environments.

# 7 REPRODUCIBILITY STATEMENT

We will release our complete code implementation upon acceptance to ensure full reproducibility of our results. Detailed method configurations and implementation specifics are provided in Appendix D and all the optimized DOP policies are in Tables 15 and 16. Our released codebase will include functionality for optimizing and evaluating DOP policies across all models and benchmarks utilized in this work, facilitating straightforward extension to additional models and benchmarks as well as seamless comparison with other methods.

# 8 ETHICS STATEMENT

DOP is designed to improve MLLM efficiency while preserving performance. This is a general technical improvement without any purpose or direct applications we can anticipate that would raise ethical considerations. We encourage responsible use of DOP and recommend that users avoid deploying it for applications that may raise ethical considerations.

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

## A  LLM USAGE

Large Language Models were used in this work solely as general-purpose assistance tools for minor editorial tasks, including grammar checking, basic text polishing to improve readability and academic writing style, and table formatting adjustments. LLMs did not play any significant role in research ideation, methodology development, experimental design, data analysis, or the primary writing of this manuscript. All technical contributions, scientific insights, experimental results, and core content originate entirely from the authors' original research work. The authors take full responsibility for all content presented in this paper.

## B  ADDITIONAL RELATED WORKS

### B.1  MULTIMODAL LARGE LANGUAGE MODEL (MLLM)

With the advancement of Large Language Models (LLMs) (Achiam et al., 2023; Touvron et al., 2023; Peng et al., 2023; Brown et al., 2020; Gunasekar et al., 2023; Abdin et al., 2024), Multimodal Large Language Models (MLLMs) (Liu et al., 2024b;a; Li et al., 2023b; Dai et al., 2023; Team et al., 2023; Bai et al., 2023a;b; Chen et al., 2023b; Ye et al., 2023; Du et al., 2021; Zhu et al., 2023b; Lin et al., 2023a; Liu et al., 2023a; Wang et al., 2023; Chen et al., 2023a; Zhao et al., 2024b; Abdin et al., 2024; Lin et al., 2023b) have successfully inherited the powerful reasoning capabilities of LLMs and extended them to multimodal tasks with visual inputs (Hudson & Manning, 2019; Li et al., 2023a; Chen et al., 2015; Fu et al., 2023; Liu et al., 2023b; Plummer et al., 2015; Agrawal et al., 2019; Lu et al., 2022). These models encode visual inputs as tokens to fully leverage LLM

capabilities, mapping visual features extracted from visual encoders (Radford et al., 2021; Zhu et al., 2023a; Zhao et al., 2024a) into text embedding spaces through modality alignment and instruction fine-tuning.

However, these MLLMs feature excessive visual tokens, with most computation used for processing these tokens, predominantly occurring in the LLM decoder part as its parameter size usually far exceeds other components. For instance, LLaVA-1.5 (Liu et al., 2024b) converts a 336×336 image into 576 visual tokens, while LLaVA-NeXT (Liu et al., 2024a) generates 2,880 tokens from twice the resolution. This trend of increasing token usage has led to significantly higher computational costs, highlighting the growing need for efficient inference and eliminating redundant computations.

## B.2 Neural Architecture Search (NAS)

While sharing some similarity with NAS (Zoph & Le, 2016; Liu et al., 2018; 2021; Wu et al., 2021; Sukthanker et al., 2024; Sarah et al., 2025; Teterwak et al., 2024), DOP is the extension of token reduction, not NAS. The key distinction is that NAS modifies network structure, while token reduction methods and our DOP both operate on input tokens, with DOP working at more fine-grained network components. DOP can be viewed as fine-grained token pruning on local components. While DOP considers network structures for computational subdivision, these structures remain fixed. Similarly, prior progressive pruning methods (Ye et al., 2024; Xing et al., 2024; Zhang et al., 2024c) also leverage network structures (layer positions) for finer granularity. DOP's optimization-based policy search is not exclusive to NAS and has been adopted in prior token pruning methods (Ye et al., 2024).

## B.3 Dynamic Computation Methods

While dynamic computation methods (Wu et al., 2024; Li et al., 2021; Zhao et al., 2025) and DOP both comprehensively consider layer/module redundancy and token redundancy, they diverge fundamentally in their approach to token redundancy: existing methods focus on redundancy differences between individual tokens, requiring model modifications (e.g., routers, token selectors) and substantial training data, whereas DOP recognizes modality-level redundancy differences in MLLMs, enabling modality-wide dynamic computation without architectural changes or extensive fine-tuning. Most dynamic computation methods require additional router modules and substantial data and computational resources for retraining - for example, Wu et al. (2024) requires 333k samples for three-stage training, Li et al. (2021) needs additional Token-Level Off-Ramps with extensive fine-tuning data, and Zhao et al. (2025) introduces routers requiring 850,000 samples for training. In contrast, DOP achieves strong generalization with minimal data (as few as 25 samples) and enables plug-and-play deployment across different models, thus achieving our objective: MLLM acceleration with minimal resources.

## B.4 Comparison with Layer Pruning Methods

Layer pruning (Ma et al., 2025; Gromov et al.) represents another direction for accelerating LLMs and MLLMs by leveraging redundancy differences across different layers and modules. Methods like Short-LVLM (Ma et al., 2025) also consider token and structural redundancy jointly, validating our motivation. However, the fundamental difference is that layer pruning methods perform structural modifications that alter model architecture, while DOP preserves the original model structure by only changing computation through operation pruning for specific tokens within modules. This distinction makes DOP essentially a token manipulation approach similar to token pruning methods, but operating at a finer operation level rather than requiring architectural changes.

# C Method Details

## C.1 Sparse Divergence Evaluation and Interpolation

We exploit the monotonically decreasing and smooth properties of $\mathcal{D}_A$, $\mathcal{D}_P$, and $\mathcal{D}_v$ with respect to increasing $d_A$, $d_P$, and $n_v$. Instead of exhaustively evaluating all possible values for each of $d_A$, $d_P$, and $n_v$, we sample $I$ key values per parameter, evaluate their individual divergences, and estimate

remaining values via linear interpolation. For $\mathcal{D}_A$, the interpolated divergence is:

$$\hat{\mathcal{D}}_A(d_A) = \mathcal{D}_A(d_{A,i}) + \frac{d_A - d_{A,i}}{d_{A,i+1} - d_{A,i}} \cdot (\mathcal{D}_A(d_{A,i+1}) - \mathcal{D}_A(d_{A,i})) \tag{8}$$

where $d_{A,i}$ and $d_{A,i+1}$ are the nearest sampled values with $d_{A,i} \leq d_A \leq d_{A,i+1}$. The final number of divergence evaluations is reduced to $3 \cdot I$.

## C.2 OUTPUT DIVERGENCE

We compute KL divergence between original and pruned model outputs to measure pruning impact. Specifically, we focus on the first output token generated during the prefilling stage. Given the original model's output probability distribution $\mathbf{Pr}_{org} \in \mathbb{R}^V$ over vocabulary size $V$, we identify the top-$k$ tokens and extract their probabilities:

$$\mathbf{X}_{top} = \arg\text{top-}k(\mathbf{Pr}_{org}), \quad \mathbf{p}_{org} = [\mathbf{Pr}_{org}[i]]_{i \in \mathbf{X}_{top}} \in \mathbb{R}^k \tag{9}$$

For a pruned model with policy $\mathbf{p}(\mathcal{P}) = [\mathbf{Pr}(\mathcal{P})[i]]_{i \in \mathbf{X}_{top}}$. We normalize both distributions for numerical stability as $\tilde{\mathbf{p}}_{org} = (\mathbf{p}_{org} + \epsilon)/\|\mathbf{p}_{org} + \epsilon\|_1$ and $\tilde{\mathbf{p}}(\mathcal{P}) = (\mathbf{p}(\mathcal{P}) + \epsilon)/\|\mathbf{p}(\mathcal{P}) + \epsilon\|_1$. We set $\epsilon = 10^{-8}$ and $k = 10$ in our implementation. The divergence is computed as:

$$\mathcal{D}(\mathcal{P}) = \mathcal{D}_{KL}(\tilde{\mathbf{p}}_{org} \| \tilde{\mathbf{p}}(\mathcal{P})) = \sum_{i=1}^{k} \tilde{\mathbf{p}}_{org}[i] \log \frac{\tilde{\mathbf{p}}_{org}[i]}{\tilde{\mathbf{p}}(\mathcal{P})[i]}. \tag{10}$$

## C.3 THEORETICAL FLOPs

We calculate the theoretical FLOPs (TFLOPs) for MLLM decoder layers following the LLaMA architecture. For one layer with hidden size $h$, key/value feature dimension $d$, and MLP intermediate dimension $m$, the aTFLOPs can be expressed as:

$$\text{TFLOPs} = \underbrace{8n_{\text{A}}hd + 4n_{\text{A}}^2 d}_{\text{Multi-Head Attention}} + \underbrace{6n_{\text{P}}hm}_{\text{LLaMA-style MLP}} \tag{11}$$

where $n_{\text{A}}$ and $n_{\text{P}}$ denote the number of remaining tokens in MHA and MLP modules respectively. For architectures with different designs (e.g., Qwen), we adjust the calculations accordingly in our implementation. Note that $n_A$ and $n_P$ include not only visual tokens but also system, text, and output tokens. Some works may incorrectly consider only visual token counts when computing TFLOPs, leading to smaller TFLOPs calculations.

## C.4 MINIMAL VISUAL TOKEN COUNT CONSTRAINT

Our ablation study in Section 5 reveals that different tasks have varying sensitivity to operation pruning. To ensure cross-task generalization, we perform policy optimization on the most sensitive OCR task. However, this occasionally creates another issue: without visual token count constraints, the optimized policy tends to be overly conservative with operations, primarily reducing tokens rather than operations. This prevents the policy from fully utilizing its potential on tasks less sensitive to operation pruning, limiting overall performance.

To ensure better performance across all tasks, we add the minimal visual token count constraint $n_v \geq n_{\min}$. For determining $n_{\min}$, we make it adaptive to the given computation budget. Given budget $\tau$, if reducing visual tokens alone to $n_e$ exactly meets the computation budget, i.e., $\text{TFLOPs}(L, L, n_e) = \tau$ (if no exact $n_e$ satisfies, we find the $n_e$ that makes TFLOPs closest to $\tau$), we set $n_{\min} = \alpha \cdot n_e$. In this paper, we set $\alpha = 1.5$ for LLaVA-series models (Liu et al., 2024b;a) and $\alpha = 1.25$ for other models (Bai et al., 2025; Zhu et al., 2025).

## C.5 CROSS-MODEL DOP GENERALIZATION

As analyzed in Section 3.2.2, the most computationally expensive component of the optimization in Equation (7) is the validation runs required to evaluate individual divergences $\hat{\mathcal{D}}_A(d_A)$, $\hat{\mathcal{D}}_P(d_P)$, and $\hat{\mathcal{D}}_v(n_v)$. In contrast, all other computations are negligible. These three divergences encapsulate the

core information for our optimization, namely the distribution of operation and token redundancy within each model. Therefore, when transferring DOP to other models, we only need to transfer these three individual divergences $\hat{\mathcal{D}}_A(d_A)$, $\hat{\mathcal{D}}_P(d_P)$, and $\hat{\mathcal{D}}_v(n_v)$.

Our transfer approach is based on rescaling the policy parameters $d_A$, $d_P$, and $n_v$ according to the architectural differences between models, followed by interpolation to obtain individual divergences corresponding to feasible values on the target model.

Specifically, if our source model originally has $N_v$ visual tokens and $L$ decoder layers, while our target model has $N_v'$ visual tokens and $L'$ decoder layers, we rescale a policy $(d_A, d_P, n_v)$ from the source model to obtain a corresponding policy $(d_A', d_P', n_v')$ on the target model:

$$d_A' = d_A \cdot \frac{L'}{L}, \quad d_P' = d_P \cdot \frac{L'}{L}, \quad n_v' = n_v \cdot \frac{N_v'}{N_v} \tag{12}$$

The corresponding individual divergences remain $\hat{\mathcal{D}}_A(d_A)$, $\hat{\mathcal{D}}_P(d_P)$, and $\hat{\mathcal{D}}_v(n_v)$. However, $(d_A', d_P', n_v')$ may not represent a feasible policy since policy parameters must be integers, while $d_A'$, $d_P'$, and $n_v'$ may not be integers. Therefore, based on the rescaled points and their corresponding individual divergences, we perform interpolation identical to that described in Appendix C.1 to obtain individual divergences for feasible policies on the target model. Finally, we substitute these transferred individual divergences into Equation (7) and apply the target model's parameters to compute TFLOPs and token count constraints, thereby obtaining DOP policies for the target model.

# D EXPERIMENTAL SETUP DETAILS

## D.1 MLLM MODEL DETAILS

**LLaVA-1.5 (Liu et al., 2024b)** is a prominent open-source multimodal large language model combining a CLIP visual encoder (Radford et al., 2021) with the Vicuna language model (Vicuna, 2023) through a linear projection layer. LLaVA-1.5 enhances the original architecture with a multi-layer perceptron projection, higher resolution inputs (336×336 pixels generating 576 visual tokens), and diverse instruction-following training data. Common variants include 7B and 13B parameter versions based on corresponding Vicuna backbones.

**LLaVA-NeXT (Liu et al., 2024a)** (also known as LLaVA-1.6) adopts an adaptive resolution strategy that dynamically adjusts input dimensions based on each image's native aspect ratio, scaling up to 4× the baseline resolution. High-resolution images are partitioned into sub-regions matching the original input size, processed independently, then concatenated for language model input. This enhances visual reasoning, OCR, and world knowledge capabilities. Following CDPruner (Zhang et al., 2025) for fair comparison, we fix input resolution at 672×672 pixels (4× increase), generating 2,880 visual tokens per image.

**Qwen2.5-VL (Bai et al., 2025)** is a state-of-the-art open-source multimodal model from the Qwen-VL family, offering substantial improvements in visual comprehension, document analysis, and video processing over Qwen2-VL (Wang et al., 2024). The architecture features an enhanced Vision Transformer with window attention, SwiGLU activation, and RMSNorm components consistent with the Qwen2.5 language model. A key innovation is adaptive resolution handling that controls visual token counts through input image resizing. Following CDPruner (Zhang et al., 2025), we fix baseline resolution at 1008×1008 pixels (1296 visual tokens), with both Resizing baselines and DOP$_R$ controlling token counts through image resizing.

**InternVL3 (Zhu et al., 2025)** is a state-of-the-art open-source multimodal model that maintains the ViT-MLP-LLM framework connecting a Vision Transformer to a language model via an MLP bridge. Key innovations include Variable Visual Position Encoding for handling extensive multimodal sequences and unified multimodal pre-training that simultaneously develops linguistic and visual-linguistic competencies. The model achieves exceptional performance across diverse applications including tool integration, GUI agents, and industrial image analysis. Following CDPruner (Zhang et al., 2025), we use 896×896 input resolution (1280 visual tokens) as the 100% TFLOPs baseline, with Resizing baselines and DOP$_R$ controlling token counts through resolution adjustment.

## D.2 BENCHMARK DETAILS

**VQAv2 (Goyal et al., 2017)** is an open-ended visual question answering benchmark that assesses models' integration of visual perception, linguistic comprehension, and commonsense reasoning. The dataset contains 265,016 images from COCO (Chen et al., 2015) and synthetic abstract scenes, with approximately 5.4 questions per image. Each question includes 10 verified correct answers and 3 plausible incorrect alternatives. We evaluate on the test-dev partition.

**GQA (Hudson & Manning, 2019)** is a comprehensive open-ended visual question answering dataset constructed from Visual Genome images (Krishna et al., 2017), designed to evaluate compositional reasoning and visual comprehension. The benchmark contains over 22 million balanced question-answer pairs, with each image annotated with detailed scene graphs specifying object categories, attributes, and relationships. We evaluate on the test-dev balanced partition.

**VizWiz (Gurari et al., 2018)** is a visual question answering dataset collected from real-world accessibility scenarios, where blind users captured images and posed spoken questions about their content. Each visual inquiry is paired with 10 crowd-sourced responses, with evaluation focusing on answering visual questions and determining when questions are unanswerable from the available visual information. The benchmark emphasizes real-world challenges including poor image quality and unclear visual content that complicate automated understanding. We evaluate on the test partition.

**ScienceQA (Saikh et al., 2022)** is a comprehensive multimodal multiple-choice dataset for evaluating scientific reasoning across natural science, language science, and social science domains. The benchmark contains 21,208 questions organized into 26 topics, 127 categories, and 379 skills, with 48.7% including images, 48.2% including text, and 30.8% combining both modalities. Most questions provide educational resources including grounded lectures (83.9%) and detailed explanations (90.5%). We evaluate on the test split containing image-based questions, referred to as SQA[I].

**SeedBench (Li et al., 2023a)** is a multiple-choice evaluation framework for MLLMs featuring 19,242 human-verified questions across 12 assessment dimensions for spatial (image) and temporal (video) understanding. The benchmark covers nine spatial dimensions including scene understanding, instance identity, attributes, location, counting, spatial relations, interactions, visual reasoning, and text recognition, plus three temporal dimensions for video analysis. We focus on the 14,233 image-based questions spanning the nine spatial dimensions, referred to as Seed[I].

**POPE (Li et al., 2023c)** is a benchmark designed to measure object hallucination in large vision-language models using COCO images (Chen et al., 2015). The dataset focuses on simple yes/no questions about whether specific objects appear in each image, allowing evaluation of how often models incorrectly claim to see absent objects. The benchmark uses precision, recall, and F1 score as evaluation metrics. We evaluate on the test split.

**MME (Fu et al., 2023)** is a comprehensive evaluation framework that assesses both perception and cognition capabilities of multimodal large language models through 14 subtasks. The perception component evaluates visual recognition from coarse-grained tasks (object presence, counting, positioning, color) to fine-grained recognition (posters, celebrities, scenes, landmarks, artworks) and OCR capabilities. The cognition component tests higher-level reasoning through commonsense reasoning, numerical calculations, text translation, and code reasoning.

**MMBench (Liu et al., 2023b)** is a comprehensive evaluation framework that assesses vision-language capabilities through an extensive and diverse collection of evaluation questions. A key innovation is the CircularEval strategy, which uses ChatGPT to transform open-ended model responses into structured multiple-choice formats for more consistent evaluation. The benchmark is available in English (MMB) and Chinese (MMB[CN]) versions.

**TextVQA (Singh et al., 2019)** is a benchmark that tests models' ability to read and understand text embedded within images. The dataset draws from Open Images v3 (Krasin et al., 2017) and features diverse real-world scenarios including signs, billboards, and product packaging containing textual information. The benchmark requires integration of optical character recognition (OCR) capabilities with visual reasoning skills for successful multimodal comprehension. We evaluate on the validation split and optimize DOP policies using samples from the training split.

**AI2D (Kembhavi et al., 2016)** is a diagram-based question answering benchmark focusing on scientific visual reasoning using grade school science materials. The dataset contains over 5,000 science diagrams with more than 150,000 structured annotations and over 15,000 multiple-choice questions, creating opportunities for research in visual reasoning and diagram comprehension within scientific domains. We evaluate on the test split with mask.

**ChartQA (Masry et al., 2022)** is a large-scale benchmark evaluating question answering over charts, requiring complex reasoning that combines visual interpretation with logical and arithmetic operations. The dataset includes 9.6K human-written questions and 23.1K questions generated from chart summaries. ChartQA requires models to perform multistep reasoning using both visual chart content and underlying data tables, demonstrating the need for sophisticated multimodal understanding. We evaluate on the test split.

**OCRBench (Liu et al., 2024c)** is a comprehensive evaluation framework that measures optical character recognition capabilities of large multimodal models across diverse text-related visual tasks. The benchmark encompasses 29 datasets covering text recognition, scene text-focused VQA, document-oriented VQA, key information extraction, and handwritten mathematical expression recognition.

### D.3 BASELINE DETAILS

**FastV (Chen et al., 2024)** is the first work to identify redundant visual attention in multimodal large language models, addressing this through a pruning strategy that removes visual tokens with the lowest visual-text attention scores after the second layer for training-free inference acceleration.

**PyramidDrop (Xing et al., 2024)** is a progressive pruning method extending FastV's insights by observing that visual token pruning in earlier layers significantly affects performance, while token redundancy becomes more pronounced at deeper layers. The method introduces multi-stage hierarchical pruning that partitions the MLLM into phases, systematically removing a predetermined fraction of visual tokens at each stage boundary.

**FitPrune (Ye et al., 2024)** is a progressive pruning method advancing FastV by investigating optimal timing and methodology for visual token pruning. The approach employs binary search to maintain visual-text attention distribution as closely as possible to the original under computational budgets, deriving an optimal pruning configuration. Visual tokens are then removed according to this strategy to maximize retention of original model capabilities.

**SparseVLM (Zhang et al., 2024c)** is a progressive token pruning approach that distinguishes itself by examining instruction tokens' role in vision-language attention mechanisms. The method argues that textual tokens contribute unequally to visual token pruning decisions, with only those highly correlated with visual content being significant. SparseVLM first identifies text tokens most relevant to visual input as evaluators, then leverages their attention patterns toward visual tokens to direct pruning, achieving enhanced performance.

**LLaVA-Prumerge (Shang et al., 2024)** pioneers token pruning based exclusively on visual information and token merging. The method identifies significant visual tokens using attention scores from the visual encoder, then merges each remaining token with its most semantically similar token through clustering techniques. LLaVA-Prumerge+ extends this framework by incorporating spatially uniform sampling for additional performance enhancements.

**VisionZip (Yang et al., 2024)** shares similarities with LLaVA-Prumerge in utilizing visual information for token pruning decisions. The method observes that visual encoder attention exhibits high concentration patterns and leverages this to extract dominant tokens based on visual attention weights. VisionZip then applies clustering to identify contextual tokens from the remaining pool, combining both groups before feeding them into the language model to maximize visual information preservation.

**VisPruner (Zhang et al., 2024a)** uses visual information for token pruning by identifying limitations in existing text-visual attention-based approaches and implementing a dual-phase strategy combining visual attention-based selection of significant tokens with similarity-based duplicate removal to maximize retention of diverse visual information.

**TRIM (Song et al., 2025)** addresses the limitation that pruning based solely on visual information while disregarding user instructions may result in suboptimal performance. The method incorporates CLIP metrics by calculating cosine similarity between image tokens from the visual encoder and text tokens from the text encoder, using these similarity measures to assess visual token significance. Visual tokens with lower similarity scores are eliminated to achieve inference acceleration.

**DART (Wen et al., 2025)** proposes that token duplication presents a more critical concern than token importance in pruning strategies. The method begins by identifying a small collection of pivot tokens, then iteratively preserves the most diverse tokens by selecting those with minimal similarity to previously chosen tokens. This process yields maximally diverse visual tokens for downstream processing.

**DivPrune (Alvar et al., 2025)** also emphasizes token diversity but reformulates the token pruning challenge as a Max-Min Diversity Problem (MMDP). The method retains the most diverse token subset by maximizing the minimum pairwise distance among selected tokens, providing a mathematically principled approach to diversity optimization.

**CDPruner (Zhang et al., 2025)** addresses limitations in existing methods by introducing conditional diversity maximization for visual token pruning. Unlike attention-based approaches that retain duplicate tokens or similarity-based methods that ignore instruction relevance, CDPruner defines conditional similarity between visual tokens given instruction context and reformulates pruning as a determinantal point process (DPP) optimization. This training-free approach achieves current state-of-the-art performance across multiple models and benchmarks.

## D.4 Configuration Details

### D.4.1 Baseline Configurations on LLaVA

For experiments on the LLaVA model, we evaluated our approach against several state-of-the-art methods including FastV, PyramidDrop, FitPrune, SparseVLM, LLaVA-Prumerge, VisionZip, VisPruner, TRIM, DART, DivPrune, and CDPruner. To ensure fair comparison and minimize implementation-induced variations, we adopted the performance metrics reported in Zhang et al. (2024a; 2025) for methods evaluated at comparable TFLOPs ratios across standard benchmarks: VQAv2, GQA, VizWiz, ScienceQA, POPE, MME, MMBench (both English and Chinese variants), TextVQA, AI2D, ChartQA, and OCRBench.

For FitPrune, we conducted independent evaluation using its official implementation on LLaVA-1.5-7B, optimizing on the same validation set and assessing performance across all benchmarks to maintain experimental consistency. Since Zhang et al. (2024a; 2025) did not report baseline performance on SeedBench, we implemented and evaluated all baselines on this benchmark using their official implementations.

### D.4.2 Baseline Configurations on Qwen2.5-VL and InternVL3

We followed the evaluation protocol established in Zhang et al. (2025) using VLMEvalKit. However, since Zhang et al. (2025) has not released their official implementations and several critical implementation details remain unclear, which could lead to significant reproduction discrepancies, we opted to directly adopt the reported results for FastV, DivPrune, and CDPruner from Zhang et al. (2025).

We conducted our own evaluations of Resizing and $DOP_R$ based on VLMEvalKit, ensuring alignment with the documented specifications. Specifically, for Qwen2.5-VL-7B, we established the baseline configuration by fixing the input pixel count at 1,016,064, yielding 1,296 visual tokens as the 100% TFLOPs reference. For different resizing configurations, we set the input pixel count to #visual tokens $\times$ 784. Correspondingly, for 512, 256, and 128 visual tokens, we configured the input pixel counts to 401,408, 200,704, and 100,352, respectively. For InternVL3-8B, we adopted the same resizing methodology.

### D.4.3 DOP Configurations

DOP exclusively prunes image token operations and performs optimization once on 100 randomly selected TextVQA training samples with random seed 42. In most experiments, we conducted

| Model | L | $N_v$ | $d_A$ & $d_P$ (sampled) | $n_v$ (sampled) |
|---|---|---|---|---|
| LLaVA-1.5-7B | 32 | 576 | {26, 20, 14, 8, 4, 0} | {512, 256, 128, 64, 32, 16} |
| LLaVA-1.5-13B | 40 | 576 | {32, 24, 16, 8, 4, 0} | {512, 256, 128, 64, 32, 16} |
| LLaVA-Next-7B | 32 | 2880 | {26, 20, 14, 8, 4, 0} | {2560, 1280, 640, 320, 160, 80} |
| LLaVA-Next-13B | 40 | 2880 | {32, 24, 16, 8, 4, 0} | {2560, 1280, 640, 320, 160, 80} |
| Qwen2.5-VL-7B | 28 | 1296 | {23, 18, 13, 8, 4, 0} | {512, 256, 128, 64, 32, 16} |
| InternVL3-8B | 28 | 1008 | {23, 18, 13, 8, 4, 0} | {512, 256, 128, 64, 32, 16} |

Table 6: Sampling points for evaluating individual divergence. $L$: number of decoder layers; $N_v$: visual token count at 100% TFLOPs; $d_A$ and $d_P$: we sample identical sets for MHA depth $d_A$ and MHA depth $d_P$; $n_v$ is visual token counts.

| Model | Spearman's $\rho$ | $p$-value |
|---|---|---|
| LLaVA-1.5-7B | 0.799 | $4.19 \times 10^{-49}$ |
| LLaVA-Next-7B | 0.862 | $3.94 \times 10^{-65}$ |

Table 7: Spearman's rank correlation between joint divergence $\mathcal{D}(d_A, d_P, n_v)$ and additive approximation $\mathcal{D}_{\mathrm{sum}}(d_A, d_P, n_v)$ across 216 sampled combinations.

"model-specific" optimization, where each model undergoes optimization once, and the resulting policy is applied consistently across all benchmarks at the same TFLOPs ratio.

For the divergence terms $\hat{\mathcal{D}}_A(d_A)$, $\hat{\mathcal{D}}_P(d_P)$, and $\hat{\mathcal{D}}_v(n_v)$ in Equation (7), we sample $K = 6$ points each for visual token count, MHA depth, and MLP depth to evaluate output divergence. These sampling points are detailed in Table 6. We set $n_{min}$ according to Appendix C.4.

For LLaVA-series models, we evaluate DOP$_V$ and DOP$_{CD}$, corresponding to VisPruner and CD-Pruner respectively as token reduction methods. For Qwen2.5-VL and InternVL3, we evaluate DOP$_R$ with input image resizing as the token reduction approach.

### D.4.4 IMPLEMENTATION DETAILS

For LLaVA model experiments, we implemented DOP$_V$ using VisPruner's official codebase[1] and developed DOP$_{CD}$ based on CDPruner's official implementation[2] to eliminate potential variations caused by implementation differences. For Qwen2.5-VL and InternVL3 evaluations, we adopted the same approach as CDPruner by utilizing VLMEvalKit[3] for evaluation.

## E ADDITIONAL EXPERIMENTS

### E.1 DIVERGENCE RANKING CORRELATION ANALYSIS FOR ADDITIVE APPROXIMATION

Section 3.2.2 uses an additive approximation to reduce optimization complexity by replacing the joint divergence $\mathcal{D}(d_A, d_P, n_v)$ in Equation (5) with the sum of three individual divergences:

$$\mathcal{D}_{\mathrm{sum}}(d_A, d_P, n_v) = \mathcal{D}_A(d_A) + \mathcal{D}_P(d_P) + \mathcal{D}_v(n_v) = \mathcal{D}(d_A, L, N_v) + \mathcal{D}(L, d_P, N_v) + \mathcal{D}(L, L, n_v)$$

The key to ensuring sufficiently accurate optimization is that $\mathcal{D}_{\mathrm{sum}}(d_A, d_P, n_v)$ preserves the sufficiently accurate ranking order of $\mathcal{D}(d_A, d_P, n_v)$, allowing Equation (6) to largely hold.

We validate this by computing that rankings based on $\mathcal{D}$ and $\mathcal{D}_{\mathrm{sum}}$ are highly correlated using Spearman's rank correlation coefficient. Specifically, we sample combinations of $(d_A, d_P, n_v)$ and evaluate both $\mathcal{D}(d_A, d_P, n_v)$ and $\mathcal{D}_{\mathrm{sum}}(d_A, d_P, n_v)$ on real MLLM models with validation tasks, then rank the sampled combinations according to $\mathcal{D}(d_A, d_P, n_v)$ and $\mathcal{D}_{\mathrm{sum}}(d_A, d_P, n_v)$ respectively and compute the Spearman's rank correlation coefficient between these two rankings.

---

[1] https://github.com/Theia-4869/VisPruner

[2] https://github.com/Theia-4869/CDPruner

[3] https://github.com/open-compass/VLMEvalKit

We conduct this experiment on LLaVA-1.5-7B and LLaVA-Next-7B with evaluating divergence on the same 100 TextVQA samples as other experiments. For $d_A$ and $d_P$, we sample $I = 6$ points each: $\mathcal{S}_d = \{26, 20, 14, 8, 4, 0\}$ for both models. For $n_v$, we use $\mathcal{S}_v^{1.5} = \{512, 256, 128, 64, 32, 16\}$ for LLaVA-1.5-7B and $\mathcal{S}_v^{\text{Next}} = \{2560, 1280, 640, 320, 160, 80\}$ for LLaVA-Next-7B.

We evaluate $\mathcal{D}(d_A, d_P, n_v)$ for all combinations in $\mathcal{C} = \mathcal{S}_d \times \mathcal{S}_d \times \mathcal{S}_v$, yielding $|\mathcal{C}| = 6^3 = 216$ points. For individual divergences, we evaluate $\mathcal{D}_A(d_A)$ for $d_A \in \mathcal{S}_d$, $\mathcal{D}_P(d_P)$ for $d_P \in \mathcal{S}_d$, and $\mathcal{D}_v(n_v)$ for $n_v \in \mathcal{S}_v$, then sum these to obtain $\mathcal{D}_{\text{sum}}(d_A, d_P, n_v)$ for all combinations in $\mathcal{C}$. Let $\mathcal{R}_{\mathcal{D}}$ and $\mathcal{R}_{\text{sum}}$ denote the rankings of the combinations based on $\mathcal{D}$ and $\mathcal{D}_{\text{sum}}$ respectively. The Spearman's rank correlation coefficient without tied ranks is computed as:

$$\rho = 1 - \frac{6 \sum_{i=1}^{|\mathcal{C}|} (\mathcal{R}_{\mathcal{D}}(i) - \mathcal{R}_{\text{sum}}(i))^2}{|\mathcal{C}|(|\mathcal{C}|^2 - 1)}$$

where $|\mathcal{C}| = 216$. When tied ranks exist, the calculation becomes more complex, and we use the implemented function in scipy[4] which automatically handles such cases.

Table 7 shows strong and highly significant monotonic relationships between the additive proxy and joint divergence on both models ($\rho \geq 0.799$, $p < 10^{-48}$). The additive approximation $\mathcal{D}_{\text{sum}}$ closely preserves the ranking induced by joint divergence $\mathcal{D}$, validating our substitution of $\mathcal{D}$ with $\mathcal{D}_{\text{sum}}$ for reduced optimization complexity. While rankings between a small number of combinations may occasionally swap, the approximation is sufficiently accurate across different models.

### E.2 FULL COMPARISON ON LLAVA

Tables 8 to 11 present comprehensive evaluations of DOP on LLaVA-1.5-7B&13B and LLaVA-Next-7B&13B across three different TFLOPs ratios, along with comparisons against 11 baseline methods. The results demonstrate that at relatively high TFLOPs ratios, progressive pruning methods such as FitPrune, PDrop, and SparseVLM indeed exhibit certain advantages. However, at low TFLOPs ratios, their performance deteriorates rapidly, whereas our DOP consistently improves performance across all TFLOPs ratios.

### E.3 FULL COMPARISON ON QWEN2.5-VL AND INTERNVL3

Table 12 presents full evaluation of DOP across three different TFLOPs ratios on Qwen2.5-VL-7B, while Table 13 shows complete evaluation results across two different TFLOPs ratios on InternVL3-8B. The results reveal that across different TFLOPs ratios, CDPruner and DivPrune struggle to outperform basic FastV and Resizing baselines, which stands in stark contrast to their significant advantages on LLaVA models, raising concerns about overfitting to model-specific patterns. In contrast, DOP maintains consistent gains across architectures, indicating that different architectures share similar structural redundancy patterns and enabling strong cross-model generalization.

### E.4 FULL OPTIMIZATION COST

In Table 14, we present the optimization cost of DOP on LLaVA-1.5-7B&13B, LLaVA-Next-7B&13B, Qwen2.5-VL-7B, and InternVL3-8B using the same 100 TextVQA samples as the validation set. Since other computational costs are negligible compared to the validation runs for divergence evaluation, we primarily report the time required to complete divergence evaluation across all sampled points. Note that as mentioned in Appendix C.2, we only need to generate the first output token for each sample to compute the divergence, eliminating the need to generate complete question responses. The results demonstrate that our DOP is highly efficient, with optimization costs not exceeding 18 minutes on a single NVIDIA A100-80G GPU across all models. According to Appendix F.1, by using 25 samples for optimization, we can further reduce the optimization cost to as low as 2 minutes with comparable performance, though we default to using 100 TextVQA samples in our main experiments since this configuration is already sufficiently efficient.

| Method | VQA$^{v2}$ | GQA | VizWiz | SQA$^I$ | Seed$^I$ | VQA$^{Text}$ | POPE | MME | MMB | MMB$^{CN}$ | Rel. Avg. |
|---|---|---|---|---|---|---|---|---|---|---|---|
| *Upper Bound: 100% TFLOPs (= 576 Visual Tokens Every Layer)* | | | | | | | | | | | |
| LLaVA-1.5-7B | 78.5 | 61.9 | 50.1 | 69.5 | 66.2 | 58.2 | 85.9 | 1506.5 | 64.7 | 58.1 | 100% |
| *Retain 37% TFLOPs (= 128 Visual Tokens Every Layer)* | | | | | | | | | | | |
| FastV (Chen et al., 2024) | 71.0 | 54.0 | 51.9 | 69.2 | 55.9 | 56.4 | 68.2 | 1368.9 | 63.0 | 55.9 | 92.6% |
| PDrop (Xing et al., 2024) | 74.3 | 57.1 | 49.4 | **70.1** | 54.8 | 56.7 | 77.5 | 1444.1 | 62.3 | 55.3 | 94.4% |
| FitPrune (Ye et al., 2024) | 72.9 | 58.5 | 51.7 | 68.0 | 62.2 | 55.7 | 77.9 | 1445.3 | 62.7 | 56.2 | 95.8% |
| SparseVLM (Zhang et al., 2024c) | 75.1 | 57.3 | 49.7 | 69.0 | 58.2 | 56.3 | 83.1 | 1399.3 | 62.6 | 56.9 | 95.6% |
| PruMerge+ (Shang et al., 2024) | 75.0 | 58.2 | **53.7** | 69.1 | 62.7 | 54.0 | 83.1 | 1408.1 | 61.8 | 55.8 | 96.5% |
| TRIM (Song et al., 2025) | 75.4 | 58.4 | 51.6 | 68.6 | 62.5 | 52.2 | 85.3 | 1413.4 | 63.0 | 52.3 | 95.7% |
| VisionZip (Yang et al., 2024) | 75.6 | 57.6 | 51.6 | 68.7 | 61.8 | 56.9 | 83.3 | 1436.9 | 62.1 | 57.0 | 96.9% |
| DART (Wen et al., 2025) | 74.7 | 57.9 | 52.8 | 69.1 | 62.2 | 56.3 | 80.4 | 1408.7 | 60.7 | **57.3** | 96.4% |
| DivPrune (Alvar et al., 2025) | 76.0 | 59.4 | 52.8 | 68.6 | 63.6 | 55.9 | 87.0 | 1405.1 | 61.5 | 54.8 | 97.3% |
| VisPruner (Zhang et al., 2024a) | 75.8 | 58.2 | 52.7 | 69.1 | 62.5 | 57.0 | 84.6 | 1461.4 | 62.7 | **57.3** | 97.9% |
| **DOP$_V$ (Ours)** | 76.9 | 59.3 | 52.3 | 69.1 | 63.4 | **57.4** | 85.9 | 1447.8 | 63.1 | 57.2 | 98.4% |
| CDPruner (Zhang et al., 2025) | 76.6 | 59.9 | 52.8 | 69.0 | 64.0 | 56.2 | **87.7** | 1431.4 | 63.1 | 55.0 | 98.2% |
| **DOP$_{CD}$ (Ours)** | **77.2** | **60.5** | 52.8 | 68.7 | **64.8** | 57.1 | 87.6 | **1448.1** | 63.3 | 55.5 | **98.8%** |
| *Retain 28% TFLOPs (= 64 Visual Tokens Every Layer)* | | | | | | | | | | | |
| FastV (Chen et al., 2024) | 55.9 | 46.0 | 49.1 | **70.1** | 51.9 | 51.6 | 35.5 | 973.5 | 50.1 | 42.1 | 76.7% |
| PDrop (Xing et al., 2024) | 56.3 | 46.1 | 46.3 | 68.8 | 48.9 | 49.2 | 40.8 | 982.2 | 48.0 | 36.6 | 74.6% |
| FitPrune (Ye et al., 2024) | 63.7 | 52.3 | 51.1 | 68.0 | 55.8 | 51.2 | 73.6 | 1287.5 | 58.5 | 49.7 | 88.5% |
| SparseVLM (Zhang et al., 2024c) | 66.9 | 52.0 | 49.4 | 69.2 | 52.2 | 52.1 | 69.7 | 1190.4 | 58.3 | 49.6 | 87.1% |
| PruMerge+ (Shang et al., 2024) | 71.3 | 55.4 | **53.7** | 69.5 | 60.3 | 52.0 | 75.7 | 1316.8 | 59.6 | 52.1 | 92.5% |
| TRIM (Song et al., 2025) | 72.4 | 56.6 | 51.1 | 69.0 | 61.0 | 49.7 | 85.9 | 1350.9 | 60.9 | 48.2 | 92.9% |
| VisionZip (Yang et al., 2024) | 72.4 | 55.1 | 52.9 | 69.0 | 60.0 | 55.5 | 77.0 | 1365.2 | 60.1 | **55.4** | 94.1% |
| DART (Wen et al., 2025) | 71.3 | 54.7 | 53.5 | 69.3 | 59.6 | 54.7 | 73.8 | 1365.1 | 59.5 | 54.0 | 93.1% |
| DivPrune (Alvar et al., 2025) | 74.1 | 57.5 | 53.6 | 68.0 | 61.7 | 54.5 | 85.5 | 1334.7 | 60.1 | 52.3 | 95.0% |
| VisPruner (Zhang et al., 2024a) | 72.7 | 55.4 | 53.3 | 69.1 | 58.5 | 55.8 | 80.4 | 1369.9 | 61.3 | 55.1 | 94.6% |
| **DOP$_V$ (Ours)** | 74.6 | 57.1 | 53.6 | **69.2** | 60.3 | **56.5** | 83.1 | 1394.6 | 61.6 | **56.5** | 96.4% |
| CDPruner (Zhang et al., 2025) | 75.4 | 58.6 | 53.4 | 68.1 | 62.5 | 55.3 | 87.5 | 1415.1 | 61.1 | 53.2 | 96.7% |
| **DOP$_{CD}$ (Ours)** | **76.2** | **59.6** | 53.4 | 68.3 | **63.6** | 55.8 | **87.6** | **1436.3** | 62.4 | 55.1 | **97.9%** |
| *Retain 23% TFLOPs (= 32 Visual Tokens Every Layer)* | | | | | | | | | | | |
| PruMerge+ (Shang et al., 2024) | 65.6 | 52.9 | **53.5** | 69.5 | 55.4 | 49.2 | 66.7 | 1236.6 | 55.1 | 45.9 | 86.6% |
| TRIM (Song et al., 2025) | 68.6 | 54.5 | 50.7 | 68.1 | 58.6 | 47.6 | 84.9 | 1251.8 | 57.7 | 40.1 | 88.5% |
| VisionZip (Yang et al., 2024) | 67.1 | 51.8 | 52.4 | 69.1 | 56.2 | 53.1 | 69.4 | 1251.2 | 57.0 | 50.3 | 88.8% |
| DART (Wen et al., 2025) | 67.1 | 52.9 | 52.5 | 69.3 | 58.4 | 52.2 | 69.1 | 1273.3 | 58.5 | 50.0 | 89.5% |
| DivPrune (Alvar et al., 2025) | 71.2 | 54.9 | 53.3 | 68.6 | 58.7 | 52.9 | 81.5 | 1284.9 | 57.6 | 49.1 | 91.8% |
| VisPruner (Zhang et al., 2024a) | 67.7 | 52.2 | 53.0 | 69.2 | 54.3 | 53.9 | 72.7 | 1271.0 | 58.4 | 52.7 | 90.1% |
| **DOP$_V$ (Ours)** | 71.0 | 54.8 | 53.9 | 69.1 | 56.7 | **54.5** | 79.6 | 1306.5 | 59.4 | **53.7** | 92.9% |
| CDPruner (Zhang et al., 2025) | 73.6 | 57.0 | 53.1 | **69.5** | 60.9 | 53.2 | **87.9** | 1373.0 | 59.6 | 49.6 | 94.6% |
| **DOP$_{CD}$ (Ours)** | **74.7** | **58.1** | 53.6 | 69.3 | **62.2** | 54.2 | **87.9** | **1397.5** | 60.1 | 52.2 | **96.1%** |
| *Lower Bound: Retain 18.6% TFLOPs (= 0 Visual Token)* | | | | | | | | | | | |

Table 8: **Performance comparison on LLaVA-1.5-7B (Liu et al., 2024b) under different computation budgets.** We evaluate across 10 benchmarks: VQAv2, GQA (Hudson & Manning, 2019), VizWiz, SQA-Img, SeedBench (Li et al., 2023a), TextVQA, POPE (Li et al., 2023c), MME (Fu et al., 2023), MMBench (Liu et al., 2023b), and MMB-CN. We report accuracy (%) for most benchmarks. Avg. represents the mean performance ratio relative to Vanilla across all benchmarks (higher is better). **Bold** values indicate the best performances in each setting. DOP$_V$ uses VisPruner to reduce visual tokens, while DOP$_{CD}$ uses CDPruner. DOP consistently outperforms all baselines with policies only optimized on a subset of only 100 TextVQA training set samples. Lower Bound indicates the minimum possible computation in our settings. Our comparison matches actual TFLOPs and notes the per-layer token counts that achieve these TFLOPs.

| Method | VQA$^{V2}$ | GQA | VizWiz | SQA$^I$ | Seed$^I$ | VQA$^{Text}$ | POPE | MME | MMB | MMB$^{CN}$ | Rel. Avg. |
|---|---|---|---|---|---|---|---|---|---|---|---|
| *Upper Bound: 100% TFLOPs (= 576 Visual Tokens Every Layer)* | | | | | | | | | | | |
| LLaVA-1.5-13B | 80.0 | 63.3 | 53.6 | 72.8 | 68.3 | 61.2 | 86.0 | 1531.2 | 68.5 | 63.5 | 100% |
| *Retain 37% TFLOPs (= 128 Visual Tokens Every Layer)* | | | | | | | | | | | |
| FastV (Chen et al., 2024) | 75.3 | 58.3 | **54.6** | 74.2 | 63.2 | 58.6 | 75.5 | 1460.6 | 66.1 | 62.3 | 95.6% |
| PDrop (Xing et al., 2024) | 78.2 | 61.0 | 53.8 | 73.3 | 65.6 | 60.2 | 83.6 | 1489.5 | 67.5 | 62.8 | **98.2%** |
| SparseVLM (Zhang et al., 2024c) | 77.6 | 59.6 | 51.4 | **74.3** | 64.5 | **59.3** | 85.0 | 1487.9 | **68.4** | 62.6 | 97.5% |
| PruMerge+ (Shang et al., 2024) | 76.2 | 58.3 | 52.8 | 73.3 | 63.8 | 56.1 | 82.7 | 1445.9 | 66.3 | 61.2 | 95.5% |
| TRIM (Song et al., 2025) | 76.4 | 59.4 | 49.7 | 72.4 | 64.5 | 55.0 | 86.8 | 1426.9 | 67.1 | 58.4 | 95.0% |
| VisionZip (Yang et al., 2024) | 76.8 | 57.9 | 52.3 | 73.8 | 63.7 | 58.9 | 82.7 | 1449.2 | 67.4 | 62.5 | 96.4% |
| DART (Wen et al., 2025) | 75.7 | 57.7 | 53.0 | 74.2 | 62.9 | 58.7 | 80.4 | 1395.0 | 65.4 | 62.2 | 95.3% |
| DivPrune (Alvar et al., 2025) | 77.1 | 59.2 | 53.5 | 72.8 | 64.4 | 58.0 | 86.8 | 1457.7 | 66.3 | 60.7 | 96.7% |
| VisPruner (Zhang et al., 2024a) | 76.9 | 58.2 | 53.0 | **74.3** | 63.6 | 58.9 | 84.1 | 1449.7 | 67.3 | 62.5 | 96.8% |
| **DOP$_V$ (Ours)** | 78.1 | 59.3 | 53.4 | 73.6 | 65.1 | 59.1 | 85.4 | 1461.7 | 67.2 | **63.1** | 97.6% |
| CDPruner (Zhang et al., 2025) | 77.7 | 59.7 | 52.9 | 73.2 | 65.1 | 58.4 | **87.3** | 1478.0 | 67.5 | 61.5 | 97.5% |
| **DOP$_{CD}$ (Ours)** | **78.3** | 59.8 | 53.1 | 73.0 | **66.0** | 58.6 | 86.5 | **1490.7** | 67.5 | 62.2 | 97.9% |
| *Retain 28% TFLOPs (= 64 Visual Tokens Every Layer)* | | | | | | | | | | | |
| FastV (Chen et al., 2024) | 65.3 | 51.9 | 53.8 | 73.1 | 56.1 | 53.4 | 56.9 | 1246.4 | 59.2 | 55.1 | 85.5% |
| PDrop (Xing et al., 2024) | 70.8 | 54.1 | 50.5 | 73.1 | 58.4 | 55.3 | 66.1 | 1247.0 | 63.1 | 56.6 | 88.4% |
| SparseVLM (Zhang et al., 2024c) | 73.2 | 55.9 | 52.1 | 73.0 | 60.3 | 57.1 | 77.9 | 1374.3 | 65.2 | 60.3 | 92.9% |
| PruMerge+ (Shang et al., 2024) | 72.6 | 56.3 | 52.4 | 73.5 | 60.7 | 54.4 | 75.7 | 1338.2 | 65.0 | 59.3 | 92.0% |
| TRIM (Song et al., 2025) | 73.2 | 57.9 | 49.2 | 72.0 | 62.5 | 52.0 | 86.5 | 1406.2 | 65.0 | 52.7 | 92.0% |
| VisionZip (Yang et al., 2024) | 73.7 | 56.2 | 53.2 | **74.2** | 60.6 | 57.4 | 75.7 | 1379.6 | 64.9 | 61.3 | 93.4% |
| DART (Wen et al., 2025) | 72.4 | 55.7 | 53.4 | 73.8 | 60.1 | 57.3 | 72.8 | 1380.0 | 64.7 | 60.6 | 92.6% |
| DivPrune (Alvar et al., 2025) | 75.2 | 57.9 | **54.4** | 71.7 | 62.5 | 57.4 | 84.5 | 1454.2 | 64.1 | 59.8 | 95.2% |
| VisPruner (Zhang et al., 2024a) | 73.8 | 56.4 | 53.9 | 74.0 | 60.4 | 58.1 | 79.7 | 1376.2 | 64.4 | 60.0 | 93.8% |
| **DOP$_V$ (Ours)** | 75.7 | 57.3 | 53.7 | 73.4 | 62.6 | **58.5** | 81.1 | 1414.4 | 66.8 | **62.0** | 95.5% |
| CDPruner (Zhang et al., 2025) | 76.7 | 59.4 | 53.6 | 72.5 | 64.1 | 57.6 | **87.1** | 1466.8 | 65.5 | 58.8 | 96.3% |
| **DOP$_{CD}$ (Ours)** | **77.2** | **59.5** | 52.8 | 72.4 | **64.6** | 57.7 | 87.0 | **1484.8** | **66.9** | 60.8 | **96.9%** |
| *Retain 23% TFLOPs (= 32 Visual Tokens Every Layer)* | | | | | | | | | | | |
| PruMerge+ (Shang et al., 2024) | 66.8 | 54.1 | 52.3 | 71.7 | 57.8 | 52.4 | 67.4 | 1269.1 | 61.1 | 53.5 | 87.0% |
| TRIM (Song et al., 2025) | 69.8 | 55.6 | 48.8 | 70.4 | 59.3 | 49.6 | 85.8 | 1284.7 | 63.1 | 45.4 | 87.8% |
| VisionZip (Yang et al., 2024) | 68.4 | 52.7 | 53.0 | 72.9 | 56.2 | 55.2 | 66.8 | 1257.7 | 61.2 | 55.8 | 87.7% |
| DART (Wen et al., 2025) | 68.1 | 53.9 | 52.0 | 73.2 | 57.6 | 55.1 | 66.9 | 1282.8 | 61.9 | 56.2 | 88.3% |
| DivPrune (Alvar et al., 2025) | 72.0 | 56.2 | 54.5 | 70.9 | 60.1 | 54.6 | 79.3 | 1405.2 | 61.7 | 57.2 | 91.9% |
| VisPruner (Zhang et al., 2024a) | 69.0 | 52.6 | 52.8 | 71.7 | 56.5 | 56.0 | 71.9 | 1314.2 | 61.3 | 56.1 | 88.8% |
| **DOP$_V$ (Ours)** | 73.2 | 56.0 | 53.8 | **73.6** | 59.7 | **57.2** | 78.8 | 1365.6 | 64.3 | **59.5** | 93.1% |
| CDPruner (Zhang et al., 2025) | 75.2 | 58.5 | 53.5 | 71.9 | 62.5 | 55.3 | **87.6** | 1421.0 | 63.7 | 56.6 | 94.4% |
| **DOP$_{CD}$ (Ours)** | **76.2** | **59.0** | **54.7** | 68.9 | **63.8** | 56.7 | 87.5 | **1468.9** | **65.1** | 58.7 | **95.6%** |
| Lower Bound: *Retain 18.6% TFLOPs (= 0 Visual Token)* | | | | | | | | | | | |

Table 9: **Performance on LLaVA-1.5-13B under different computation budgets.** We evaluate across 10 benchmarks: We evaluate across 10 benchmarks: VQA$^{V2}$ (Goyal et al., 2017), VizWiz (Gurari et al., 2018), SQA$^I$ (Saikh et al., 2022), GQA (Hudson & Manning, 2019), Seed$^I$ (Li et al., 2023a), MME (Fu et al., 2023), MMB (Liu et al., 2023b), POPE (Li et al., 2023c), MMB$^{CN}$ and VQA$^{Text}$ (Singh et al., 2019). We report accuracy (%) for most benchmarks. Avg. represents the mean performance ratio relative to Vanilla across all benchmarks (higher is better). **Bold** values indicate the best performances in each setting. DOP$_V$ uses VisPruner to reduce visual tokens, while DOP$_{CD}$ uses CDPruner. DOP consistently outperforms all baselines with policies only optimized on a subset of only 100 TextVQA training set samples. Lower Bound indicates the minimum possible computation in our settings. Our comparison matches actual TFLOPs and notes the per-layer token counts that achieve these TFLOPs.

| Method | VQA$^{v2}$ | GQA | VizWiz | SQA$^I$ | Seed$^I$ | VQA$^{Text}$ | POPE | MME | MMB | MMB$^{CN}$ | Rel. Avg. |
|---|---|---|---|---|---|---|---|---|---|---|---|
| *Upper Bound: 100% TFLOPs (= 2880 Visual Tokens Every Layer)* | | | | | | | | | | | |
| LLaVA-NeXT-7B | 81.3 | 62.5 | 55.2 | 67.5 | 70.2 | 60.3 | 86.8 | 1511.8 | 65.8 | 57.3 | 100.0% |
| *Retain 23% TFLOPs (= 640 Visual Tokens Every Layer)* | | | | | | | | | | | |
| FastV (Chen et al., 2024) | 77.0 | 58.9 | 53.9 | 67.4 | 62.5 | 58.1 | 79.5 | 1412.6 | 63.1 | 53.5 | 94.6% |
| PDrop (Xing et al., 2024) | 79.1 | 60.0 | 53.8 | 66.7 | 67.4 | 57.8 | 83.8 | 1475.9 | 64.1 | 55.2 | 96.9% |
| SparseVLM (Zhang et al., 2024c) | 79.2 | 61.2 | 53.6 | 67.6 | 68.1 | 59.7 | 85.3 | 1456.8 | 65.9 | 58.6 | 98.6% |
| PruMerge+ (Shang et al., 2024) | 78.2 | 60.8 | **57.9** | 67.8 | 67.7 | 54.9 | 85.3 | 1480.2 | 64.6 | 57.3 | 98.1% |
| TRIM (Song et al., 2025) | 78.3 | 62.1 | 54.8 | 66.9 | 67.3 | 54.8 | 86.9 | 1471.8 | **66.8** | 55.8 | 97.7% |
| VisionZip (Yang et al., 2024) | 79.1 | 61.2 | 57.1 | 68.1 | 67.4 | **59.9** | 86.0 | **1493.4** | 65.8 | 58.1 | 99.4% |
| DART (Wen et al., 2025) | 78.3 | 61.3 | 57.0 | 68.2 | 67.9 | 59.5 | 85.0 | 1450.2 | 64.9 | 57.1 | 98.6% |
| DivPrune (Alvar et al., 2025) | 79.3 | 61.9 | 55.7 | 67.8 | 68.6 | 57.0 | 86.9 | 1469.7 | 65.8 | 57.3 | 98.8% |
| VisPruner | 79.8 | 61.6 | 57.1 | **69.0** | 66.2 | 59.3 | 85.9 | 1480.7 | 65.0 | 57.3 | 99.0% |
| **DOP$_V$ (Ours)** | 80.2 | 62.1 | 56.9 | 68.7 | 68.2 | 59.4 | 87.3 | 1484.1 | 66.3 | **58.7** | **100.0%** |
| CDPruner (Zhang et al., 2025) | 79.9 | **62.6** | 55.6 | 67.9 | 68.9 | 58.4 | 87.3 | 1474.5 | 66.3 | 57.5 | 99.4% |
| **DOP$_{CD}$ (Ours)** | **80.3** | 62.5 | 55.2 | 68.2 | **69.3** | 59.3 | **87.6** | 1477.3 | 66.6 | 58.3 | 99.8% |
| *Retain 14% TFLOPs (= 320 Visual Tokens Every Layer)* | | | | | | | | | | | |
| FastV (Chen et al., 2024) | 61.5 | 49.8 | 51.3 | 66.6 | 58.6 | 52.2 | 49.5 | 1099.0 | 53.4 | 42.5 | 80.2% |
| PDrop (Xing et al., 2024) | 66.8 | 50.4 | 49.7 | 66.7 | 59.7 | 49.0 | 60.8 | 1171.5 | 55.5 | 44.7 | 82.8% |
| SparseVLM (Zhang et al., 2024c) | 74.6 | 57.9 | 54.2 | 67.2 | 62.3 | 56.5 | 76.9 | 1386.1 | 63.1 | 56.7 | 94.0% |
| PruMerge+ (Shang et al., 2024) | 75.3 | 58.8 | **57.7** | 68.1 | 63.5 | 54.0 | 79.5 | 1444.3 | 63.0 | 55.6 | 95.2% |
| TRIM (Song et al., 2025) | 74.9 | 59.9 | 53.5 | 66.2 | 64.3 | 50.2 | 86.5 | 1443.8 | 63.5 | 51.0 | 93.8% |
| VisionZip (Yang et al., 2024) | 76.2 | 58.9 | 56.2 | 67.5 | 63.4 | 58.8 | 82.3 | 1397.1 | 63.3 | 55.9 | 95.8% |
| DART (Wen et al., 2025) | 75.7 | 59.5 | 56.8 | 67.5 | 64.8 | 57.6 | 81.0 | 1419.5 | 64.2 | 55.7 | 96.1% |
| DivPrune (Alvar et al., 2025) | 77.2 | 61.1 | 55.6 | 67.7 | 67.1 | 56.2 | 84.7 | 1423.3 | 63.9 | 55.7 | 96.9% |
| VisPruner | 75.7 | 58.4 | 57.0 | 69.5 | 62.8 | 57.6 | 80.4 | 1370.1 | 63.6 | 55.8 | 95.5% |
| **DOP$_V$ (Ours)** | 77.6 | 60.4 | 56.4 | **69.7** | 65.2 | **59.2** | 84.6 | 1455.7 | 65.1 | **57.5** | 98.2% |
| CDPruner (Zhang et al., 2025) | 78.4 | 61.6 | 55.8 | 67.8 | 67.3 | 57.4 | 87.2 | 1453.0 | 65.5 | 55.7 | 98.1% |
| **DOP$_{CD}$ (Ours)** | **79.1** | **61.7** | 55.4 | 68.0 | **68.4** | 57.9 | **87.4** | **1472.9** | **66.8** | 56.9 | **99.0%** |
| *Retain 9% TFLOPs (= 160 Visual Tokens Every Layer)* | | | | | | | | | | | |
| PruMerge+ (Shang et al., 2024) | 70.5 | 56.2 | **57.2** | 66.9 | 60.1 | 50.3 | 71.1 | 1289.6 | 58.0 | 48.9 | 88.9% |
| TRIM (Song et al., 2025) | 71.0 | 57.4 | 52.9 | 65.5 | 61.8 | 45.8 | 84.8 | 1275.8 | 61.6 | 45.2 | 89.1% |
| VisionZip (Yang et al., 2024) | 71.4 | 55.2 | 55.5 | 67.9 | 58.3 | 55.0 | 74.9 | 1327.8 | 58.6 | 50.4 | 90.3% |
| DART (Wen et al., 2025) | 72.5 | 56.8 | 56.7 | 67.8 | 60.6 | 54.9 | 75.3 | 1325.4 | 62.0 | 53.6 | 92.3% |
| DivPrune (Alvar et al., 2025) | 75.0 | 59.3 | 56.1 | 67.1 | 64.9 | 54.1 | 80.0 | 1356.6 | 62.9 | 53.7 | 94.2% |
| VisPruner | 70.6 | 54.7 | 57.1 | 68.6 | 58.1 | 56.0 | 72.7 | 1226.0 | 60.5 | 51.4 | 90.2% |
| **DOP$_V$ (Ours)** | 74.1 | 57.6 | 56.5 | **68.7** | 61.2 | **57.1** | 79.4 | 1355.6 | 62.1 | 55.1 | 94.1% |
| CDPruner (Zhang et al., 2025) | 76.7 | 60.8 | 55.2 | 67.5 | 65.9 | 55.4 | 86.8 | 1425.3 | 64.2 | 53.8 | 96.3% |
| **DOP$_{CD}$ (Ours)** | **77.5** | **61.2** | 55.1 | 67.5 | **66.7** | 56.7 | **87.5** | **1446.4** | **64.8** | 55.1 | **97.2%** |

Table 10: **Performance comparison of different pruning methods on LLaVA-NeXT-7B (Liu et al., 2024a) under different computation budgets.** We evaluate across 10 benchmarks: We evaluate across 10 benchmarks: VQA$^{V2}$ (Goyal et al., 2017), VizWiz (Gurari et al., 2018), SQA$^I$ (Saikh et al., 2022), GQA (Hudson & Manning, 2019), Seed$^I$ (Li et al., 2023a), MME (Fu et al., 2023), MMB (Liu et al., 2023b), POPE (Li et al., 2023c), MMB$^{CN}$ and VQA$^{Text}$ (Singh et al., 2019). We report accuracy (%) for most benchmarks. Avg. represents the mean performance ratio relative to Vanilla across all benchmarks (higher is better). **Bold** values indicate the best performances in each setting. DOP$_V$ uses VisPruner to reduce visual tokens, while DOP$_{CD}$ uses CDPruner. DOP consistently outperforms all baselines with policies only optimized on a subset of only 100 TextVQA training set samples. Lower Bound indicates the minimum possible computation in our settings. Our comparison matches actual TFLOPs and notes the per-layer token counts that achieve these TFLOPs.

| Method | VQA$^{v2}$ | GQA | VizWiz | SQA$^I$ | Seed$^I$ | VQA$^{Text}$ | POPE | MME | MMB | MMB$^{CN}$ | Rel. Avg. |
|---|---|---|---|---|---|---|---|---|---|---|---|
| *Upper Bound: 100% TFLOPs (= 576 Visual Tokens Every Layer)* | | | | | | | | | | | |
| LLaVA-NeXT-13B | 82.3 | 64.4 | 59.1 | 73.1 | 71.2 | 63.2 | 85.3 | 1539.5 | 68.5 | 61.2 | 100% |
| *Retain 23% TFLOPs (= 640 Visual Tokens Every Layer)* | | | | | | | | | | | |
| FastV (Chen et al., 2024) | 79.4 | 60.9 | 56.4 | 71.7 | 67.5 | 60.7 | 80.2 | 1516.7 | 65.5 | 59.9 | 96.1% |
| PDrop (Xing et al., 2024) | 81.1 | 62.8 | **58.1** | 71.7 | 69.6 | 62.1 | 84.4 | 1559.1 | 66.6 | 60.8 | 98.5% |
| SparseVLM (Zhang et al., 2024c) | 79.9 | 62.7 | 57.5 | **72.5** | 69.5 | **62.8** | 85.6 | 1562.7 | 68.8 | **64.0** | **99.5%** |
| PruMerge+ (Shang et al., 2024) | 78.7 | 62.8 | 56.2 | 70.6 | 69.7 | 56.2 | 83.7 | 1497.3 | 67.4 | 61.9 | 96.7% |
| TRIM (Song et al., 2025) | 79.4 | 63.1 | 54.1 | 71.2 | 69.7 | 57.6 | **87.3** | 1554.6 | 68.7 | 61.2 | 97.6% |
| VisionZip (Yang et al., 2024) | 79.7 | 62.9 | 56.2 | 70.8 | 69.8 | 62.1 | 85.8 | 1549.2 | 68.1 | 62.6 | 98.6% |
| DART (Wen et al., 2025) | 79.3 | 62.7 | 56.2 | 71.0 | 69.2 | 61.3 | 85.2 | 1542.4 | 67.6 | 61.9 | 98.0% |
| DivPrune (Alvar et al., 2025) | 80.4 | 63.5 | 56.7 | 72.2 | 70.1 | 59.2 | 86.5 | 1526.1 | 67.5 | 62.9 | 98.5% |
| VisPruner (Zhang et al., 2024a) | 79.7 | 63.0 | 56.7 | 71.2 | 68.6 | 62.0 | 84.8 | 1561.2 | 67.4 | 62.9 | 98.4% |
| **DOP$_V$ (Ours)** | 80.8 | 63.5 | 55.3 | 71.7 | 70.5 | 61.2 | 86.4 | **1592.2** | **69.5** | 62.6 | 99.3% |
| CDPruner (Zhang et al., 2025) | 81.0 | **64.0** | 57.1 | 71.8 | 70.6 | 61.0 | 86.1 | 1545.6 | 68.9 | 62.1 | 99.2% |
| **DOP$_{CD}$ (Ours)** | **81.4** | 63.9 | 55.7 | **72.5** | 71.2 | 60.0 | 86.1 | 1567.4 | 69.0 | 62.5 | 99.2% |
| *Retain 14% TFLOPs (= 320 Visual Tokens Every Layer)* | | | | | | | | | | | |
| FastV (Chen et al., 2024) | 69.8 | 54.6 | 53.3 | 70.5 | 59.9 | 55.4 | 63.6 | 1279.0 | 59.8 | 54.4 | 86.2% |
| PDrop (Xing et al., 2024) | 75.4 | 57.7 | 52.1 | 72.1 | 63.3 | 56.2 | 74.6 | 1386.3 | 62.8 | 55.3 | 90.5% |
| SparseVLM (Zhang et al., 2024c) | 76.7 | 60.9 | 54.7 | 70.9 | 66.8 | 60.0 | 81.5 | 1491.6 | **68.0** | **63.5** | 96.2% |
| PruMerge+ (Shang et al., 2024) | 75.9 | 61.1 | 53.6 | 70.7 | 66.9 | 55.9 | 79.1 | 1426.5 | 66.6 | 60.6 | 93.9% |
| TRIM (Song et al., 2025) | 75.9 | 61.3 | 52.2 | 69.9 | 67.0 | 52.8 | 87.2 | 1476.6 | 67.3 | 57.4 | 93.9% |
| VisionZip (Yang et al., 2024) | 76.8 | 60.7 | 54.8 | 70.2 | 66.6 | 60.7 | 82.3 | 1487.3 | 66.5 | 62.3 | 95.8% |
| DART (Wen et al., 2025) | 76.4 | 60.9 | 54.2 | 69.8 | 66.8 | 59.7 | 81.1 | 1457.4 | 65.9 | 61.9 | 95.0% |
| DivPrune (Alvar et al., 2025) | 78.1 | 61.8 | 55.0 | **72.3** | 67.8 | 57.6 | 85.2 | 1473.0 | 65.9 | 61.9 | 96.2% |
| VisPruner (Zhang et al., 2024a) | 76.7 | 60.6 | 54.4 | 70.1 | 65.1 | 60.5 | 81.7 | 1523.6 | 66.2 | 62.3 | 95.6% |
| **DOP$_V$ (Ours)** | 78.6 | 61.7 | 54.7 | 71.1 | 67.5 | **61.3** | 84.3 | 1544.7 | 67.2 | 62.4 | 97.2% |
| CDPruner (Zhang et al., 2025) | 79.6 | **63.1** | **55.1** | 71.6 | 68.9 | 58.7 | **87.6** | 1498.5 | 66.3 | 61.8 | 97.3% |
| **DOP$_{CD}$ (Ours)** | **80.2** | 63.1 | 54.5 | 71.5 | 69.9 | 59.5 | **87.6** | **1548.2** | 67.7 | 62.8 | **98.2%** |
| *Retain 9% TFLOPs (= 160 Visual Tokens Every Layer)* | | | | | | | | | | | |
| PruMerge+ (Shang et al., 2024) | 71.6 | 57.9 | 50.8 | 70.1 | 62.5 | 52.8 | 72.1 | 1345.9 | 63.2 | 57.1 | 88.8% |
| TRIM (Song et al., 2025) | 72.1 | 58.9 | 51.2 | 69.1 | 63.4 | 49.2 | 87.0 | 1392.3 | 65.7 | 51.6 | 90.0% |
| VisionZip (Yang et al., 2024) | 72.4 | 57.8 | 52.5 | 69.7 | 62.4 | 58.6 | 76.8 | 1393.9 | 64.8 | 60.0 | 91.5% |
| DART (Wen et al., 2025) | 72.8 | 58.7 | 52.1 | 70.1 | 63.5 | 57.2 | 75.7 | 1389.3 | 64.6 | 60.8 | 91.6% |
| DivPrune (Alvar et al., 2025) | 75.6 | 60.0 | **53.5** | 71.4 | 64.9 | 56.3 | 81.9 | 1436.7 | 65.1 | 60.9 | 93.7% |
| VisPruner (Zhang et al., 2024a) | 72.4 | 57.7 | 52.2 | 70.9 | 61.1 | 58.3 | 76.6 | 1397.0 | 63.4 | 60.2 | 91.2% |
| **DOP$_V$ (Ours)** | 75.1 | 59.3 | 52.7 | 70.7 | 63.6 | **59.5** | 79.7 | 1467.4 | 65.7 | 61.5 | 93.8% |
| CDPruner (Zhang et al., 2025) | 77.8 | 62.2 | 53.1 | 71.7 | 67.1 | 56.7 | **88.3** | 1476.9 | 65.9 | 60.1 | 95.7% |
| **DOP$_{CD}$ (Ours)** | **78.8** | **62.6** | 52.6 | **71.8** | **68.2** | 57.7 | 88.1 | **1504.1** | **66.7** | **62.2** | **96.7%** |

Table 11: **Performance on LLaVA-NeXT-13B (Liu et al., 2024a) under different computation budgets.** We evaluate across 10 benchmarks: We evaluate across 10 benchmarks: VQA$^{V2}$ (Goyal et al., 2017), VizWiz (Gurari et al., 2018), SQA$^I$ (Saikh et al., 2022), GQA (Hudson & Manning, 2019), Seed$^I$ (Li et al., 2023a), MME (Fu et al., 2023), MMB (Liu et al., 2023b), POPE (Li et al., 2023c), MMB$^{CN}$ and VQA$^{Text}$ (Singh et al., 2019). We report accuracy (%) for most benchmarks. Avg. represents the mean performance ratio relative to Vanilla across all benchmarks (higher is better). **Bold** values indicate the best performances in each setting. DOP$_V$ uses VisPruner to reduce visual tokens, while DOP$_{CD}$ uses CDPruner. DOP consistently outperforms all baselines with policies only optimized on a subset of only 100 TextVQA training set samples. Lower Bound indicates the minimum possible computation in our settings. Our comparison matches actual TFLOPs and notes the per-layer token counts that achieve these TFLOPs.

| Method | TextVQA | ChartQA | AI2D | OCRBench | MME | MMB-EN | MMB-CN | Rel. Avg. |
|---|---|---|---|---|---|---|---|---|
| *Upper Bound: 100% TFLOPS (= 1296 Visual Tokens Every Layer)* | | | | | | | | |
| Qwen2.5-VL-7B | 85.5 | 86.5 | 81.0 | 879 | 2286 | 80.2 | 82.7 | 100.0% |
| *Retain 43% TFLOPS (= 512 Visual Tokens Every Layer)* | | | | | | | | |
| FastV (Chen et al., 2024) | 84.1 | 82.2 | 78.8 | 815 | 2317 | 82.0 | 81.8 | 98.0% |
| DivPrune (Alvar et al., 2025) | 81.8 | 79.6 | 78.6 | 800 | 2279 | 81.6 | 82.1 | 96.6% |
| CDPruner (Zhang et al., 2025) | **84.2** | 82.8 | 78.9 | 827 | 2327 | **82.2** | 82.6 | 98.5% |
| Resizing | 81.8 | **85.8** | **80.8** | 856 | 2351 | 79.8 | **82.6** | 99.1% |
| **DOP$_R$ (Ours)** | 82.2 | **85.8** | **80.8** | 868 | **2355** | 80.1 | 82.6 | **99.5%** |
| *Retain 26% TFLOPS (= 256 Visual Tokens Every Layer)* | | | | | | | | |
| FastV (Chen et al., 2024) | 81.5 | 70.9 | 76.2 | 703 | 2238 | 79.6 | 78.9 | 92.0% |
| DivPrune (Alvar et al., 2025) | 76.0 | 65.1 | 76.5 | 692 | 2184 | 80.0 | 79.6 | 89.8% |
| CDPruner (Zhang et al., 2025) | **82.4** | 73.0 | 77.5 | 749 | 2245 | **80.9** | 79.9 | 93.9% |
| Resizing | 76.6 | 82.5 | **79.9** | 813 | 2346 | 80.0 | 81.7 | 96.7% |
| **DOP$_R$ (Ours)** | 79.8 | **83.1** | 79.8 | 828 | **2351** | 79.7 | **82.4** | **97.7%** |
| *Retain 16% TFLOPS (= 128 Visual Tokens Every Layer)* | | | | | | | | |
| FastV (Chen et al., 2024) | 73.8 | 52.2 | 71.4 | 531 | 2008 | 72.9 | 72.2 | 80.2% |
| DivPrune (Alvar et al., 2025) | 67.0 | 50.4 | 72.1 | 549 | 2108 | 77.8 | 77.8 | 81.6% |
| CDPruner (Zhang et al., 2025) | **77.8** | 59.2 | 74.0 | 632 | 2127 | 76.2 | 76.5 | 86.2% |
| Resizing | 67.0 | 61.8 | 78.5 | 728 | **2322** | 79.3 | 81.3 | 89.7% |
| **DOP$_R$ (Ours)** | 69.9 | **70.8** | **79.1** | 741 | 2303 | **79.9** | **81.5** | **92.1%** |

Table 12: **Performance comparison on Qwen2.5-VL-7B under different computation budgets.** We evaluate across 7 benchmarks: TextVQA, ChartQA, AI2D, OCRBench, MME, MMB-EN, and MMB-CN. **Rel. Avg.** is the mean performance ratio relative to the full model.

| Method | TextVQA | ChartQA | AI2D | OCRBench | MME | MMB-EN | MMB-CN | Rel. Avg. |
|---|---|---|---|---|---|---|---|---|
| *Upper Bound: All 1280 Tokens (100%)* | | | | | | | | |
| InternVL3-8B | 81.5 | 85.1 | 85.2 | 853 | 2394 | 83.9 | 82.6 | 100.0% |
| *Retain 27% TFLOPs (= 256 Visual Tokens Every Layer)* | | | | | | | | |
| FastV (Chen et al., 2024) | 74.4 | 70.7 | 82.2 | 632 | 2348 | 83.6 | 82.0 | 91.7% |
| DivPrune (Alvar et al., 2025) | 64.7 | 57.5 | 80.9 | 477 | 2249 | 80.8 | 80.2 | 83.6% |
| CDPruner (Zhang et al., 2025) | **75.7** | 72.0 | 82.7 | 640 | 2334 | 83.5 | 81.7 | 92.2% |
| Resizing | 70.3 | **75.0** | 84.1 | **743** | **2369** | 84.4 | 85.0 | 94.7% |
| **DOP$_R$ (Ours)** | 71.9 | 74.4 | **85.1** | 732 | 2339 | **85.3** | **86.1** | **95.0%** |
| *Retain 18% TFLOPs (= 128 Visual Tokens Every Layer)* | | | | | | | | |
| FastV (Chen et al., 2024) | 63.7 | 46.9 | 77.3 | 426 | 2250 | 81.3 | 80.2 | 80.3% |
| DivPrune (Alvar et al., 2025) | 55.6 | 42.7 | 76.4 | 378 | 2166 | 78.4 | 77.6 | 75.8% |
| CDPruner (Zhang et al., 2025) | **67.5** | 50.8 | 79.9 | 471 | 2282 | 82.1 | 80.3 | 83.1% |
| Resizing | 57.6 | 49.0 | 82.0 | 573 | 2301 | 82.5 | 82.5 | 83.7% |
| **DOP$_R$ (Ours)** | 64.6 | **60.9** | **83.5** | 659 | **2354** | **84.8** | **84.9** | **89.8%** |

Table 13: **Performance comparison on InternVL3-8B under different computation budgets.** We evaluate across 7 benchmarks (TextVQA, ChartQA, AI2D, OCRBench, MME, MMB-EN, MMB-CN). Rel. Avg. denotes the mean performance ratio relative to the original model. **Bold** indicates the best result within each budget.

### E.5 ALL DOP POLICIES

DOP policies for LLaVA-series models across different TFLOPs ratios are presented in Table 15, while policies for Qwen2.5-VL and InternVL3 are shown in Table 16.

---

[4] https://docs.scipy.org/doc/scipy/reference/generated/scipy.stats.spearmanr.html

| Model | LLaVA-1.5 | LLaVA-Next | Qwen2.5-VL | InternVL3 |
|---|---|---|---|---|
| Size | 7B / 13B | 7B / 13B | 7B | 8B |
| Cost (min) | 5 / 7 | 12 / 18 | 10 | 7 |

Table 14: Optimization cost (minutes) on a single NVIDIA A100-80G GPU across different models. This cost represents the one-time expense to evaluate sampled individual divergences on 100 TextVQA samples (note that we only need to generate the first token for each sample). Subsequently, solving optimization for different TFLOPs ratios to obtain policies requires only negligible CPU operations.

| | LLaVA-1.5-7B | | | | LLaVA-1.5-13B | | | | LLaVA-Next-7B | | | | LLaVA-Next-13B | | | |
|---|---|---|---|---|---|---|---|---|---|---|---|---|---|---|---|---|
| TFLOPs | 100% | 37% | 28% | 23% | 100% | 37% | 28% | 23% | 100% | 23% | 14% | 9% | 100% | 23% | 14% | 9% |
| $d_A$ | 32 | 25 | 26 | 25 | 40 | 40 | 40 | 31 | 32 | 25 | 29 | 30 | 40 | 340 | 38 | 38 |
| $d_P$ | 32 | 19 | 19 | 16 | 40 | 20 | 21 | 16 | 32 | 16 | 16 | 16 | 40 | 19 | 20 | 20 |
| $n_v$ | 576 | 192 | 97 | 51 | 576 | 192 | 96 | 58 | 2880 | 965 | 485 | 240 | 2880 | 960 | 480 | 240 |

Table 15: **DOP policies for all LLaVA models across TFLOPs ratios.** Rows list MHA depth ($d_A$), MLP depth ($d_P$), and visual token count ($n_v$). For each model, the 100% TFLOPs configuration corresponds to $d_A$ and $d_P$ equal to the number of decoder layers, and $n_v$ equal to the original visual token count. The policies shown are for DOP$_{\text{CD}}$, while DOP$_{\text{V}}$ policies are basically identical.

### E.6 Performance on Image Captioning Tasks

We evaluated DOP's performance on image captioning tasks, which involve long-form generation, using NoCaps (Agrawal et al., 2019) and TextCaps (Sidorov et al., 2020) datasets. Experiments were conducted on LLaVA-1.5-7B with CDPruner for token reduction, comparing against CDPruner at equivalent TFLOPs using the lmm-eval framework.

Table 17 shows that DOP consistently outperforms CDPruner on image captioning tasks, further demonstrating its generalization capability. We will include additional long-form generation tasks such as detailed image analysis and story generation with comprehensive analysis of visual grounding abilities in the final version.

### E.7 Video Understanding Tasks

We followed VisPruner's (Zhang et al., 2024a) video benchmark settings, applying DOP on Video-LLaVA (Lin et al., 2023a), evaluating on TGIF-QA (Jang et al., 2017), MSVD-QA and MSRVTT-QA (Xu et al., 2017), using the first 1K samples from each benchmark and employing VisPruner as the visual token reduction method. Video-LLaVA processes 8 frames of 224-resolution video, totaling 2048 visual tokens.

Table 18 shows that DOP provides consistent improvements over VisPruner baseline across video understanding tasks. At both 16% and 11% TFLOPs retention levels, DOP + VisPruner achieves better performance than VisPruner alone across all three benchmarks in both accuracy and score metrics, demonstrating that DOP's operation-level pruning strategy effectively extends to video understanding scenarios.

### E.8 CDPruner Implementation and Reproduction Results

We did not combine DOP with CDPruner on Qwen2.5-VL in our main results because CDPruner's authors have not released official implementation for Qwen2.5-VL, and our reproduction fails to match their reported performance. We therefore used resizing as the token reduction baseline and compared against CDPruner's reported performance. Notably, resizing already outperforms CDPruner's reported results, yet DOP still achieves significant improvements.

Below we provide results with our reproduced CDPruner, showing DOP consistently delivers stable improvements over our reproduced CDPruner on Qwen2.5-VL.

| | Qwen2.5-VL-7B | | | | InternVL3-8B | | |
|---|---|---|---|---|---|---|---|
| **TFLOPs** | 100% | 43% | 26% | 16% | 100% | 27% | 18% |
| $d_A$ | 28 | 28 | 28 | 28 | 28 | 28 | 28 |
| $d_P$ | 28 | 20 | 16 | 17 | 28 | 17 | 19 |
| $n_v$ | 1296 | 640 | 384 | 165 | 1280 | 361 | 169 |

Table 16: **DOP policies for Qwen2.5-VL-7B and InternVL3-8B across TFLOPs ratios.** Rows list MHA depth ($d_A$), MLP depth ($d_P$), and visual token count ($n_v$), where 100% TFLOPs corresponds to $d_A = d_P =$ number of decoder layers and $n_v =$ original visual token count.

| TFLOPs | Method | NoCaps | TextCaps |
|---|---|---|---|
| 100% | Baseline | 1.043 | 0.992 |
| 28% | CDPruner | 0.912 | 0.863 |
| | $DOP_C$ | 0.948 | 0.883 |
| 23% | CDPruner | 0.844 | 0.767 |
| | $DOP_C$ | 0.881 | 0.790 |

Table 17: Performance comparison on image captioning tasks. CIDEr score is reported.

### E.9 POLICY ANALYSIS ACROSS DIFFERENT OPTIMIZATION TASKS

We also detail the policies optimized on different tasks in Table 20, using LLaVA-1.5-7B optimized on TextVQA, VQAv2, and SeedBench as examples. As shown in the table, policies optimized on TextVQA and VQAv2 are largely similar, while SeedBench exhibits significantly different behavior—more aggressively pruning operations while retaining more tokens. These policy differences explain the performance variations in Table 4. Appendix F.2 demonstrates through divergence analysis that SeedBench exhibits significantly higher redundancy in deeper layer operations compared to other benchmarks, leading to more aggressive pruning of deep operations and retention of more tokens for important shallow operations.

## F ADDITIONAL ABLATION

### F.1 OPTIMIZATION EFFICIENCY ABLATION

Table 21 shows the performance and optimization cost with different numbers of optimization samples on LLaVA-1.5-7B at 28% TFLOPs. The results show minimal performance differences across 25-500 samples, with only 10 samples causing substantial degradation while still outperforming CDPruner. This validates our default 100 samples setting and shows we can reduce to 25 samples to reduce time cost to 2 minutes with comparable performance, though we default to using 100 TextVQA samples in our main experiments since this configuration is already sufficiently efficient.

### F.2 DIVERGENCE VISUALIZATION

In Figure 4, we present individual divergences evaluated on different benchmarks with respect to MHA depth, MLP depth, and visual token count. We observe that on SeedBench, pruning MHA and MLP visual operations after half of the layers causes virtually no divergence, whereas TextVQA and VQAv2 only exhibit small divergence after the 26th layer. This phenomenon may be attributed to SeedBench's multiple-choice instruction format, where answers are embedded within the text, allowing the model to accomplish the task using only shallow visual operations.

Note that the absolute values of divergences across different tasks are not directly comparable due to variations in data characteristics and task formats. For instance, higher divergence values on VQAv2 compared to TextVQA do not imply that VQAv2 is more sensitive to pruning; only divergences within the same task can be meaningfully compared.

| Method | TGIF-QA | | MSVD-QA | | MSRVTT-QA | | Average | |
|---|---|---|---|---|---|---|---|---|
| | Acc. | Score | Acc. | Score | Acc. | Score | Acc. | Score |
| **Upper Bound, All 2048 Tokens (100%)** | | | | | | | | |
| Video-LLaVA | 19.8 | 2.53 | 70.5 | 3.93 | 57.5 | 3.50 | 49.3 | 3.32 |
| **Retain 16% TFLOPs** | | | | | | | | |
| VisPruner | 15.9 | 2.41 | 69.3 | 3.92 | 55.6 | 3.45 | 46.9 | 3.26 |
| DOP + VisPruner | 16.3 | 2.44 | 69.5 | 3.92 | 55.9 | 3.46 | 47.2 | 3.27 |
| **Retain 11% TFLOPs** | | | | | | | | |
| VisPruner | 14.1 | 2.35 | 65.4 | 3.79 | 54.1 | 3.41 | 44.5 | 3.18 |
| DOP + VisPruner | 14.6 | 2.36 | 65.8 | 3.80 | 54.4 | 3.42 | 44.9 | 3.19 |

Table 18: Performance comparison on video understanding tasks.

| Method | TextVQA | ChartQA | AI2D | OCRBench | MME | MMB-EN | MMB-CN | Rel. Avg. |
|---|---|---|---|---|---|---|---|---|
| **1296 (Full)** | | | | | | | | |
| Full Model | 85.5 | 86.5 | 81 | 879 | 2286 | 80.2 | 82.7 | **100%** |
| **26% TFLOPs (= 256 Visual Tokens Every Layer)** | | | | | | | | |
| CDPruner (paper) | 82.4 | 73 | 77.5 | 749 | 2245 | 80.9 | 79.9 | 93.9% |
| CDPruner (reproduce) | 76.3 | 70.4 | 76.3 | 705 | 2234 | 79.8 | 80.4 | 91.4% |
| CDPruner (reproduce)+DOP | 78.2 | 72.1 | 77.5 | 723 | 2232 | 79.7 | 80.3 | 92.4% |
| **16% TFLOPs (= 128 Visual Tokens Every Layer)** | | | | | | | | |
| CDPruner (paper) | 77.8 | 59.2 | 74 | 632 | 2127 | 76.2 | 76.5 | 86.1% |
| CDPruner (reproduce) | 70.5 | 52.4 | 72.6 | 562 | 2187 | 77.4 | 78.1 | 83.3% |
| CDPruner (reproduce)+DOP | 73.2 | 57.8 | 73.2 | 583 | 2203 | 77.6 | 78.1 | 85.2% |

Table 19: Comparison of CDPruner reproduction results with DOP on Qwen2.5-VL.

## F.3 ADDITIVE APPROXIMATION VS. DIRECT JOINT DIVERGENCE OPTIMIZATION

We include a direct validation by evaluating joint KL divergence in the policy subspace through grid search for optimal policies and comparing with policies obtained via additive approximation. The minimal differences in downstream task performance demonstrate the reasonableness of our additive approximation.

We conducted a grid search experiment in a policy subspace based on direct joint divergence evaluation. We create a sufficiently small policy space that allows us to exhaustively evaluate the joint divergence of each policy by sampling MHA and MLP depths every 2 steps ($d_A, d_P \in \{1, 3, 5, 7, ...\}$) and visual token counts every 16 steps ($n_v \in \{0, 16, 32, ...\}$). Given different TFLOPs targets, we first solved Equation (7) using additive approximation, then evaluated joint KL divergence for all constraint-satisfying policies on the same validation set, selecting the policy with minimum divergence. On LLaVA-1.5-7B with 28% and 23% target TFLOPs ratios, we found 658 and 384 valid policies respectively.

Table 22 show that direct grid search with joint KL divergence (w/o Additive Approximation) achieves slightly better performance, but policies with additive approximation achieve very similar results. This further validates our additive approximation approach and confirms that despite being a coarse approximation, it enables DOP to achieve good policies with significantly reduced computational cost.

## F.4 ROBUSTNESS ANALYSIS OF MULTIPLE OPTIMIZATION RUNS

To verify the robustness of our method across multiple optimization runs, we evaluate our method's stability across different validation sets by using 5 different random seeds to create validation sets of 100 TextVQA samples. The mean performance and standard deviations across these runs demonstrate the consistency of our optimization approach.

| TFLOPs | 37% | | | 28% | | | 23% | | |
|---|---|---|---|---|---|---|---|---|---|
| Optimization Task | TextVQA | VQAv2 | SeedBench | TextVQA | VQAv2 | SeedBench | TextVQA | VQAv2 | SeedBench |
| $d_A$ | 25 | 25 | 16 | 26 | 25 | 12 | 25 | 25 | 12 |
| $d_P$ | 19 | 18 | 12 | 19 | 19 | 12 | 16 | 18 | 12 |
| $n_v$ | 192 | 197 | 293 | 97 | 98 | 167 | 51 | 48 | 78 |

Table 20: Policy differences across optimization tasks for LLaVA-1.5-7B.

| #Samples | 10 | 25 | 50 | 100 | 200 | 500 | CDPruner |
|---|---|---|---|---|---|---|---|
| Cost (min) | 1 | 2 | 3 | 5 | 10 | 24 | - |
| Rel. Avg. | 97.0% | 97.7% | 97.3% | 97.9% | 97.6% | 97.5% | 96.7% |

Table 21: **Optimization Efficiency Ablation.** Performance of LLaVA-1.5-7B with different numbers of samples for optimization, showing optimization cost on a single NVIDIA A100-80G GPU and relative performance when retaining 28% TFLOPs across the same benchmarks as Table 8. Optimization with 25 to 500 samples shows minimal performance differences.

According to Table 23, the low standard deviations across different validation sets confirm that our DOP method produces consistent and robust optimization results, demonstrating stability across multiple optimization runs with different data sampling.

### F.5 DIRECT DOWNSTREAM METRIC OPTIMIZATION

DOP is not limited to KL divergence optimization - it can optimize directly using downstream task metrics. We replace the divergence $D(d_A, d_P, n_v)$ in Section 3.2.2 with performance drop $D_{acc}(d_A, d_P, n_v) = \text{Accuracy}(L, \tilde{L}, N_v) - \text{Accuracy}(d_A, d_P, n_v)$, optimizing on 100 TextVQA validation samples. The Equation (7) is adjusted accordingly:

$$(d_A^*, d_P^*, n_v^*) = \arg \min_{(d_A, d_P, n_v)} \hat{D}_{A,acc}(d_A) + \hat{D}_{P,acc}(d_P) + \hat{D}_{v,acc}(n_v) \tag{13}$$

$$\text{s.t.} \quad \text{TFLOPs}(d_A, d_P, n_v) \leq \tau$$

$$n_v \geq n_{\min}$$

, where $\hat{D}_{A,acc}$, $\hat{D}_{P,acc}$, $\hat{D}_{v,acc}$ are interpolated individual accuracy drops for MHA depth, MLP depth, and visual token count respectively, and $n_{\min}$ is the minimum visual token count. The interpolation process follows the same procedure as in Appendix C.1, but replaces divergence measurements with accuracy drop measurements.

Table 24 shows the results of optimizing directly on downstream metrics, which still outperform the CDPruner baseline. However, KL divergence yields better policies because the effect of pruning on downstream metrics is less smooth than on KL divergence and cannot accurately reflect changes in model capabilities: some questions are always correct or incorrect regardless of pruning, while even those samples that do change only shift abruptly after pruning accumulates to a certain threshold, rather than showing gradual degradation.

### F.6 EXTENSION TO MULTIPLE TOKEN GROUPS

We primarily apply DOP to visual tokens for fair comparison with token pruning baselines that only prune visual tokens. However, DOP can indeed be extended to other token groups. Here we demonstrate pruning operations for both system and visual tokens on LLaVA-1.5-7B.

Since we cannot change the number of system tokens, we prune their MHA and MLP operations in depth order by adding $d_A^s$ and $d_P^s$ (system token MHA and MLP operation depths) and their corresponding individual divergences to Equation (7):

$$(d_A^{s*}, d_P^{s*}, d_A^*, d_P^*, n_v^*) = \arg \min_{(d_A^s, d_P^s, d_A, d_P, n_v)} \hat{\mathcal{D}}_A^s(d_A^s) + \hat{\mathcal{D}}_P^s(d_P^s) + \hat{\mathcal{D}}_A(d_A) + \hat{\mathcal{D}}_P(d_P) + \hat{\mathcal{D}}_v(n_v)$$

$$\text{s.t.} \quad \text{TFLOPs}(d_A^s, d_P^s, d_A, d_P, n_v) \leq \tau \tag{14}$$

$$n_v \geq n_{\min}$$

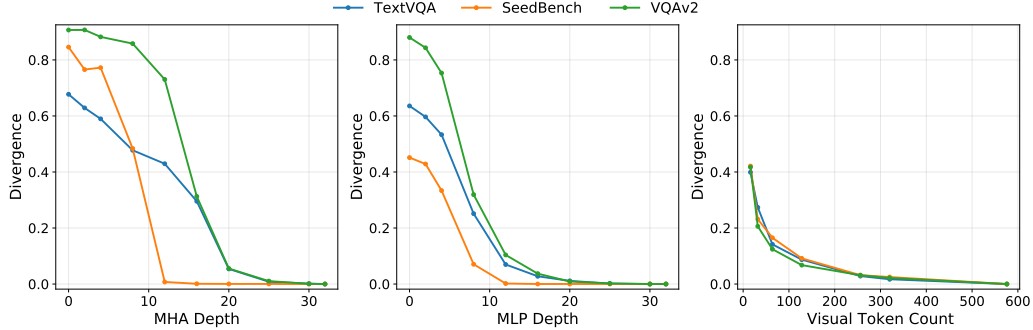

Figure 4: Individual divergence patterns across different benchmarks on LLaVA-1.5-7B. We visualize the three individual divergences $\hat{\mathcal{D}}_A(d_A)$, $\hat{\mathcal{D}}_P(d_P)$, and $\hat{\mathcal{D}}_v(n_v)$ with respect to MHA depth, MLP depth, and visual token count respectively. Each subplot shows divergences evaluated on three benchmarks, including TextVQA (blue), VQAv2 (green), and SeedBench (orange). Notably, Seed-Bench exhibits significantly higher redundancy in deeper layer operations compared to the other two benchmarks.

| Optimization Method | $d_A$ | $d_P$ | $n_v$ | Performance | | |
| --- | --- | --- | --- | --- | --- | --- |
| | | | | TextVQA | GQA | SeedBench |
| **28% TFLOPs** | | | | | | |
| w/ Additive Approximation | 28 | 18 | 96 | 55.7 | 59.4 | 63.5 |
| w/o Additive Approximation | 26 | 20 | 96 | 55.8 | 59.6 | 63.4 |
| **23% TFLOPs** | | | | | | |
| w/ Additive Approximation | 28 | 16 | 48 | 53.8 | 57.7 | 62.0 |
| w/o Additive Approximation | 24 | 18 | 48 | 54.1 | 57.8 | 62.0 |

Table 22: Comparison of additive approximation with direct joint divergence optimization.

, where $\hat{\mathcal{D}}_A^s(d_A^s)$ and $\hat{\mathcal{D}}_P^s(d_P^s)$ are interpolated KL divergences corresponding to pruning MHA and MLP operations for system tokens, respectively.

Table 25 shows the results of simultaneously pruning visual and system operations, with the same settings as in the main paper, performing optimization on LLaVA-1.5-7B with CDPruner for visual token reduction. Simultaneously pruning vision and system token operations provides consistent performance improvements over vision-only pruning, demonstrating DOP's generality.

### F.7 VALIDATION OF DEPTH-WISE REDUNDANCY PATTERN

To validate our depth-wise pruning assumption, we conduct experiments to directly assess operation redundancy at different depths through their impact on downstream task performance.

#### F.7.1 PRUNING CONSECUTIVE OPERATIONS AT DIFFERENT DEPTHS

We prune consecutive operations of the same module and group type at various layer positions. We first prune a single MHA/MLP operation for visual tokens on LLaVA-1.5-7B, then prune three consecutive MHA/MLP operations at different positions to make the redundancy pattern more pronounced. We also conduct similar experiments on Qwen2.5-VL-7B.

The results are shown in Tables 26 to 28. Generally, pruning deeper layers results in smaller performance drops, indicating that the pruned operations are more redundant. The pattern of deeper operations being more redundant becomes more evident when pruning three consecutive operations. On Qwen2.5-VL-7B, the trend generally holds with some variations in MHA operations but clear patterns in MLP operations, confirming that our depth-wise assumption is largely reasonable across different model architectures.

| TFLOPs Ratio | TextVQA | GQA | SeedBench |
|:---:|:---:|:---:|:---:|
| 28% | 55.7±0.2 | 59.5±0.1 | 63.4±0.1 |
| 23% | 54.1±0.3 | 58.0±0.2 | 62.1±0.3 |

Table 23: Robustness analysis across multiple optimization runs with different validation sets.

| TFLOPs Ratio | Method | Metric | TextVQA | GQA | SeedBench |
|:---:|:---:|:---:|:---:|:---:|:---:|
| 28% | CDPruner | - | 55.3 | 58.6 | 62.5 |
| | $DOP_R$ | KL Divergence | 55.8 | 59.6 | 63.6 |
| | | Accuracy Drop | 55.6 | 59.3 | 63.4 |
| 23% | CDPruner | - | 53.2 | 57.0 | 60.9 |
| | $DOP_R$ | KL Divergence | 54.2 | 58.1 | 62.2 |
| | | Accuracy Drop | 53.7 | 57.6 | 61.6 |

Table 24: Comparison of DOP optimization using KL divergence vs. direct accuracy metrics.

### F.7.2 POLICY PERTURBATION ANALYSIS

We conduct policy perturbation experiments to validate the near-optimality of our DOP constraint. Starting from our original DOP policy, we create alternative non-consecutive pruning strategies by randomly swapping $n$ operations within the same module-group type: removing $n$ retained operations and keeping $n$ originally pruned operations, thereby breaking the depth-wise constraint. We repeat this process 10 times to sample alternative policies. This reverse validation approach is straightforward: if our DOP policy were far from optimal, we should easily find significantly better policies through random perturbations. Conversely, if DOP consistently outperforms the majority of these random alternatives, this provides strong evidence for its near-optimality.

The results are shown in Tables 29 and 30. For swapping 1 operation, while we can occasionally find slightly better policies, our original DOP policy significantly outperforms the average performance of perturbed policies. When we increase the perturbation magnitude by swapping 3 operations, we can hardly find any policy that outperforms DOP, and the average performance of perturbed policies deteriorates further. These consistent results across different models provide strong evidence that our depth-wise constraint indeed captures fundamental principles of MLLMs, rather than being an arbitrary design choice.

| TFLOPs Ratio | Group | TextVQA | GQA | SeedBench |
|---|---|---|---|---|
| 28% | Vision | 55.8 | 59.6 | 63.6 |
| | Vision+System | 55.9 | 60.1 | 63.8 |
| 23% | Vision | 54.2 | 58.1 | 62.2 |
| | Vision+System | 54.4 | 58.5 | 62.7 |

Table 25: Comparison of DOP applied to vision tokens only vs. both vision and system tokens.

| Operation Pruned | Dataset | $l=1$ | $l=4$ | $l=8$ | $l=12$ | $l=16$ | $l=20$ | $l=24$ | $l=28$ | $l=32$ |
|---|---|---|---|---|---|---|---|---|---|---|
| $(g_v, l, \mathrm{MHA})$ | TextVQA | 55.8 | 57.2 | 57.2 | 56.6 | 58.1 | 57.9 | 57.9 | 58.1 | 58.1 |
| | SeedBench | 26.1 | 66.2 | 65.4 | 64.3 | 66.1 | 66.2 | 66.1 | 66.2 | 66.2 |
| $(g_v, l, \mathrm{MLP})$ | TextVQA | 58.3 | 58.2 | 57.9 | 58.2 | 58.0 | 58.3 | 58.1 | 58.1 | 58.3 |
| | SeedBench | 66.1 | 66.0 | 65.5 | 66.0 | 66.1 | 66.1 | 66.1 | 66.1 | 66.1 |

Table 26: Performance of pruning single operation on LLaVA-1.5-7B.

| Operation Pruned | Dataset | $l=1,2,3$ | $l=4,5,6$ | $l=8,9,10$ | $l=12,13,14$ | $l=16,17,18$ | $l=20,21,22$ | $l=24,25,26$ | $l=28,29,30$ |
|---|---|---|---|---|---|---|---|---|---|
| $(g_v, l, \mathrm{MHA})$ | TextVQA | 44.9 | 55.3 | 53.5 | 56.5 | 56.5 | 56.7 | 58.0 | 57.7 |
| | SeedBench | 60.4 | 62.9 | 59.7 | 62.2 | 66.1 | 66.2 | 66.1 | 65.8 |
| $(g_v, l, \mathrm{MLP})$ | TextVQA | 58.3 | 57.3 | 55.4 | 57.6 | 57.7 | 58.2 | 58.2 | 58.4 |
| | SeedBench | 66.1 | 64.8 | 63.2 | 66.1 | 66.2 | 66.1 | 66.1 | 66.1 |

Table 27: Performance of pruning 3 consecutive operations on LLaVA-1.5-7B.

| Operation Pruned | $l=1,2,3$ | $l=4,5,6$ | $l=8,9,10$ | $l=12,13,14$ | $l=16,17,18$ | $l=20,21,22$ | $l=24,25,26$ |
|---|---|---|---|---|---|---|---|
| $(g_v, l, \mathrm{MHA})$ | 1656 | 2148 | 2194 | 1959 | 2213 | 2295 | 2295 |
| $(g_v, l, \mathrm{MLP})$ | 1816 | 2236 | 2244 | 2242 | 2308 | 2303 | 2297 |

Table 28: MME performance of pruning 3 consecutive operations on Qwen2.5-VL-7B.

| Policy | Metric | TextVQA | GQA | SeedBench |
|---|---|---|---|---|
| Original DOP | - | 55.8 | 59.6 | 63.6 |
| Swap 1 MHA OP | Best | 55.8 | 59.8 | 63.6 |
| | Mean | 54.3 | 57.7 | 61.4 |
| Swap 3 MHA OP | Best | 55.1 | 58.9 | 62.9 |
| | Mean | 51.8 | 54.7 | 59.5 |
| Swap 1 MLP OP | Best | 55.9 | 59.7 | 63.6 |
| | Mean | 55.3 | 58.8 | 63.0 |
| Swap 3 MLP OP | Best | 55.4 | 59.3 | 63.2 |
| | Mean | 52.7 | 56.1 | 60.6 |

Table 29: Policy perturbation results on LLaVA-1.5-7B at 28% TFLOPs. 10 repetitions for each perturbation.

| Policy | Metric | TextVQA | ChartQA | MME |
|---|---|---|---|---|
| Original DOP | - | 79.8 | 83.1 | 2351 |
| Swap 1 MLP OP | Best | 80.2 | 82.6 | 2347 |
| | Mean | 77.5 | 81.6 | 2331 |
| Swap 3 MLP OP | Best | 77.1 | 81.4 | 2316 |
| | Mean | 73.8 | 78.5 | 2236 |

Table 30: Policy perturbation results on Qwen2.5-VL-7B at 26% TFLOPs. 10 repetitions for each perturbation.

