# OpenReview forum: "Fine-grained Token Allocation Via Operation Pruning for Efficient MLLMs"
_ICLR.cc/2026/Conference — Submitted to ICLR 2026_

### Official Review · Reviewer_ZZ6f · 2025-10-16

**Soundness:** 3
**Presentation:** 3
**Contribution:** 2
**Rating:** 4
**Confidence:** 3

**Summary:**

This paper investigates the token computation allocation problem in multimodal large models across three dimensions: tokens, model depth, and computational modules. It proposes DOP, a token pruning method (can be considered as token early exiting). To reduce the size of the strategy design space, this paper introduces an Additive Approximation approach, which effectively reduces the complexity of divergence evaluation. Furthermore, sparse sampling with interpolation is adopted to reduce the final strategy search complexity, enabling the algorithm to complete with only a small calibration set and limited search time. The experimental section is comprehensive, evaluating multimodal large models with different architectures and scales across multiple benchmarks and comparative methods, demonstrating the effectiveness of the proposed approach.

**Strengths:**

* The paper proposes several improvements addressing aspects such as strategy search complexity and data dependency. The resulting algorithm achieves low computational complexity and requires only a minimal calibration set.
* Searching different exiting point for {attn, mlp} is interesting.
* The paper provides comprehensive experiments, validating the algorithm's generalizability across multimodal large models with diverse architectures and scales.

**Weaknesses:**

* Limited overall novelty. The core contribution--allocating computation for tokens along the depth dimension--is closely related to existing ideas of token-level early exit and dynamic computation allocation. Similarly, the decoupling of {attn, mlp} modules has also been explored in prior work on dynamic computation allocation. Here lists some works:

1. [ICLR 25] Routing Experts: Learning to Route Dynamic Experts in Multi-modal Large Language Models. It learns to dynamically skip some layer for <Q, A> pair tokens, achieves depth flexibility, which can also be applied to visual tokens.
2. [ACL 21] Accelerating BERT Inference for Sequence Labeling via Early-Exit. It learns token-level early-exit policy for inference acceleration. Its search strategy differs from that of this paper, but their core objectives are consistent.
3. [ICML 25] SkipGPT: Dynamic Layer Pruning Reinvented with Token Awareness and Module Decoupling. Token-level dynamic computation allocation with decoupled modules, i.e. attn/mlp.

* I think the paper appears to overestimate the complexity of the original search space (Sec. 3.1 - policy space size). In general, token pruning can be categorized as: (1) One-shot pruning: Redundant tokens are filtered out based on importance before being processed by the large model. (2) Early exit: The system decides during the forward pass at which layer a token should exit, after which it undergoes no further computation.(3) Dynamic computation allocation: Before each computational unit, a decision is made independently on whether to execute the current unit for the token. Unlike early exit, a token that "exiting" for one unit may still be executed in the next. In my view, the method in this paper falls into category (2), early exit, with a search space of $2 \times L \times N$ (finding an exit point for each token in both the attn and mlp modules across $L=32$ layers). In contrast, the search space for the  dynamic computation allocation in category (3) would be $2^{(2L)} \times N$.


* Need additional evidence. In Sec.3.2.1, the authors claim that: ‘we observe that for the same group-module type (g, m), a deeper operation (g, l2, m) remains generally more redundant than a shallower operation (g, l1, m) with $l2 > l1’$. However, no experimental evidence or cites are included. I think it is a crucial observation for the designing of method, i.e., if deeper operations are more redundant, then token early-exit (type 2) is sufficient, re-use token (type 3) in deeper layers is unnecessary.

**Questions:**

Although the authors empirically demonstrate a high correlation coefficient between the estimated divergence and the joint divergence when used as a ranking criterion, I remain interested in the error of the Additive Approximation. It would be very informative to see: how large is the gap between the strategies searched using the approximated divergence versus the true joint divergence (can be examined in smaller-scale scenarios, e.g., with smaller MLLM and less data)? Furthermore, what would be the corresponding impact on downstream task performance? I believe the paper would be strengthened by including such an ablation study.

---

> ### Author Response · Authors · 2025-11-25
>
> >Limited overall novelty. The core contribution--allocating computation for tokens along the depth dimension--is closely related to existing ideas of token-level early exit and dynamic computation allocation. Similarly, the decoupling of {attn, mlp} modules has also been explored in prior work on dynamic computation allocation.
>
> We respectfully disagree with framing the broad research area of dynamic computation allocation and token-level early exit as opposing our novelty. The similarity lies in the general improvement direction, which validates the reasonableness of our motivation and design philosophy but does not diminish our novelty. The primary novelty lies in the adaptation and implementation details for specific tasks and scenarios.
>
> Moreover, we want to emphasize a fundamental difference in design philosophy between our DOP and these representative works. While both approaches comprehensively consider layer/module redundancy and token redundancy, DOP and prior token-level early exit and dynamic computation allocation works diverge fundamentally in token redundancy: they focus on redundancy differences between individual tokens, requiring model modifications (e.g., routers, token selectors) and substantial training data. In contrast, DOP recognizes modality-level redundancy differences in MLLMs, enabling modality-wide dynamic computation without architectural changes or extensive fine-tuning, thus achieving our objective: MLLM acceleration with minimal resources. We discuss the three representative works provided by the reviewer in the following:
>
> **1. Routing Experts:** This method requires expensive three-stage training with 333k samples due to Router and Adapter components, making it unsuitable for resource-efficient acceleration and preventing plug-and-play deployment. In contrast, DOP achieves strong generalization with minimal data (as few as 25 samples). Additionally, their approach operates at the layer level, while our method achieves finer module/operation-level granularity.
>
> **2. Accelerating BERT Inference:** This work requires additional Token-Level Off-Ramps and extensive fine-tuning data, making it neither directly applicable nor adaptable for few-shot optimization. While both works recognize that different tokens need different processing depths, they address intra-modal token redundancy whereas we target inter-modal redundancy differences in MLLMs—a fundamental distinction driving all subsequent design choices.
>
> **3. SkipGPT:** Similarly, this work introduces routers that require training with substantial data (850,000 samples), making it unsuitable for direct application and preventing few-shot fine-tuning or plug-and-play deployment across different models.
>
> We appreciate the reviewer for highlighting the relevant field of dynamic computation allocation and token-level early exit. We will enhance our related work section in the final version to better position our contributions within this broader research landscape.
>
> >I think the paper appears to overestimate the complexity of the original search space (Sec. 3.1 - policy space size).
>
> Thank you for your careful reading. In fact, in Section 3.1, we present a general operation pruning framework where operations do not necessarily need to be pruned consecutively, which aligns with your category (3) Dynamic computation allocation. In Section 3.2, we introduce the Depth-wise Pruning Constraint to improve optimization efficiency, which transforms it into your category (2) Early exit scenario, and we provide the corresponding policy space size in lines 250-252. This policy space size is $(L+1)^2 \times (N_v+1)$, corresponding to: first pruning from $N_v$ visual tokens to $n_v$ tokens as a whole group before entering the LLM decoder, then finding 2 exit points where all $n_v$ visual tokens collectively "early exit" from attention and MLP modules respectively across layers, yielding the final combinations.

---

> ### Author Response · Authors · 2025-11-25
>
> >Need additional evidence...deeper operations are more redundant
>
> Our depth-wise assumption is a coarse-grained simplification based on the commonly observed empirical pattern that deeper layers tend to exhibit greater redundancy than shallower layers [7]. While this assumption may not hold strictly at every individual layer, it represents a practical approximation that enables efficient pruning optimization.
>
> 7. Gromov, Andrey, et al. "The unreasonable ineffectiveness of the deeper layers." arXiv preprint arXiv:2403.17887 (2024).
>
>
> In the following sections, we conduct two experiments to verify: 1) the assumption that deeper operations within the same module-group type are more redundant largely holds; and 2) despite this assumption not being strictly precise (i.e., being a coarse approximation), it does not prevent DOP from achieving near-optimal policies. We also updated those experiments in the **Section F.7 of the revised paper**.
>
> **Experiment 1: Pruning Consecutive Operations at Different Depths**
>
> We prune consecutive operations of the same module and group type at various layer positions to directly assess operation redundancy at different depths through their impact on downstream task performance. We first prune a single MHA/MLP operation for visual tokens on LLaVA-1.5-7B:
>
> **Table: Performance of Pruning Single Operation on LLaVA-1.5-7B**
>
> | Operation Pruned | Dataset | $l$=1 | $l$=4 | $l$=8 | $l$=12 | $l$=16 | $l$=20 | $l$=24 | $l$=28 | $l$=32 |
> |--------------------|---------|-------|-------|-------|--------|--------|--------|--------|--------|--------|
> | $\text{(}g_v,l,\text{MHA}\text{)}$ | TextVQA | 55.8 | 57.2 | 57.2 | 56.6 | 58.1 | 57.9 | 57.9 | 58.1 | 58.1 |
> |                                      | SeedBench | 26.1 | 66.2 | 65.4 | 64.3 | 66.1 | 66.2 | 66.1 | 66.2 | 66.2 |
> | $\text{(}g_v,l,\text{MLP}\text{)}$ | TextVQA | 58.3 | 58.2 | 57.9 | 58.2 | 58.0 | 58.3 | 58.1 | 58.1 | 58.3 |
> |                                      | SeedBench | 66.1 | 66.0 | 65.5 | 66.0 | 66.1 | 66.1 | 66.1 | 66.1 | 66.1 |
>
> We further prune three consecutive MHA/MLP operations for visual tokens at different positions to make the redundancy pattern more pronounced:
>
> **Table: Performance of Pruning 3 Consecutive Operations on LLaVA-1.5-7B**
>
> | Operation Pruned | Dataset | $l$=1,2,3 | $l$=4,5,6 | $l$=8,9,10 | $l$=12,13,14 | $l$=16,17,18 | $l$=20,21,22 | $l$=24,25,26 | $l$=28,29,30 |
> |--------------------|---------|------------|-----------|------------|--------------|--------------|--------------|--------------|--------------|
> | $\text{(}g_v,l,\text{MHA}\text{)}$ | TextVQA | 44.9 | 55.3 | 53.5 | 56.5 | 56.5 | 56.7 | 58.0 | 57.7 |
> |                                      | SeedBench | 60.4 | 62.9 | 59.7 | 62.2 | 66.1 | 66.2 | 66.1 | 65.8 |
> | $\text{(}g_v,l,\text{MLP}\text{)}$ | TextVQA | 58.3 | 57.3 | 55.4 | 57.6 | 57.7 | 58.2 | 58.2 | 58.4 |
> |                                      | SeedBench | 66.1 | 64.8 | 63.2 | 66.1 | 66.2 | 66.1 | 66.1 | 66.1 |
>
> We also conducted this experiment on Qwen2.5-VL-7B:
>
> **Table: MME Performance of Pruning 3 Consecutive Operations on Qwen2.5-VL-7B**
>
> | Operation Pruned | $l$=1,2,3 | $l$=4,5,6 | $l$=8,9,10 | $l$=12,13,14 | $l$=16,17,18 | $l$=20,21,22 | $l$=24,25,26 |
> |--------------------|-----------|-----------|------------|--------------|--------------|--------------|--------------|
> | $\text{(}g_v,l,\text{MHA}\text{)}$ | 1656 | 2148 | 2194 | 1959 | 2213 | 2295 | 2295 |
> | $\text{(}g_v,l,\text{MLP}\text{)}$ | 1816 | 2236 | 2244 | 2242 | 2308 | 2303 | 2297 |
>
> Generally, pruning deeper layers results in smaller performance drops, indicating that the pruned operations are more redundant. The pattern of deeper operations being more redundant becomes more evident when pruning three consecutive operations. On Qwen2.5-VL-7B, the trend generally holds with some variations in MHA operations but clear patterns in MLP operations, confirming that our depth-wise assumption is largely reasonable across different model architectures.

---

> > ### Author Response · Authors · 2025-11-25
> >
> > **Experiment 2: Policy Perturbation Analysis**
> >
> > We conduct policy perturbation experiments to validate the near-optimality of our DOP constraint. Starting from our original DOP policy, we create alternative non-consecutive pruning strategies by randomly swapping $n$ operations within the same module-group type: removing $n$ retained operations and keeping $n$ originally pruned operations, thereby breaking the depth-wise constraint. We repeat this process 10 times to sample alternative policies.
> >
> > This reverse validation approach is straightforward: if our DOP policy were far from optimal, we should easily find significantly better policies through random perturbations. Conversely, if DOP consistently outperforms the majority of these random alternatives, this provides strong evidence for its near-optimality and validates our depth-wise constraint as a fundamental efficiency principle rather than an arbitrary design choice.
> >
> > **Results on LLaVA-1.5-7B at 28% TFLOPs:**
> >
> > | Policy | Metric | TextVQA | GQA | SeedBench |
> > |--------|--------|---------|-----|-----------|
> > | Original DOP | - | 55.8 | 59.6 | 63.6 |
> > | Swap 1 MHA OP | Best | 55.8 | 59.8 | 63.6 |
> > |  | Mean | 54.3 | 57.7 | 61.4 |
> > | Swap 3 MHA OP | Best | 55.1 | 58.9 | 62.9 |
> > |  | Mean | 51.8 | 54.7 | 59.5 |
> > | Swap 1 MLP OP | Best | 55.9 | 59.7 | 63.6 |
> > |  | Mean | 55.3 | 58.8 | 63.0 |
> > | Swap 3 MLP OP | Best | 55.4 | 59.3 | 63.2 |
> > |  | Mean | 52.7 | 56.1 | 60.6 |
> >
> > For swapping 1 operation, while we can occasionally find slightly better policies, our original DOP policy significantly outperforms the average performance of perturbed policies. When we increase the perturbation magnitude by swapping 3 operations, we can hardly find any policy that outperforms DOP, and the average performance of perturbed policies deteriorates further from the original DOP baseline.
> >
> > **Results on Qwen2.5-VL-7B at 26% TFLOPs:**
> >
> > Since our original DOP policy does not prune any MHA operations ($d_A = 28$), we only performed perturbations on MLP operations:
> >
> > | Policy | Metric | TextVQA | ChartQA | MME |
> > |--------|--------|---------|---------|-----|
> > | Original DOP | - | 79.8 | 83.1 | 2351 |
> > | Swap 1 MLP OP | Best | 80.2 | 82.6 | 2347 |
> > |  | Mean | 77.5 | 81.6 | 2331 |
> > | Swap 3 MLP OP | Best | 77.1 | 81.4 | 2316 |
> > |  | Mean | 73.8 | 78.5 | 2236 |
> >
> > The results demonstrate that perturbed policies consistently fail to surpass our original DOP policy. Even in the best case, swapping 1 MLP operation only marginally exceeds performance on TextVQA (80.2 vs 79.8), while underperforming on other metrics. When increasing the perturbation magnitude (swap 3 operations), performance degrades significantly, further confirming the near-optimality of our DOP policy.
> >
> > These consistent results across different models provide strong evidence that our depth-wise constraint indeed captures fundamental principles of MLLMs, rather than being an arbitrary design choice.

---

> ### Author Response · Authors · 2025-11-25
>
> >how large is the gap between the strategies searched using the approximated divergence versus the true joint divergence
>
> **Experiment: Additive Approximation vs. Direct Joint Divergence Optimization  (Section F.4 in the revised paper)**
>
> We greatly appreciate your constructive suggestion. We conducted this experiment within a policy subspace through grid search based on direct joint divergence evaluation. We set step sizes of 2 for MHA and MLP depths ($d_A, d_P \in \{0,2,4,6,...\}$) and 16 for visual token count ($n_v \in \{0,16,32,...\}$) to create a sufficiently small policy space for exhaustive joint divergence evaluation. Given different TFLOPs targets, we first solved Equation 7 using additive approximation, then evaluated joint KL divergence for all constraint-satisfying policies on the same 100 TextVQA samples validation set, selecting the policy with minimum divergence. On LLaVA-1.5-7B with 28% and 23% target TFLOPs ratios, we found 658 and 384 valid policies respectively.
>
> | Optimization Method | $d_A$ | $d_P$ | $n_v$ | TextVQA | GQA | SeedBench |
> |---------------------|-------|-------|-------|---------|-----|-----------|
> | **28% TFLOPs** |  |  |  |  |  |  |
> | w/ Additive Approximation | 28 | 18 | 96 | 55.7 | 59.4 | 63.5 |
> | w/o Additive Approximation | 26 | 20 | 96 | 55.8 | 59.6 | 63.4 |
> | **23% TFLOPs** |  |  |  |  |  |  |
> | w/ Additive Approximation | 28 | 16 | 48 | 53.8 | 57.7 | 62.0 |
> | w/o Additive Approximation | 24 | 18 | 48 | 54.1 | 57.8 | 62.0 |
>
> The results show that direct grid search with joint KL divergence (w/o Additive Approximation) achieves slightly better performance, but policies with additive approximation achieve very similar results. This further validates our additive approximation approach and confirms that despite being a coarse approximation, it enables DOP to achieve near-optimal policies with significantly reduced computational cost.

---

### Official Review · Reviewer_aoQh · 2025-10-26

**Soundness:** 3
**Presentation:** 3
**Contribution:** 3
**Rating:** 4
**Confidence:** 4

**Summary:**

This paper proposes Depth-wise Operation Pruning (DOP), a fine-grained token allocation framework for accelerating Multimodal Large Language Models (MLLMs). DOP defines an "operation" as the computation of a specific module processing a group of tokens and enables different modules to process varying numbers of visual tokens, moving beyond uniform token reduction. It is a data-driven method that optimizes a pruning policy by minimizing the KL divergence from the original model's output distribution under a computational budget. To ensure efficiency, DOP employs depth-wise pruning constraints and an additive approximation to drastically reduce the policy search space and validation cost. Experiments show that DOP outperforms existing token pruning baselines across multiple MLLMs and benchmarks.

**Strengths:**

The paper tackles the critical problem of MLLM inference efficiency with a conceptually clear and technically sound approach. The shift from "token-level" to "operation-level" pruning is a well-motivated extension of the existing paradigm, offering finer-grained control with clear theoretical and practical value. The optimization strategy—combining depth-wise constraints with additive approximation—is cleverly designed to balance performance and efficiency, making DOP feasible for real-world deployment. The evaluation is comprehensive, covering 6 prominent MLLMs and 13 benchmarks, and includes real GPU latency measurements, lending strong credibility to the results.

**Weaknesses:**

1. The paper fails to adequately discuss or compare against recent works that also focus on "operation pruning," notably GSOP [1], Short-LVLM [2], and Skip-Vision [3]. These works similarly aim to accelerate models by pruning internal operations rather than just tokens, yet they are omitted from both the Related Work and experiments. This omission weakens the paper's novelty claim. The authors should clearly articulate the core methodological differences between DOP and GSOP (e.g., DOP's depth-wise policy vs. GSOP's greedy sorting) and include direct experimental comparisons.

2. The potential of DOP is not sufficiently explored, as follows:

- (a) All experiments are confined to static image VQA tasks, with no evaluation on caption tasks like Nocaps and TextCaps or video understanding tasks like VideoMME or MLVU. Given the inherent redundancy in video data and the fact that recent works (e.g., VisPruner, CDPruner) have validated their methods on video tasks, demonstrating DOP's efficacy in video domain is crucial.
- (b) DOP is only applied to the LLM decoder, ignoring the computationally expensive visual encoder (e.g., ViT), which is a major bottleneck for high-resolution image and video understanding.
- (c) The paper does not investigate DOP's applicability to pure long-context text tasks (e.g., LongBench, RULER), limiting the demonstration of its generalizability.

3. For Qwen2.5-VL, the authors only report results using input resizing as the underlying token reduction method. In contrast, for LLaVA-NeXT and InternVL3, they evaluate DOP with multiple token compression strategies (e.g., VisPruner, CDPruner). To ensure fair and comprehensive assessment, the authors should at least include one experiment on Qwen2.5-VL combining DOP with a state-of-the-art token pruning method like CDPruner.

4. The paper does not provide any visualization of how DOP dynamically skips computations across different benchmarks or models. For instance, showing which layers or modules are pruned under DOP for Qwen2.5-VL on tasks like TextVQA versus MME would greatly enhance interpretability and help readers intuitively understand the adaptive nature of the proposed strategy.

5. The depth-wise pruning constraint is central to DOP’s efficiency, yet the paper justifies it only with the empirical observation that “deeper layers are typically more redundant in LLMs”, which is insufficient. The authors should provide stronger theoretical or empirical support for why pruning all operations beyond a fixed depth per module type is a valid and near-optimal assumption. As presented in Figure 2 (b), the DOP Rules appear as an unverified design choice; without ablation against more flexible strategies (e.g., non-consecutive layer pruning), this key simplification lacks credible justification.

[1] Beyond Token Pruning: Operation Pruning in Vision-Language Models. arXiv 2025.

[2] Skip-Vision: Efficient and Scalable Acceleration of Vision-Language Models via Adaptive Token Skipping. ICCV 2025.

[3] Short-LVLM: Compressing and Accelerating Large Vision-Language Models by Pruning Redundant Layers. ACM MM 2025.

**Questions:**

1. Could the authors clarify the core methodological differences between DOP and recent works like GSOP and Skip-Vision, and please include experimental comparisons with these methods?

2. Why is DOP only applied to the LLM decoder and not to the computationally intensive visual encoder (e.g., ViT)?

3. Can the authors provide experimental results for DOP on video understanding benchmarks (e.g., using Qwen2.5-VL on VideoMME) and pure long-context text benchmarks (e.g., LongBench)?

---

> ### Author Response · Authors · 2025-11-25
>
> >The paper fails to adequately discuss or compare against recent works that also focus on "operation pruning," notably GSOP [1], Short-LVLM [2], and Skip-Vision [3]. These works similarly aim to accelerate models by pruning internal operations rather than just tokens, yet they are omitted from both the Related Work and experiments. This omission weakens the paper's novelty claim. The authors should clearly articulate the core methodological differences between DOP and GSOP (e.g., DOP's depth-wise policy vs. GSOP's greedy sorting) and include direct experimental comparisons.
>
> **1. Discussion of GSOP:**
>
> We refer the reviewer to this note copied from reviewer guidelines-
>
> *Note that arXiv is not considered a peer-reviewed venue. As such, authors are not required to compare to papers solely on arXiv: they may be excused for not knowing about papers not published in peer-reviewed conference proceedings or journals, which includes papers exclusively available on arXiv*
>
> GSOP falls under this category as it is on arXiv alone, making any comparison to it not required.  We have several advantages over this method including using a continuous pruning strategy, simpler grouping strategy, and a less complex search.  This not only results in a model that is less susceptible to the choice of validation set, with higher performance, but whose optimization procedure is at least 15x faster.  However, we emphasize that any comparison is not required.
>
> **2. Comparison with Short-LVLM:**
>
> The fundamental difference is that Short-LVLM performs layer pruning that modifies model architecture, while DOP preserves the original model structure. DOP only changes computation by pruning operations for specific tokens within modules, essentially performing only token manipulation like token pruning methods. Methodologically, Short-LVLM also does not propose a similar "operation pruning" framework. However, we acknowledge that both approaches consider token and structural redundancy jointly, validating our motivation. We will include this discussion in the revised version.
>
> **3. Comparison with Skip-Vision:**
>
> DOP differs from Skip-Vision in two fundamental aspects. First, regarding "operation pruning": While Skip-Vision implicitly achieves pruning of FFN (MLP) module operations for visual tokens, it does not extend to other modules or propose a comprehensive operation pruning framework, remaining limited to FFN modules only. Second, regarding practical deployment: Skip-Vision introduces new adaptive summary layers, requiring extensive data and computational resources for SFT retraining, which limits its usability. In contrast, DOP does not modify model structure or weights, requiring only minimal data and computational resources for policy optimization. Moreover, DOP's policies demonstrate strong cross-model generalization, making it essentially tuning-free and enabling plug-and-play compatibility with different models.

---

> ### Author Response · Authors · 2025-11-25
>
> > All experiments are confined to static image VQA tasks, with no evaluation on caption tasks like Nocaps and TextCaps or video understanding tasks like VideoMME or MLVU. Given the inherent redundancy in video data and the fact that recent works (e.g., VisPruner, CDPruner) have validated their methods on video tasks, demonstrating DOP's efficacy in video domain is crucial.
>
> > Can the authors provide experimental results for DOP on video understanding benchmarks (e.g., using Qwen2.5-VL on VideoMME) and pure long-context text benchmarks (e.g., LongBench)?
>
> **Image Captioning Tasks (Section E.6 in the revised paper):** We evaluated DOP's performance on image captioning tasks using NoCaps and TextCaps datasets. Experiments were conducted on LLaVA-1.5-7B with CDPruner for token reduction, comparing against CDPruner at equivalent TFLOPs using the lmm-eval framework. CIDEr score is reported.
>
> | TFLOPs | Method | NoCaps | TextCaps |
> |--------|--------|---------|----------|
> | 100% | Baseline | 1.043 | 0.992 |
> | 28% | CDPruner | 0.912 | 0.863 |
> |      | DOP+CDPruner | 0.948 | 0.883 |
> | 23% | CDPruner | 0.844 | 0.767 |
> |      | DOP+CDPruner | 0.881 | 0.790 |
> DOP consistently outperforms CDPruner on image captioning tasks, demonstrating its effectiveness beyond VQA scenarios.
>
> **Video Understanding Tasks (Section E.7 in the revised paper):** We followed VisPruner's video benchmark settings, applying DOP on Video-LLaVA, evaluating on TGIF-QA, MSVD-QA and MSRVTT-QA, using the first 1K samples from each benchmark and employing VisPruner as the visual token reduction method. Video-LLaVA processes 8 frames of 224-resolution video, totaling 2048 visual tokens.
>
> | Method | TGIF-QA |      | MSVD-QA |      | MSRVTT-QA |      | Average |      |
> |--------|---------|------|---------|------|-----------|------|---------|------|
> |        | Acc.    | Score| Acc.    | Score| Acc.      | Score| Acc.    | Score|
> | **Upper Bound, All 2048 Tokens (100%)** |
> | Video-LLaVA | 19.8 | 2.53 | 70.5 | 3.93 | 57.5 | 3.50 | 49.3 | 3.32 |
> | **Retain 16% TFLOPs** |
> | VisPruner | 15.9 | 2.41 | 69.3 | 3.92 | 55.6 | 3.45 | 46.9 | 3.26 |
> | DOP + VisPruner | 16.3 | 2.44 | 69.5 | 3.92 | 55.9 | 3.46 | 47.2 | 3.27 |
> | **Retain 11% TFLOPs** |
> | VisPruner | 14.1 | 2.35 | 65.4 | 3.79 | 54.1 | 3.41 | 44.5 | 3.18 |
> | DOP + VisPruner | 14.6 | 2.36 | 65.8 | 3.80 | 54.4 | 3.42 | 44.9 | 3.19 |
>
> DOP provides consistent improvements over VisPruner baseline across video understanding tasks. In the final version, we will include additional video benchmarks such as VideoMME and MLVU, along with more video models like LLaVA-Video to provide comprehensive validation of DOP's efficacy in the video domain.

---

> ### Author Response · Authors · 2025-11-25
>
> > DOP is only applied to the LLM decoder, ignoring the computationally expensive visual encoder (e.g., ViT), which is a major bottleneck for high-resolution image and video understanding.
> > Why is DOP only applied to the LLM decoder and not to the computationally intensive visual encoder (e.g., ViT)?
>
> We agree that DOP has the potential to reduce visual encoder computation. However, we focus on the LLM decoder in this work because in most MLLMs, the LLM decoder constitutes the primary computational bottleneck due to its significantly larger parameter size compared to the visual encoder (e.g., 20× larger in LLaVA). This focus is aligned with most prior MLLM token pruning works that primarily prune visual tokens for the LLM decoder [8,9,10,11]. While DOP for visual encoders is indeed a promising direction for future work, demonstrating DOP's acceleration effects on the LLM decoder already represents a substantial contribution in this paper.
>
> 8. Zhang, Qizhe, et al. "Beyond Attention or Similarity: Maximizing Conditional Diversity for Token Pruning in MLLMs." arXiv preprint arXiv:2506.10967 (2025).
>
> 9. Zhang, Qizhe, et al. "Beyond text-visual attention: Exploiting visual cues for effective token pruning in vlms." Proceedings of the IEEE/CVF International Conference on Computer Vision. 2025.
>
> 10. Chen, Liang, et al. "An image is worth 1/2 tokens after layer 2: Plug-and-play inference acceleration for large vision-language models." European Conference on Computer Vision. Cham: Springer Nature Switzerland, 2024.
>
> 11. Zhang, Yuan, et al. "Sparsevlm: Visual token sparsification for efficient vision-language model inference." arXiv preprint arXiv:2410.04417 (2024).
>
> > The paper does not investigate DOP's applicability to pure long-context text tasks (e.g., LongBench, RULER), limiting the demonstration of its generalizability.
>
> > Can the authors provide experimental results for DOP on video understanding benchmarks (e.g., using Qwen2.5-VL on VideoMME) and pure long-context text benchmarks (e.g., LongBench)?
>
> We believe DOP's design principles can be applied to pure long-context text tasks such as LongBench and RULER. However, this work focuses on MLLMs and vision-language tasks, and this contribution already has substantial impact and significance.
>
> First, MLLMs are known to use large numbers of tokens, leading to extremely high computational costs and creating a greater computational burden than pure text models in real-world applications. Second, MLLMs are increasingly becoming the dominant paradigm in AI applications, with multimodal capabilities becoming a standard feature in most models. Therefore, the computational burden introduced by the vision modality represents a universal and major challenge, and addressing this can produce broad and significant impact. Notably, representative MLLM token pruning works [8,9,10,11] have also not explored pure long-context text tasks, indicating that the MLLM-focused scope is well-established and appropriate in this research area. Adapting our framework to pure text domains would require considerable effort to explore text-specific redundancy patterns and develop appropriate operation pruning strategies, which is beyond the scope of this work.

---

> ### Author Response · Authors · 2025-11-25
>
> > For Qwen2.5-VL, the authors only report results using input resizing as the underlying token reduction method. In contrast, for LLaVA-NeXT and InternVL3, they evaluate DOP with multiple token compression strategies (e.g., VisPruner, CDPruner). To ensure fair and comprehensive assessment, the authors should at least include one experiment on Qwen2.5-VL combining DOP with a state-of-the-art token pruning method like CDPruner.
>
> We did not combine DOP with CDPruner on Qwen2.5-VL in our main results because CDPruner's authors have not released official implementation for Qwen2.5-VL, and our reproduction fails to match their reported performance. According to GitHub issues, many users face similar reproduction challenges due to missing implementation details. We therefore used resizing as the token reduction baseline and compared against CDPruner's reported performance. Notably, resizing already outperforms CDPruner's reported results, yet DOP still achieves significant improvements.
>
> Below we provide results with our reproduced CDPruner, showing DOP consistently delivers stable improvements over our reproduced CDPruner on Qwen2.5-VL. (Section E.8 in the revised paper)
>
> | Method | TextVQA | ChartQA | AI2D | OCRBench | MME | MMB-EN | MMB-CN | Rel. Avg. |
> |--------|---------|---------|------|----------|-----|--------|--------|-----------|
> | **1296 (Full)** | 85.5 | 86.5 | 81 | 879 | 2286 | 80.2 | 82.7 | **100%** |
> | | | **26% TFLOPs** | | | | | |
> | CDPruner (paper) | 82.4 | 73 | 77.5 | 749 | 2245 | 80.9 | 79.9 | 93.9% |
> | CDPruner (reproduce) | 76.3 | 70.4 | 76.3 | 705 | 2234 | 79.8 | 80.4 | 91.4% |
> | CDPruner (reproduce)+DOP | 78.2 | 72.1 | 77.5 | 723 | 2232 | 79.7 | 80.3 | 92.4% |
> | | | **16% TFLOPs** | | | | | |
> | CDPruner (paper) | 77.8 | 59.2 | 74 | 632 | 2127 | 76.2 | 76.5 | 86.1% |
> | CDPruner (reproduce) | 70.5 | 52.4 | 72.6 | 562 | 2187 | 77.4 | 78.1 | 83.3% |
> | CDPruner (reproduce)+DOP | 73.2 | 57.8 | 73.2 | 583 | 2203 | 77.6 | 78.1 | 85.2% |
>
> >The paper does not provide any visualization of how DOP dynamically skips computations across different benchmarks or models. For instance, showing which layers or modules are pruned under DOP for Qwen2.5-VL on tasks like TextVQA versus MME would greatly enhance interpretability and help readers intuitively understand the adaptive nature of the proposed strategy.
>
> We detailed the policies in Tables 15 and 16 in the appendix, showing optimized policies for all evaluated models on the same TextVQA subset under different TFLOPs targets. Additionally, we analyze policy differences when optimizing on different tasks, using LLaVA-1.5-7B optimized on TextVQA, VQAv2, and SeedBench as examples. (Section E.9 in the revised paper)
>
> | TFLOPs | 37% | | | 28% | | | 23% | | |
> |--------|-----|-----|-----|-----|-----|-----|-----|-----|-----|
> | Optimization Task | TextVQA | VQAv2 | SeedBench | TextVQA | VQAv2 | SeedBench | TextVQA | VQAv2 | SeedBench |
> | $d_A$ | 25 | 25 | 16 | 26 | 25 | 12 | 25 | 25 | 12 |
> | $d_P$ | 19 | 18 | 12 | 19 | 19 | 12 | 16 | 18 | 12 |
> | $n_v$ | 192 | 197 | 293 | 97 | 98 | 167 | 51 | 48 | 78 |
>
> As shown in the table, policies optimized on TextVQA and VQAv2 are largely similar, while SeedBench exhibits significantly different behavior—more aggressively pruning operations while retaining more tokens. These policy differences explain the performance variations in Table 4: Cross-task Generalization Ablation. In Appendix F.2, Figure 4 demonstrates through divergence analysis that SeedBench exhibits significantly higher redundancy in deeper layer operations compared to other benchmarks, leading to more aggressive pruning of deep operations and retention of more tokens for important shallow operations.

---

> ### Author Response · Authors · 2025-11-25
>
> >The depth-wise pruning constraint is central to DOP’s efficiency, yet the paper justifies it only with the empirical observation that “deeper layers are typically more redundant in LLMs”, which is insufficient. The authors should provide stronger theoretical or empirical support for why pruning all operations beyond a fixed depth per module type is a valid and near-optimal assumption. As presented in Figure 2 (b), the DOP Rules appear as an unverified design choice; without ablation against more flexible strategies (e.g., non-consecutive layer pruning), this key simplification lacks credible justification.
>
> Our depth-wise assumption is a coarse-grained simplification based on the commonly observed empirical pattern that deeper layers tend to exhibit greater redundancy than shallower layers [7]. While this assumption may not hold strictly at every individual layer, it represents a practical approximation that enables efficient pruning optimization.
>
> 7. Gromov, Andrey, et al. "The unreasonable ineffectiveness of the deeper layers." arXiv preprint arXiv:2403.17887 (2024).
>
>
> In the following sections, we conduct two experiments to verify: 1) the assumption that deeper operations within the same module-group type are more redundant largely holds; and 2) despite this assumption not being strictly precise (i.e., being a coarse approximation), it does not prevent DOP from achieving near-optimal policies. We also updated those experiments in the **Section F.7 of the revised paper**.
>
> **Experiment 1: Pruning Consecutive Operations at Different Depths**
>
> We prune consecutive operations of the same module and group type at various layer positions to directly assess operation redundancy at different depths through their impact on downstream task performance. We first prune a single MHA/MLP operation for visual tokens on LLaVA-1.5-7B:
>
> **Table: Performance of Pruning Single Operation on LLaVA-1.5-7B**
>
> | Operation Pruned | Dataset | $l$=1 | $l$=4 | $l$=8 | $l$=12 | $l$=16 | $l$=20 | $l$=24 | $l$=28 | $l$=32 |
> |--------------------|---------|-------|-------|-------|--------|--------|--------|--------|--------|--------|
> | $\text{(}g_v,l,\text{MHA}\text{)}$ | TextVQA | 55.8 | 57.2 | 57.2 | 56.6 | 58.1 | 57.9 | 57.9 | 58.1 | 58.1 |
> |                                      | SeedBench | 26.1 | 66.2 | 65.4 | 64.3 | 66.1 | 66.2 | 66.1 | 66.2 | 66.2 |
> | $\text{(}g_v,l,\text{MLP}\text{)}$ | TextVQA | 58.3 | 58.2 | 57.9 | 58.2 | 58.0 | 58.3 | 58.1 | 58.1 | 58.3 |
> |                                      | SeedBench | 66.1 | 66.0 | 65.5 | 66.0 | 66.1 | 66.1 | 66.1 | 66.1 | 66.1 |
>
> We further prune three consecutive MHA/MLP operations for visual tokens at different positions to make the redundancy pattern more pronounced:
>
> **Table: Performance of Pruning 3 Consecutive Operations on LLaVA-1.5-7B**
>
> | Operation Pruned | Dataset | $l$=1,2,3 | $l$=4,5,6 | $l$=8,9,10 | $l$=12,13,14 | $l$=16,17,18 | $l$=20,21,22 | $l$=24,25,26 | $l$=28,29,30 |
> |--------------------|---------|------------|-----------|------------|--------------|--------------|--------------|--------------|--------------|
> | $\text{(}g_v,l,\text{MHA}\text{)}$ | TextVQA | 44.9 | 55.3 | 53.5 | 56.5 | 56.5 | 56.7 | 58.0 | 57.7 |
> |                                      | SeedBench | 60.4 | 62.9 | 59.7 | 62.2 | 66.1 | 66.2 | 66.1 | 65.8 |
> | $\text{(}g_v,l,\text{MLP}\text{)}$ | TextVQA | 58.3 | 57.3 | 55.4 | 57.6 | 57.7 | 58.2 | 58.2 | 58.4 |
> |                                      | SeedBench | 66.1 | 64.8 | 63.2 | 66.1 | 66.2 | 66.1 | 66.1 | 66.1 |
>
> We also conducted this experiment on Qwen2.5-VL-7B:
>
> **Table: MME Performance of Pruning 3 Consecutive Operations on Qwen2.5-VL-7B**
>
> | Operation Pruned | $l$=1,2,3 | $l$=4,5,6 | $l$=8,9,10 | $l$=12,13,14 | $l$=16,17,18 | $l$=20,21,22 | $l$=24,25,26 |
> |--------------------|-----------|-----------|------------|--------------|--------------|--------------|--------------|
> | $\text{(}g_v,l,\text{MHA}\text{)}$ | 1656 | 2148 | 2194 | 1959 | 2213 | 2295 | 2295 |
> | $\text{(}g_v,l,\text{MLP}\text{)}$ | 1816 | 2236 | 2244 | 2242 | 2308 | 2303 | 2297 |
>
> Generally, pruning deeper layers results in smaller performance drops, indicating that the pruned operations are more redundant. The pattern of deeper operations being more redundant becomes more evident when pruning three consecutive operations. On Qwen2.5-VL-7B, the trend generally holds with some variations in MHA operations but clear patterns in MLP operations, confirming that our depth-wise assumption is largely reasonable across different model architectures.

---

> ### Author Response · Authors · 2025-11-25
>
> **Experiment 2: Policy Perturbation Analysis**
>
> We conduct policy perturbation experiments to validate the near-optimality of our DOP constraint. Starting from our original DOP policy, we create alternative non-consecutive pruning strategies by randomly swapping $n$ operations within the same module-group type: removing $n$ retained operations and keeping $n$ originally pruned operations, thereby breaking the depth-wise constraint. We repeat this process 10 times to sample alternative policies.
>
> This reverse validation approach is straightforward: if our DOP policy were far from optimal, we should easily find significantly better policies through random perturbations. Conversely, if DOP consistently outperforms the majority of these random alternatives, this provides strong evidence for its near-optimality and validates our depth-wise constraint as a fundamental efficiency principle rather than an arbitrary design choice.
>
> **Results on LLaVA-1.5-7B at 28% TFLOPs:**
>
> | Policy | Metric | TextVQA | GQA | SeedBench |
> |--------|--------|---------|-----|-----------|
> | Original DOP | - | 55.8 | 59.6 | 63.6 |
> | Swap 1 MHA OP | Best | 55.8 | 59.8 | 63.6 |
> |  | Mean | 54.3 | 57.7 | 61.4 |
> | Swap 3 MHA OP | Best | 55.1 | 58.9 | 62.9 |
> |  | Mean | 51.8 | 54.7 | 59.5 |
> | Swap 1 MLP OP | Best | 55.9 | 59.7 | 63.6 |
> |  | Mean | 55.3 | 58.8 | 63.0 |
> | Swap 3 MLP OP | Best | 55.4 | 59.3 | 63.2 |
> |  | Mean | 52.7 | 56.1 | 60.6 |
>
> For swapping 1 operation, while we can occasionally find slightly better policies, our original DOP policy significantly outperforms the average performance of perturbed policies. When we increase the perturbation magnitude by swapping 3 operations, we can hardly find any policy that outperforms DOP, and the average performance of perturbed policies deteriorates further from the original DOP baseline.
>
> **Results on Qwen2.5-VL-7B at 26% TFLOPs:**
>
> Since our original DOP policy does not prune any MHA operations ($d_A = 28$), we only performed perturbations on MLP operations:
>
> | Policy | Metric | TextVQA | ChartQA | MME |
> |--------|--------|---------|---------|-----|
> | Original DOP | - | 79.8 | 83.1 | 2351 |
> | Swap 1 MLP OP | Best | 80.2 | 82.6 | 2347 |
> |  | Mean | 77.5 | 81.6 | 2331 |
> | Swap 3 MLP OP | Best | 77.1 | 81.4 | 2316 |
> |  | Mean | 73.8 | 78.5 | 2236 |
>
> The results demonstrate that perturbed policies consistently fail to surpass our original DOP policy. Even in the best case, swapping 1 MLP operation only marginally exceeds performance on TextVQA (80.2 vs 79.8), while underperforming on other metrics. When increasing the perturbation magnitude (swap 3 operations), performance degrades significantly, further confirming the near-optimality of our DOP policy.
>
> These consistent results across different models provide strong evidence that our depth-wise constraint indeed captures fundamental principles of MLLMs, rather than being an arbitrary design choice.

---

### Official Review · Reviewer_DwzN · 2025-10-31

**Soundness:** 3
**Presentation:** 3
**Contribution:** 3
**Rating:** 6
**Confidence:** 3

**Summary:**

This paper presents the Depth-wise Operation Pruning (DOP) framework, designed to improve the inference efficiency of Multimodal Large Language Models (MLLMs). DOP breaks the computation process into small “operations” and focuses on reducing structural redundancy that traditional token-pruning methods often miss. By applying MHA/MLP depth decoupling and a depth-wise continuous pruning strategy, it turns a difficult policy search problem into a much simpler optimization task with only three parameters: $(\mathbf{D_A}, \mathbf{D_P}, \mathbf{n_v})$. Experiments on many MLLM models show that DOP delivers strong performance and clear efficiency gains.

**Strengths:**

1. Through depth-wise pruning constraints and additive approximation, the complex optimization problem becomes highly efficient to solve.
2. The experimental validation is strong. The method is tested on diverse MLLMs (LLaVA, Qwen, InternVL) and 13 benchmarks, with comparisons against recent baselines. Using fixed TFLOPs budgets ensures fair evaluation.

**Weaknesses:**

1.	The method relies on two strong simplifying assumptions—monotonic depth redundancy and additive independence across $\mathbf{D_A}$, $\mathbf{D_P}$, and $\mathbf{n_v}$—which may not always hold, as deeper layers are not necessarily more redundant for all tasks. This can limit the method’s ability to reach the global optimum and lead to suboptimal strategies in certain scenarios.
2.	The performance of DPO is highly sensitive to the validation set used for optimization, and strategies optimized on one task or data distribution may not generalize well to others.

**Questions:**

1.	Please see the weakness above.
2.	The evaluation benchmarks mainly focus on short-answer VQA tasks. How might DOP perform on multimodal tasks that require either short- or long-form generation, such as detailed image analysis or story generation? Could aggressively pruning visual operations in deeper layers prematurely remove the visual grounding necessary to maintain factual consistency throughout a longer narrative?

---

> ### Author Response · Authors · 2025-11-25
>
> >The method relies on two strong simplifying assumptions—monotonic depth redundancy and additive independence across $d_A,d_P,n_v$ —which may not always hold, as deeper layers are not necessarily more redundant for all tasks. This can limit the method’s ability to reach the global optimum and lead to suboptimal strategies in certain scenarios.
>
> Our depth-wise assumption is a coarse-grained simplification based on the commonly observed empirical pattern that deeper layers tend to exhibit greater redundancy than shallower layers [7]. While this assumption may not hold strictly at every individual layer, it represents a practical approximation that enables efficient pruning optimization.
>
> 7. Gromov, Andrey, et al. "The unreasonable ineffectiveness of the deeper layers." arXiv preprint arXiv:2403.17887 (2024).
>
> In the following sections, we conduct two experiments to verify: 1) the assumption that deeper operations within the same module-group type are more redundant largely holds; and 2) despite this assumption not being strictly precise (i.e., being a coarse approximation), it does not prevent DOP from achieving near-optimal policies. We also updated those experiments in the **Section F.7 of the revised paper**.
>
> **Experiment 1: Pruning Consecutive Operations at Different Depths**
>
> We prune consecutive operations of the same module and group type at various layer positions to directly assess operation redundancy at different depths through their impact on downstream task performance. We first prune a single MHA/MLP operation for visual tokens on LLaVA-1.5-7B:
>
> **Table: Performance of Pruning Single Operation on LLaVA-1.5-7B**
>
> | Operation Pruned | Dataset | $l$=1 | $l$=4 | $l$=8 | $l$=12 | $l$=16 | $l$=20 | $l$=24 | $l$=28 | $l$=32 |
> |--------------------|---------|-------|-------|-------|--------|--------|--------|--------|--------|--------|
> | $\text{(}g_v,l,\text{MHA}\text{)}$ | TextVQA | 55.8 | 57.2 | 57.2 | 56.6 | 58.1 | 57.9 | 57.9 | 58.1 | 58.1 |
> |                                      | SeedBench | 26.1 | 66.2 | 65.4 | 64.3 | 66.1 | 66.2 | 66.1 | 66.2 | 66.2 |
> | $\text{(}g_v,l,\text{MLP}\text{)}$ | TextVQA | 58.3 | 58.2 | 57.9 | 58.2 | 58.0 | 58.3 | 58.1 | 58.1 | 58.3 |
> |                                      | SeedBench | 66.1 | 66.0 | 65.5 | 66.0 | 66.1 | 66.1 | 66.1 | 66.1 | 66.1 |
>
> We further prune three consecutive MHA/MLP operations for visual tokens at different positions to make the redundancy pattern more pronounced:
>
> **Table: Performance of Pruning 3 Consecutive Operations on LLaVA-1.5-7B**
>
> | Operation Pruned | Dataset | $l$=1,2,3 | $l$=4,5,6 | $l$=8,9,10 | $l$=12,13,14 | $l$=16,17,18 | $l$=20,21,22 | $l$=24,25,26 | $l$=28,29,30 |
> |--------------------|---------|------------|-----------|------------|--------------|--------------|--------------|--------------|--------------|
> | $\text{(}g_v,l,\text{MHA}\text{)}$ | TextVQA | 44.9 | 55.3 | 53.5 | 56.5 | 56.5 | 56.7 | 58.0 | 57.7 |
> |                                      | SeedBench | 60.4 | 62.9 | 59.7 | 62.2 | 66.1 | 66.2 | 66.1 | 65.8 |
> | $\text{(}g_v,l,\text{MLP}\text{)}$ | TextVQA | 58.3 | 57.3 | 55.4 | 57.6 | 57.7 | 58.2 | 58.2 | 58.4 |
> |                                      | SeedBench | 66.1 | 64.8 | 63.2 | 66.1 | 66.2 | 66.1 | 66.1 | 66.1 |
>
> We also conducted this experiment on Qwen2.5-VL-7B:
>
> **Table: MME Performance of Pruning 3 Consecutive Operations on Qwen2.5-VL-7B**
>
> | Operation Pruned | $l$=1,2,3 | $l$=4,5,6 | $l$=8,9,10 | $l$=12,13,14 | $l$=16,17,18 | $l$=20,21,22 | $l$=24,25,26 |
> |--------------------|-----------|-----------|------------|--------------|--------------|--------------|--------------|
> | $\text{(}g_v,l,\text{MHA}\text{)}$ | 1656 | 2148 | 2194 | 1959 | 2213 | 2295 | 2295 |
> | $\text{(}g_v,l,\text{MLP}\text{)}$ | 1816 | 2236 | 2244 | 2242 | 2308 | 2303 | 2297 |
>
> Generally, pruning deeper layers results in smaller performance drops, indicating that the pruned operations are more redundant. The pattern of deeper operations being more redundant becomes more evident when pruning three consecutive operations. On Qwen2.5-VL-7B, the trend generally holds with some variations in MHA operations but clear patterns in MLP operations, confirming that our depth-wise assumption is largely reasonable across different model architectures.

---

> > ### Author Response · Authors · 2025-11-25
> >
> > **Experiment 2: Policy Perturbation Analysis**
> >
> > We conduct policy perturbation experiments to validate the near-optimality of our DOP constraint. Starting from our original DOP policy, we create alternative non-consecutive pruning strategies by randomly swapping $n$ operations within the same module-group type: removing $n$ retained operations and keeping $n$ originally pruned operations, thereby breaking the depth-wise constraint. We repeat this process 10 times to sample alternative policies.
> >
> > This reverse validation approach is straightforward: if our DOP policy were far from optimal, we should easily find significantly better policies through random perturbations. Conversely, if DOP consistently outperforms the majority of these random alternatives, this provides strong evidence for its near-optimality and validates our depth-wise constraint as a fundamental efficiency principle rather than an arbitrary design choice.
> >
> > **Results on LLaVA-1.5-7B at 28% TFLOPs:**
> >
> > | Policy | Metric | TextVQA | GQA | SeedBench |
> > |--------|--------|---------|-----|-----------|
> > | Original DOP | - | 55.8 | 59.6 | 63.6 |
> > | Swap 1 MHA OP | Best | 55.8 | 59.8 | 63.6 |
> > |  | Mean | 54.3 | 57.7 | 61.4 |
> > | Swap 3 MHA OP | Best | 55.1 | 58.9 | 62.9 |
> > |  | Mean | 51.8 | 54.7 | 59.5 |
> > | Swap 1 MLP OP | Best | 55.9 | 59.7 | 63.6 |
> > |  | Mean | 55.3 | 58.8 | 63.0 |
> > | Swap 3 MLP OP | Best | 55.4 | 59.3 | 63.2 |
> > |  | Mean | 52.7 | 56.1 | 60.6 |
> >
> > For swapping 1 operation, while we can occasionally find slightly better policies, our original DOP policy significantly outperforms the average performance of perturbed policies. When we increase the perturbation magnitude by swapping 3 operations, we can hardly find any policy that outperforms DOP, and the average performance of perturbed policies deteriorates further from the original DOP baseline.
> >
> > **Results on Qwen2.5-VL-7B at 26% TFLOPs:**
> >
> > Since our original DOP policy does not prune any MHA operations ($d_A = 28$), we only performed perturbations on MLP operations:
> >
> > | Policy | Metric | TextVQA | ChartQA | MME |
> > |--------|--------|---------|---------|-----|
> > | Original DOP | - | 79.8 | 83.1 | 2351 |
> > | Swap 1 MLP OP | Best | 80.2 | 82.6 | 2347 |
> > |  | Mean | 77.5 | 81.6 | 2331 |
> > | Swap 3 MLP OP | Best | 77.1 | 81.4 | 2316 |
> > |  | Mean | 73.8 | 78.5 | 2236 |
> >
> > The results demonstrate that perturbed policies consistently fail to surpass our original DOP policy. Even in the best case, swapping 1 MLP operation only marginally exceeds performance on TextVQA (80.2 vs 79.8), while underperforming on other metrics. When increasing the perturbation magnitude (swap 3 operations), performance degrades significantly, further confirming the near-optimality of our DOP policy.
> >
> > These consistent results across different models provide strong evidence that our depth-wise constraint indeed captures fundamental principles of MLLMs, rather than being an arbitrary design choice.

---

> ### Author Response · Authors · 2025-11-25
>
> >The performance of DOP is highly sensitive to the validation set used for optimization, and strategies optimized on one task or data distribution may not generalize well to others.
>
> While our ablation study in Section 5 shows that DOP can produce non-generalizable policies on tasks with severely imbalanced depth-wise redundancy distributions (e.g., SeedBench), we have identified both the cause and solution. Tasks like SeedBench are exceptional cases - optimizing on tasks with more balanced redundancy distributions yields policies that generalize well across benchmarks.
>
> **Table 4 Excerpt - Cross-task Generalization:**
>
> | Validation Set | VQAv2 | GQA | TextVQA | SeedBench | Rel. Avg. |
> |----------------|-------|-----|---------|-----------|-----------|
> | TextVQA        | **76.2** | **59.6** | **55.8** | 63.6 | **97.9%** |
> | VQAv2          | 76.0 | 59.5 | **55.8** | 63.5 | 97.3% |
> | SeedBench      | 60.5 | 50.3 | 50.5 | **64.2** | 91.8% |
>
> In Table 4, policies optimized on TextVQA and VQAv2 consistently generalize to all other benchmarks, including SeedBench, demonstrating DOP's strong generalization capability.
>
> **Table 17 Excerpt - Optimization Efficiency:**
>
> | #Samples | 25 | 50 | 100 | 200 | 500 | CDPruner |
> |----------|----|----|-----|-----|-----|----------|
> | Rel. Avg.| 97.7% | 97.3% | 97.9% | 97.6% | 97.5% | 96.7% |
> | Cost (min)| 2 | 3 | 5 | 10 | 24 | - |
>
> Table 17 further shows DOP's robustness to validation set size, requiring only 25+ samples for effective optimization and significantly outperforming CDPruner across all sample sizes.

---

> ### Author Response · Authors · 2025-11-25
>
> >The evaluation benchmarks mainly focus on short-answer VQA tasks. How might DOP perform on multimodal tasks that require either short- or long-form generation, such as detailed image analysis or story generation? Could aggressively pruning visual operations in deeper layers prematurely remove the visual grounding necessary to maintain factual consistency throughout a longer narrative?
>
> We evaluated DOP's performance on image captioning tasks, which involve long-form generation, using NoCaps and TextCaps datasets (**Section E.6 in the revised paper**). Experiments were conducted on LLaVA-1.5-7B with CDPruner for token reduction, comparing against CDPruner at equivalent TFLOPs using the lmm-eval framework. CIDEr score is reported.
>
> | TFLOPs | Method | NoCaps | TextCaps |
> |--------|--------|---------|----------|
> | 100% | Baseline | 1.043 | 0.992 |
> | 28% | CDPruner | 0.912 | 0.863 |
> |      | DOP$_\text{C}$ | 0.948 | 0.883 |
> | 23% | CDPruner | 0.844 | 0.767 |
> |      | DOP$_\text{C}$ | 0.881 | 0.790 |
>
> DOP consistently outperforms CDPruner on image captioning tasks, further demonstrating its generalization capability. We will include additional long-form generation tasks such as detailed image analysis and story generation with comprehensive analysis of visual grounding abilities in the final version.

---

### Official Review · Reviewer_GM9q · 2025-11-01

**Soundness:** 2
**Presentation:** 3
**Contribution:** 2
**Rating:** 4
**Confidence:** 3

**Summary:**

The paper proposes **Depth-wise Operation Pruning (DOP)**, an operation-level pruning strategy for MLLMs that claims finer control than token pruning. The method prunes redundant operations (defined as module–token computations) using two heuristics: (1) depth-wise layer pruning and (2) additive divergence approximation for efficient search. Experiments on six MLLMs across 13 benchmarks show large FLOPs savings with minimal performance loss.

**Strengths:**

* Addresses an important and timely efficiency problem in multimodal LLMs.

* The method is simple, easy to integrate, and empirically effective across different architectures.

* Extensive experimental coverage with clear ablations and latency analysis.

**Weaknesses:**

* **Incremental novelty:** The idea of pruning at finer granularity is not fundamentally new; the contribution is mostly heuristic refinements (depth ordering + additive scoring).

* **Limited theoretical insight:** The additive approximation is empirical, with no formal justification or guarantees; the correlation analysis is weak evidence for correctness.

* **Heavy reliance on validation heuristics:** The optimization objective (KL divergence on a small validation set) may not correlate well with downstream task performance; no robustness or sensitivity analysis is provided.

 * **Selective scope:** Despite framing as general “operation pruning,” the paper only applies it to visual tokens; this weakens the generality claim.

* **Unclear comparison fairness:** Some baselines (e.g., progressive pruning) are not re-tuned under identical FLOPs constraints. It’s unclear whether hyperparameters were optimized equally.

 * **Overemphasis on numbers:** The main results show marginal performance differences compared to strong baselines like CDPruner; the claimed 7\% gain seems dataset-specific.

 * **Ablations lack depth:** No analysis of failure cases, gradient sensitivity, or interplay between MHA vs. MLP pruning depth.

**Questions:**

The work would benefit from a deeper analysis of why DOP works and under what conditions it fails.

There can be an analysis of failure cases, gradient sensitivity, or interplay between MHA vs. MLP pruning depth.

---

> ### Author Response · Authors · 2025-11-25
>
> >The idea of pruning at finer granularity is not fundamentally new; the contribution is mostly heuristic refinements (depth ordering + additive scoring).
>
> We respectfully disagree that our contribution is merely heuristic refinements. While pruning at finer granularity is a well-motivated direction, this validates our approach rather than diminishing its novelty. The key contribution lies in how we achieve fine-grained pruning for MLLMs with low resources. Existing approaches either cannot be directly applied to our target task of achieving MLLM acceleration with low resources, or still operate at coarser granularities.
>
> For example, the dynamic computation methods mentioned by Reviewer ZZ6f [1,2,3] require architectural modifications and extensive training data and computational resources, making them expensive to deploy. In contrast, our DOP requires as few as 25 samples and demonstrates strong cross-model generalization capabilities, enabling plug-and-play deployment across different MLLMs.
>
> Progressive token pruning approaches [4,5,6] represent closer related work but operate exclusively at the token level without considering finer operation/module-level granularity . Our DOP is the first to achieve operation-level pruning for MLLMs, and our demonstrated performance advantages validate the significance of this finer-grained contribution.
> We would welcome the reviewer's guidance on specific citations for similar pruning at finer granularity frameworks for MLLMs, which would help us provide a more complete comparison and better articulate the distinctions of our approach.
>
> [1] Wu, Qiong, et al. "Routing experts: Learning to route dynamic experts in multi-modal large language models." arXiv preprint arXiv:2407.14093 (2024).
>
> [2] Li, Xiaonan, et al. "Accelerating bert inference for sequence labeling via early-exit." Proceedings of the 59th annual meeting of the Association for Computational Linguistics and the 11th international joint conference on natural language processing (volume 1: long papers). 2021.
>
> [3] Zhao, Anhao, et al. "SkipGPT: Dynamic Layer Pruning Reinvented with Token Awareness and Module Decoupling." arXiv preprint arXiv:2506.04179 (2025).
>
> [4] Ye, Weihao, et al. "Fit and prune: Fast and training-free visual token pruning for multi-modal large language models." Proceedings of the AAAI Conference on Artificial Intelligence. Vol. 39. No. 21. 2025.
>
> [5] Zhang, Yuan, et al. "Sparsevlm: Visual token sparsification for efficient vision-language model inference." arXiv preprint arXiv:2410.04417 (2024).
>
> [6] Xing, Long, et al. "Pyramiddrop: Accelerating your large vision-language models via pyramid visual redundancy reduction." arXiv preprint arXiv:2410.17247 (2024).

---

> ### Author Response · Authors · 2025-11-25
>
> > Limited theoretical insight: The additive approximation is empirical, with no formal justification or guarantees; the correlation analysis is weak evidence for correctness.
>
> We appreciate the reviewer's comment about theoretical foundations. Our additive approximation is designed to achieve a favorable tradeoff between policy performance and optimization cost. While deeper theoretical analysis could be valuable, practical effectiveness serves as the ultimate validation. Our evaluation across 6 MLLM models and 13 multimodal benchmarks demonstrates significant advantages over 12 baselines, including state-of-the-art methods like CDPruner. This extensive and broad experimental validation provides sufficient evidence for the effectiveness of our additive approximation.
>
> Additionally, we include a direct validation by evaluating joint KL divergence in the policy subspace through grid search for optimal policies and comparing with policies obtained via additive approximation. The minimal differences in downstream task performance demonstrate the reasonableness of our additive approximation. (**Section F.3 in the revised paper.**)
>
> **Experiment : Additive Approximation vs. Direct Joint Divergence Optimization**
>
> We conducted a grid search experiment in a policy subspace based on direct joint divergence evaluation. We create a sufficiently small policy space that allows us to exhaustively evaluate the joint divergence of each policy by sampling MHA and MLP depths every 2 steps ($d_A, d_P \in$ {0,2,4,6,...}) and visual token counts every 16 steps ($n_v \in $ {0,16,32,...}). Given different TFLOPs targets, we first solved Equation 7 using additive approximation, then evaluated joint KL divergence for all constraint-satisfying policies on the same validation set, selecting the policy with minimum divergence. On LLaVA-1.5-7B with 28% and 23% target TFLOPs ratios, we found 658 and 384 valid policies respectively.
>
> | Optimization Method | $d_A$ | $d_P$ | $n_v$ | TextVQA | GQA | SeedBench |
> |---------------------|-----|-----|-----|---------|-----|-----------|
> | **28% TFLOPs** |  |  |  |  |  |  |
> | w/ Additive Approximation | 28 | 18 | 96 | 55.7 | 59.4 | 63.5 |
> | w/o Additive Approximation | 26 | 20 | 96 | 55.8 | 59.6 | 63.4 |
> | **23% TFLOPs** |  |  |  |  |  |  |
> | w/ Additive Approximation | 28 | 16 | 48 | 53.8 | 57.7 | 62.0 |
> | w/o Additive Approximation | 24 | 18 | 48 | 54.1 | 57.8 | 62.0 |
>
> The results show that direct grid search with joint KL divergence (w/o Additive Approximation) achieves slightly better performance, but policies with additive approximation achieve very similar results. This further validates our additive approximation approach and confirms that despite being a coarse approximation, it enables DOP to achieve near-optimal policies with significantly reduced computational cost.

---

> ### Author Response · Authors · 2025-11-25
>
> > Selective scope: Despite framing as general “operation pruning,” the paper only applies it to visual tokens; this weakens the generality claim.
>
> We primarily apply DOP to visual tokens for fair comparison with token pruning baselines that only prune visual tokens. However, DOP can indeed be extended to other token groups. Here we demonstrate pruning operations for both system and visual tokens on LLaVA-1.5-7B (**Section F.6 in the revised paper**).
>
> Since we cannot change the number of system tokens, we prune their MHA and MLP operations in depth order by adding $d^s_A$ and $d^s_P$ (system token MHA and MLP operation depths) and their corresponding individual divergences to Equation 7, resulting in the Equation 14 in the revised paper.
>
> We maintain the same settings as in the main paper, performing optimization on LLaVA-1.5-7B with CDPruner for token reduction. The results are as follows:
>
> | TFLOPs Ratio | Group | TextVQA | GQA | SeedBench |
> |--------------|--------|---------|-----|-----------|
> | 28% | Vision | 55.8 | 59.6 | 63.6 |
> |     | Vision+System | 55.9 | 60.1 | 63.8 |
> | 23% | Vision | 54.2 | 58.1 | 62.2 |
> |     | Vision+System | 54.4 | 58.5 | 62.7 |
>
> Simultaneously pruning vision and system token operations provides consistent performance improvements over vision-only pruning, demonstrating DOP's generality.

---

> ### Author Response · Authors · 2025-11-25
>
> > Heavy reliance on validation heuristics: The optimization objective (KL divergence on a small validation set) may not correlate well with downstream task performance; no robustness or sensitivity analysis is provided.
>
> We note that Table 17 in Appendix F.1 shows ablation results for policy performance across different validation set sizes, demonstrating that DOP robustly identifies good policies regardless of validation set size. Here we provide additional robustness analysis (**Section F.4 in the revised paper**):
>
> **1. Validation Set Robustness:** Using five different random seeds to create validation sets of 100 TextVQA samples, the mean performance and standard deviations are:
>
> | TFLOPs Ratio | TextVQA | GQA | SeedBench |
> |--------------|---------|-----|-----------|
> | 28% | 55.7±0.2 | 59.5±0.1 | 63.4±0.1 |
> | 23% | 54.1±0.3 | 58.0±0.2 | 62.1±0.3 |
>
> **2. Direct Downstream Metric Optimization (Section F.5 in the revised paper):** DOP is not limited to KL divergence - it can optimize directly using downstream task metrics. We replace the divergence $D(d_A,d_P,n_v)$ in Section 3.2.2 with performance drop $D_{acc}(d_A,d_P,n_v) = Accuracy(L,L,N_v) - Accuracy(d_A,d_P,n_v)$, optimizing on 100 TextVQA validation samples:
>
> | TFLOPs Ratio | Method | Metric | TextVQA | GQA | SeedBench |
> |--------------|--------|--------|---------|-----|-----------|
> | 28% | CDPruner | - | 55.3 | 58.6 | 62.5 |
> |     | DOP$_\text{C}$ | KL Divergence | 55.8 | 59.6 | 63.6 |
> |     |       | Accuracy | 55.6 | 59.3 | 63.4 |
> | 23% | CDPruner | - | 53.2 | 57.0 | 60.9 |
> |     | DOP$_\text{C}$ | KL Divergence | 54.2 | 58.1 | 62.2 |
> |     |       | Accuracy | 53.7 | 57.6 | 61.6 |
>
> The table shows that optimizing directly on downstream metrics still outperforms CDPruner. However, KL divergence produces better policies because downstream metrics are less smooth than KL divergence - some questions remain consistently correct/incorrect regardless of pruning, while others change abruptly only after reaching certain pruning thresholds, rather than showing gradual degradation that accurately reflects model capability changes.

---

> ### Author Response · Authors · 2025-11-25
>
> > Unclear comparison fairness: Some baselines (e.g., progressive pruning) are not re-tuned under identical FLOPs constraints. It’s unclear whether hyperparameters were optimized equally.
>
> We have already addressed this concern in **lines 360-361** of the paper. For FitPrune, which requires optimization, we used the same validation set as our DOP to ensure fair comparison. For the other two Progressive Pruning baselines, they do not involve optimization components, so no hyperparameter tuning is required for fair comparison.
>
> >Overemphasis on numbers: The main results show marginal performance differences compared to strong baselines like CDPruner; the claimed 7% gain seems dataset-specific.
>
> We would like to emphasize that in LLaVA model settings, CDPruner already retains over 90% of the original model's performance. In such scenarios, preserving an additional 1% of model performance is significantly more challenging than achieving 10% improvement over a poor baseline. Our DOP improvements are not marginal - rather, the baseline performance is already quite saturated. When applied with the relatively weaker VisPruner baseline, or when evaluated on Qwen2.5-VL or InternVL3 where CDPruner performs less effectively, DOP's performance improvements become more substantial.
>
> >The claimed 7% gain seems dataset-specific.
>
> The "up to 7% advantage over CDPruner" is not dataset-specific but represents the advantage averaged across 7 diverse benchmarks on InternVL3-8B at 18% TFLOPs (**refer to Figure 3 or Table 13**).

---

> ### Author Response · Authors · 2025-11-25
>
> >Ablations lack depth: No analysis of failure cases, gradient sensitivity, or interplay between MHA vs. MLP pruning depth.
> >The work would benefit from a deeper analysis of why DOP works and under what conditions it fails.
> >There can be an analysis of failure cases, gradient sensitivity, or interplay between MHA vs. MLP pruning depth.
>
> We have identified several failure cases and provided solutions. In **Section 5 (Table 4)**, we reveal that DOP policies optimized on tasks with greater deep-layer redundancy (like SeedBench) may fail to generalize to other benchmarks, and we provide a solution: using tasks with more balanced redundancy distribution (like TextVQA) yields policies that generalize well across tasks. Additionally, in **Section F.2 (Table 17)**, we demonstrate another failure case where validation sets smaller than 25 samples lead to less effective policies.
>
> We are open to including additional analysis and discussion. Could you please provide more details about what specific aspects of gradient sensitivity or the interplay between MHA vs. MLP pruning depth should be analyzed? This would help us better address your concerns in the revision.

---

### Author Response · Authors · 2025-12-01
**Global Response**

We thank all reviewers for their thoughtful feedback and for recognizing DOP's key contributions, including the novel operation-level pruning framework that offers finer-grained control than existing token pruning methods (aoQh), the clever optimization strategy combining depth-wise constraints with additive approximation that makes real-world deployment feasible (DwzN, aoQh, ZZ6f), the comprehensive experimental validation across 6 MLLMs and 13 benchmarks demonstrating strong empirical effectiveness (GM9q,DwzN,aoQh,ZZ6f), and the practical impact of achieving significant computational savings with minimal performance loss through a simple, plug-and-play approach (GM9q,aoQh).

We have addressed the following questions in our individual responses to each reviewer:

**Reviewer GM9q:**
- Comparisons with related works
- Validation of additive approximation
- DOP that prunes more than just visual operations
- Robustness regarding validation set and policy performance metrics
- Clarifications on comparison settings and experimental results
- Analysis of failure cases

**Reviewer DwzN:**
- Stronger justification for the depth-wise pruning constraint
- Clarifications on robustness regarding validation set
- Performance on long-form generation tasks and image captioning tasks

**Reviewer aoQh:**
- Comparisons with related works (Short-LVLM, Skip-Vision)
- Evaluation on video understanding and image captioning tasks
- Clarifications on application scope of DOP
- Clarifications on Qwen2.5-VL and InternVL3 comparison settings
- Stronger justification for the depth-wise pruning constraint

**Reviewer ZZ6f:**
- Comparisons with token-level early exit and dynamic computation allocation works
- Additional evidence for the depth redundancy assumption
- Clarifications on policy space size
- Validation of additive approximation

In the following part, we provide integrated responses to four key questions that are most valuable for comprehensively evaluating our DOP, including:
- Comparisons with Additional Related Works
- Validation of Depth-wise Pruning
- Validation of Additive Approximation
-  Additional Experimental Results.

---

> ### Author Response · Authors · 2025-12-01
> **1. Discussion with Additional Related Works**
>
> We appreciate the reviewers for highlighting relevant works including Short-LVLM, Skip-Vision and dynamic computation works. While we acknowledge similarities in the general improvement direction, which validates our motivation and design philosophy, we respectfully disagree that this diminishes our technical novelty. More fundamentally, these works target different objectives and scenarios compared to DOP, requiring model modifications (e.g., routers, token selectors) with substantial training data, making them essentially inapplicable for direct adaptation to achieve our core objective: MLLM acceleration with minimal resources.
>
> **Comparison with Short-LVLM [1]:** Short-LVLM performs layer pruning that modifies model architecture, while DOP preserves the original model structure intact, only changing computation by pruning operations for specific tokens within modules. Methodologically, Short-LVLM does not propose a comparable "operation pruning" framework. This architectural modification approach prevents direct application to our resource-efficient acceleration objective.
>
> **Comparison with Skip-Vision [2]:** DOP differs in two fundamental aspects. First, regarding operation pruning: while Skip-Vision implicitly achieves FFN module operation pruning for visual tokens, it remains limited exclusively to FFN modules without proposing a comprehensive operation pruning framework. Second, regarding practical deployment: Skip-Vision introduces new adaptive summary layers requiring extensive data and computational resources for SFT retraining, significantly limiting practical usability. In contrast, DOP requires only minimal resources for policy optimization and demonstrates strong cross-model generalization, enabling plug-and-play compatibility.
>
> **Comparison with dynamic computation works:** These works require substantial training data and architectural modifications, contrasting with DOP's resource efficiency.
> - Routing Experts [3] requires expensive three-stage training with 333k samples due to Router and Adapter components, while DOP achieves strong generalization with as few as 25 samples;
> - Accelerating BERT Inference [4] requires additional Token-Level Off-Ramps and extensive fine-tuning data, addressing intra-modal token redundancy whereas DOP targets inter-modal redundancy differences;
> - SkipGPT [5] introduces routers requiring 850,000 training samples, preventing few-shot fine-tuning or plug-and-play deployment. These design choices make them unsuitable for direct adaptation to resource-constrained MLLM acceleration scenarios.
>
> In summary, our DOP achieves MLLM acceleration without model architecture modifications, requiring only minimal optimization data and computational resources in an essentially plug-and-play manner. In contrast, these related works target scenarios with substantial fine-tuning data and computational resources to support architectural modifications for acceleration, making them inapplicable for low-resource MLLM acceleration. Despite similarities in high-level design philosophy, the specific methodological designs, application scenarios, and practical values are completely non-overlapping, therefore these works do not impact DOP's novelty.
>
> 1. Ma, Ji, et al. "Short-LVLM: Compressing and Accelerating Large Vision-Language Models by Pruning Redundant Layers." Proceedings of the 33rd ACM International Conference on Multimedia. 2025.
>
> 2. Zeng, Weili, et al. "Skip-Vision: Efficient and Scalable Acceleration of Vision-Language Models via Adaptive Token Skipping." arXiv preprint arXiv:2503.21817 (2025).
>
> 3. Wu, Qiong, et al. "Routing experts: Learning to route dynamic experts in multi-modal large language models." arXiv preprint arXiv:2407.14093 (2024).
>
> 4. Li, Xiaonan, et al. "Accelerating bert inference for sequence labeling via early-exit." Proceedings of the 59th annual meeting of the Association for Computational Linguistics and the 11th international joint conference on natural language processing (volume 1: long papers). 2021.
>
> 5. Zhao, Anhao, et al. "SkipGPT: Dynamic Layer Pruning Reinvented with Token Awareness and Module Decoupling." arXiv preprint arXiv:2506.04179 (2025).

---

> ### Author Response · Authors · 2025-12-01
> **2. Validation of Depth-wise Pruning - Part I**
>
> Our depth-wise assumption is a coarse-grained simplification based on the commonly observed empirical pattern that deeper layers tend to exhibit greater redundancy than shallower layers [6]. While this assumption may not hold strictly at every individual layer, it represents a practical approximation that enables efficient pruning optimization.
>
> 6. Gromov, Andrey, et al. "The unreasonable ineffectiveness of the deeper layers." arXiv preprint arXiv:2403.17887 (2024).
>
> In the following sections, we conduct two experiments to verify: 1) the assumption that deeper operations within the same module-group type are more redundant largely holds; and 2) despite this assumption not being strictly precise (i.e., being a coarse approximation), it does not prevent DOP from achieving near-optimal policies. We also updated those experiments in the **Section F.7 of the revised paper**.
>
> **Experiment 1: Pruning Consecutive Operations at Different Depths**
>
> We prune consecutive operations of the same module and group type at various layer positions to directly assess operation redundancy at different depths through their impact on downstream task performance. We first prune a single MHA/MLP operation for visual tokens on LLaVA-1.5-7B:
>
> **Table: Performance of Pruning Single Operation on LLaVA-1.5-7B**
>
> | Operation Pruned | Dataset | $l$=1 | $l$=4 | $l$=8 | $l$=12 | $l$=16 | $l$=20 | $l$=24 | $l$=28 | $l$=32 |
> |--------------------|---------|-------|-------|-------|--------|--------|--------|--------|--------|--------|
> | $\text{(}g_v,l,\text{MHA}\text{)}$ | TextVQA | 55.8 | 57.2 | 57.2 | 56.6 | 58.1 | 57.9 | 57.9 | 58.1 | 58.1 |
> |                                      | SeedBench | 26.1 | 66.2 | 65.4 | 64.3 | 66.1 | 66.2 | 66.1 | 66.2 | 66.2 |
> | $\text{(}g_v,l,\text{MLP}\text{)}$ | TextVQA | 58.3 | 58.2 | 57.9 | 58.2 | 58.0 | 58.3 | 58.1 | 58.1 | 58.3 |
> |                                      | SeedBench | 66.1 | 66.0 | 65.5 | 66.0 | 66.1 | 66.1 | 66.1 | 66.1 | 66.1 |
>
> We further prune three consecutive MHA/MLP operations for visual tokens at different positions to make the redundancy pattern more pronounced:
>
> **Table: Performance of Pruning 3 Consecutive Operations on LLaVA-1.5-7B**
>
> | Operation Pruned | Dataset | $l$=1,2,3 | $l$=4,5,6 | $l$=8,9,10 | $l$=12,13,14 | $l$=16,17,18 | $l$=20,21,22 | $l$=24,25,26 | $l$=28,29,30 |
> |--------------------|---------|------------|-----------|------------|--------------|--------------|--------------|--------------|--------------|
> | $\text{(}g_v,l,\text{MHA}\text{)}$ | TextVQA | 44.9 | 55.3 | 53.5 | 56.5 | 56.5 | 56.7 | 58.0 | 57.7 |
> |                                      | SeedBench | 60.4 | 62.9 | 59.7 | 62.2 | 66.1 | 66.2 | 66.1 | 65.8 |
> | $\text{(}g_v,l,\text{MLP}\text{)}$ | TextVQA | 58.3 | 57.3 | 55.4 | 57.6 | 57.7 | 58.2 | 58.2 | 58.4 |
> |                                      | SeedBench | 66.1 | 64.8 | 63.2 | 66.1 | 66.2 | 66.1 | 66.1 | 66.1 |
>
> We also conducted this experiment on Qwen2.5-VL-7B:
>
> **Table: MME Performance of Pruning 3 Consecutive Operations on Qwen2.5-VL-7B**
>
> | Operation Pruned | $l$=1,2,3 | $l$=4,5,6 | $l$=8,9,10 | $l$=12,13,14 | $l$=16,17,18 | $l$=20,21,22 | $l$=24,25,26 |
> |--------------------|-----------|-----------|------------|--------------|--------------|--------------|--------------|
> | $\text{(}g_v,l,\text{MHA}\text{)}$ | 1656 | 2148 | 2194 | 1959 | 2213 | 2295 | 2295 |
> | $\text{(}g_v,l,\text{MLP}\text{)}$ | 1816 | 2236 | 2244 | 2242 | 2308 | 2303 | 2297 |
>
> Generally, pruning deeper layers results in smaller performance drops, indicating that the pruned operations are more redundant. The pattern of deeper operations being more redundant becomes more evident when pruning three consecutive operations. On Qwen2.5-VL-7B, the trend generally holds with some variations in MHA operations but clear patterns in MLP operations, confirming that our depth-wise assumption is largely reasonable across different model architectures.

---

> > ### Author Response · Authors · 2025-12-01
> > **2. Validation of Depth-wise Pruning - Part II**
> >
> > **Experiment 2: Policy Perturbation Analysis**
> >
> > We conduct policy perturbation experiments to validate the near-optimality of our DOP constraint. Starting from our original DOP policy, we create alternative non-consecutive pruning strategies by randomly swapping $n$ operations within the same module-group type: removing $n$ retained operations and keeping $n$ originally pruned operations, thereby breaking the depth-wise constraint. We repeat this process 10 times to sample alternative policies.
> >
> > This reverse validation approach is straightforward: if our DOP policy were far from optimal, we should easily find significantly better policies through random perturbations. Conversely, if DOP consistently outperforms the majority of these random alternatives, this provides strong evidence for its near-optimality and validates our depth-wise constraint as a fundamental efficiency principle rather than an arbitrary design choice.
> >
> > **Results on LLaVA-1.5-7B at 28% TFLOPs:**
> >
> > | Policy | Metric | TextVQA | GQA | SeedBench |
> > |--------|--------|---------|-----|-----------|
> > | Original DOP | - | 55.8 | 59.6 | 63.6 |
> > | Swap 1 MHA OP | Best | 55.8 | 59.8 | 63.6 |
> > |  | Mean | 54.3 | 57.7 | 61.4 |
> > | Swap 3 MHA OP | Best | 55.1 | 58.9 | 62.9 |
> > |  | Mean | 51.8 | 54.7 | 59.5 |
> > | Swap 1 MLP OP | Best | 55.9 | 59.7 | 63.6 |
> > |  | Mean | 55.3 | 58.8 | 63.0 |
> > | Swap 3 MLP OP | Best | 55.4 | 59.3 | 63.2 |
> > |  | Mean | 52.7 | 56.1 | 60.6 |
> >
> > For swapping 1 operation, while we can occasionally find slightly better policies, our original DOP policy significantly outperforms the average performance of perturbed policies. When we increase the perturbation magnitude by swapping 3 operations, we can hardly find any policy that outperforms DOP, and the average performance of perturbed policies deteriorates further from the original DOP baseline.
> >
> > **Results on Qwen2.5-VL-7B at 26% TFLOPs:**
> >
> > Since our original DOP policy does not prune any MHA operations ($d_A = 28$), we only performed perturbations on MLP operations:
> >
> > | Policy | Metric | TextVQA | ChartQA | MME |
> > |--------|--------|---------|---------|-----|
> > | Original DOP | - | 79.8 | 83.1 | 2351 |
> > | Swap 1 MLP OP | Best | 80.2 | 82.6 | 2347 |
> > |  | Mean | 77.5 | 81.6 | 2331 |
> > | Swap 3 MLP OP | Best | 77.1 | 81.4 | 2316 |
> > |  | Mean | 73.8 | 78.5 | 2236 |
> >
> > The results demonstrate that perturbed policies consistently fail to surpass our original DOP policy. Even in the best case, swapping 1 MLP operation only marginally exceeds performance on TextVQA (80.2 vs 79.8), while underperforming on other metrics. When increasing the perturbation magnitude (swap 3 operations), performance degrades significantly, further confirming the near-optimality of our DOP policy.
> >
> > These consistent results across different models provide strong evidence that our depth-wise constraint indeed captures fundamental principles of MLLMs, rather than being an arbitrary design choice.

---

> ### Author Response · Authors · 2025-12-01
> **3. Validation of Additive Approximation**
>
> We validate our additive approximation through correlation analysis in Section E.1. Additionally, we include a direct validation in Section F.3 of the revised paper by evaluating joint KL divergence in the policy subspace through grid search for optimal policies and comparing with policies obtained via additive approximation. The minimal differences in downstream task performance demonstrate the reasonableness of our additive approximation.
>
> **Experiment : Additive Approximation vs. Direct Joint Divergence Optimization**
>
> We conducted a grid search experiment in a policy subspace based on direct joint divergence evaluation. We create a sufficiently small policy space that allows us to exhaustively evaluate the joint divergence of each policy by sampling MHA and MLP depths every 2 steps ($d_A, d_P \in$ {0,2,4,6,...}) and visual token counts every 16 steps ($n_v \in $ {0,16,32,...}). Given different TFLOPs targets, we first solved Equation 7 using additive approximation, then evaluated joint KL divergence for all constraint-satisfying policies on the same validation set, selecting the policy with minimum divergence. On LLaVA-1.5-7B with 28% and 23% target TFLOPs ratios, we found 658 and 384 valid policies respectively.
>
> | Optimization Method | $d_A$ | $d_P$ | $n_v$ | TextVQA | GQA | SeedBench |
> |---------------------|-----|-----|-----|---------|-----|-----------|
> | **28% TFLOPs** |  |  |  |  |  |  |
> | w/ Additive Approximation | 28 | 18 | 96 | 55.7 | 59.4 | 63.5 |
> | w/o Additive Approximation | 26 | 20 | 96 | 55.8 | 59.6 | 63.4 |
> | **23% TFLOPs** |  |  |  |  |  |  |
> | w/ Additive Approximation | 28 | 16 | 48 | 53.8 | 57.7 | 62.0 |
> | w/o Additive Approximation | 24 | 18 | 48 | 54.1 | 57.8 | 62.0 |
>
> The results show that direct grid search with joint KL divergence (w/o Additive Approximation) achieves slightly better performance, but policies with additive approximation achieve very similar results. This further validates our additive approximation approach and confirms that despite being a coarse approximation, it enables DOP to achieve near-optimal policies with significantly reduced computational cost.

---

> ### Author Response · Authors · 2025-12-01
> **4. Additional Experimental Results**
>
> **Image Captioning Tasks (Section E.6 in the revised paper):** We evaluated DOP's performance on image captioning tasks using NoCaps and TextCaps datasets. Experiments were conducted on LLaVA-1.5-7B with CDPruner for token reduction, comparing against CDPruner at equivalent TFLOPs using the lmm-eval framework. CIDEr score is reported.
>
> | TFLOPs | Method | NoCaps | TextCaps |
> |--------|--------|---------|----------|
> | 100% | Baseline | 1.043 | 0.992 |
> | 28% | CDPruner | 0.912 | 0.863 |
> |      | DOP+CDPruner | 0.948 | 0.883 |
> | 23% | CDPruner | 0.844 | 0.767 |
> |      | DOP+CDPruner | 0.881 | 0.790 |
>
> DOP consistently outperforms CDPruner on image captioning tasks, demonstrating its effectiveness beyond VQA scenarios.
>
> **Video Understanding Tasks (Section E.7 in the revised paper):** We followed VisPruner's video benchmark settings, applying DOP on Video-LLaVA, evaluating on TGIF-QA, MSVD-QA and MSRVTT-QA, using the first 1K samples from each benchmark and employing VisPruner as the visual token reduction method. Video-LLaVA processes 8 frames of 224-resolution video, totaling 2048 visual tokens.
>
> | Method | TGIF-QA |      | MSVD-QA |      | MSRVTT-QA |      | Average |      |
> |--------|---------|------|---------|------|-----------|------|---------|------|
> |        | Acc.    | Score| Acc.    | Score| Acc.      | Score| Acc.    | Score|
> | **Upper Bound, All 2048 Tokens (100%)** |
> | Video-LLaVA | 19.8 | 2.53 | 70.5 | 3.93 | 57.5 | 3.50 | 49.3 | 3.32 |
> | **Retain 16% TFLOPs** |
> | VisPruner | 15.9 | 2.41 | 69.3 | 3.92 | 55.6 | 3.45 | 46.9 | 3.26 |
> | DOP + VisPruner | 16.3 | 2.44 | 69.5 | 3.92 | 55.9 | 3.46 | 47.2 | 3.27 |
> | **Retain 11% TFLOPs** |
> | VisPruner | 14.1 | 2.35 | 65.4 | 3.79 | 54.1 | 3.41 | 44.5 | 3.18 |
> | DOP + VisPruner | 14.6 | 2.36 | 65.8 | 3.80 | 54.4 | 3.42 | 44.9 | 3.19 |
>
> DOP provides consistent improvements over VisPruner baseline across video understanding tasks. In the final version, we will include additional video benchmarks such as VideoMME and MLVU, along with more video models like LLaVA-Video to provide comprehensive validation of DOP's efficacy in the video domain.

---

### Meta-Review · Area_Chair_aRLd · 2026-01-07

**Summary:**

This paper proposes an operation pruning method for multimodal large language models. The authors employ heuristic techniques, such as depth-wise pruning and additive approximation, to reduce the optimization space. However, significant concerns persist regarding the limited theoretical justification for additive approximation and monotonic depth redundancy. Furthermore, the discussion and comparison with recent operation pruning studies are insufficient, which makes the paper's claims of novelty appear overstated. Consequently, the paper is not recommended for acceptance.

**Reviewer Concerns:**

**Addressed Concerns:**

1.	**Experiments on more Tasks (Reviewer DwzN, aoQh and ).** The authors successfully addressed concerns regarding the lack of empirical evaluation across diverse tasks by adding results on image caption and video understanding experiments.

**Outstanding Concerns:**

1.	**Limited theoretical justifications/insights (Reviewer GM9q, DwzN, aoQh, ZZ6f).** The authors have not adequately addressed the reviewers' concerns regarding the lack of theoretical justification and insight for the assumptions of additive approximation and monotonic depth redundancy. Although some empirical results were provided to support these assumptions, the theoretical foundations remain insufficient.

2.	**Limited comparison experiments (Reviewer aoQh, ZZ6f).** Although the authors addressed the missing related work highlighted by the reviewers in the rebuttal, a detailed comparison has not been provided. This omission further undermines the novelty of the proposed method.

3.	**The potential of DOP is not sufficiently explored. (Reviewer aoQh).** The authors claim that DOP can be applied to computationally intensive visual encoders such as Vision Transformers (ViTs), which is a major bottleneck for high-resolution image and video understanding. Including empirical results to support this claim would significantly strengthen the paper’s contribution.

**Reviewer Scores:**

- **Reviewer  GM9q:** The score is likely to remain unchanged (4), as the omission of relevant prior work undermines the novelty of the proposed method, and the authors have not offered substantial theoretical justification for their assumptions.
- **Reviewer DwzN: ** The score is likely to remain unchanged (6) since it is already positive and the authors have not offered substantial theoretical justification for their assumptions.
- **Reviewer aoQh:** The score is likely to remain unchanged (4), as the authors have not provided comparative results with the closely related works highlighted by the reviewer.
- **Reviewer ZZ6f:** The score is likely to remain unchanged (4), The score is likely to remain unchanged (4), as the authors have not provided comparative results with the closely related works highlighted by the reviewer, which weakens the novelty of this paper.

---

### Decision · Program_Chairs · 2026-01-26

Reject